# Escaping saddle points in zeroth-order optimization: two function evaluations suffice

## Abstract

Two-point zeroth order methods are important in many applications of zeroth-order optimization arising in robotics, wind farms, power systems, online optimization, and adversariable robustness to black-box attacks in deep neural networks, where the problem can be high-dimensional and/or time-varying. Furthermore, such problems may be nonconvex and contain saddle points. While existing works have shown that zeroth-order methods utilizing $\Omega(d)$ function valuations per iteration (with $d$ denoting the problem dimension) can escape saddle points efficiently, it remains an open question if zeroth-order methods based on two-point estimators can escape saddle points. In this paper, we show that by adding an appropriate isotropic perturbation at each iteration, a zeroth-order algorithm based on $2m$ (for any $1 \leq m \leq d$) function evaluations per iteration can not only find $\epsilon$-second order stationary points polynomially fast, but do so using only $\tilde{O}(d/\epsilon^{2.5})$ function evaluations.

## 1 Introduction

Two-point (or in general $2m$-point, where $1 \leq m < d$ with $d$ being the problem dimension) estimators, which approximate the gradient using two (or $2m$) function evaluations per iteration, have been widely studied by researchers in the zeroth-order optimization literature, in convex (Nesterov and Spokoiny, 2017; Duchi et al., 2015; Shamir, 2017), nonconvex (Nesterov and Spokoiny, 2017), online (Shamir, 2017), as well as distributed settings (Tang et al., 2019). A key reason for doing so is that for applications of zeroth-order optimization arising in robotics (Li et al., 2022), wind farms (Tang et al., 2020a), power systems (Chen et al., 2020), online (time-varying) optimization (Shamir, 2017), learning-based control (Malik et al., 2019; Li et al., 2021), and improving adversarial robustness to black-box attacks in deep neural networks (Chen et al., 2017), it may be costly or impractical to wait for $\Omega(d)$ (where $d$ denotes the problem dimension) function evaluations per iteration to make a step. This is especially true for high-dimensional and/or time-varying problems. See Appendix A for more discussion. However, despite the advantages of zeroth-order methods with two-point estimators, there has been a lack of existing work studying the ability of two-point estimators to escape saddle points in nonconvex optimization problems. Since nonconvex problems arise often in practice, it is crucial to know if two-point algorithms can efficiently escape saddle points of nonconvex functions and converge to second-order stationary points.

To motivate the challenges of escaping saddle points using two-point zeroth-order methods, we begin with a review of escaping saddle points using first-order methods. The problem of efficiently escaping saddle points in deterministic first-order optimization (with exact gradients) has been carefully studied in several earlier works (Jin et al., 2017; 2018). A key idea in these works is the injection of an isotropic perturbation whenever the gradient is small, facilitating escape from a saddle if a negative curvature direction exists even without actively identifying the direction. However, the analysis of efficient saddle point escape for stochastic gradient methods is often more complicated. In general, the behavior of the stochastic gradient near the saddle point can be difficult to characterize. Hence, strong concentration assumptions are typically made on the stochastic gradients being used, such as subGaussianity, boundedness of the variance or a bounded gradient estimator (Ge et al., 2015; Daneshmand et al., 2018; Xu et al., 2018; Fang et al., 2019; Roy et al., 2020; Vlaski and Sayed, 2021b), creating an analytical issue when such idealized assumptions fail to hold.

Indeed, though zeroth-order methods can be viewed as stochastic gradient methods, common zeroth order estimators, such as two-point estimators (Nesterov and Spokoiny, 2017), are not subGaussian, and can have unbounded variance. For instance, it can be shown that the variance of the two-point

estimator is on the order of $\Omega(d\|\nabla f(x)\|^2)$ (Nesterov and Spokoiny, 2017), with both a dependence on the problem dimension $d$ as well as on the norm of the gradient, which can be unbounded. Due to non-subGaussianity and unboundedness, it is tricky to bound the effect of such zeroth-order estimators and establish tight concentration inequalities that facilitate its escape near saddle points. In addition, the large variance of the zeroth-order estimator is also an issue away from saddles when the gradient is large. While this is not an issue to show function improvement in expectation, as we discuss later, this becomes an issue when guaranteeing high probability bounds.

Due to these difficulties, previous works on escaping saddle points in zeroth-order optimization have exclusively focused on approaches requiring $\Omega(d)$ function evaluations per iteration to accurately estimate the gradient (Bai et al., 2020; Vlatakis-Gkaragkounis et al., 2019), or in some cases negative curvature directions (Zhang et al., 2022; Lucchi et al., 2021) or the Hessian itself (Balasubramanian and Ghadimi, 2022), reducing in a sense the zeroth-order problem back to a first-order one. However, as explained earlier, two-point or $2m$-point zeroth-order algorithms are important for high-dimensional and/or time-varying problems in many applications areas. This raises an important question: **Can two-point zeroth-order methods escape saddle points and reach approximate second order stationary points efficiently?**

**Our Contribution.** In this work, we show that by adding an appropriate isotropic perturbation at each iteration, a zeroth-order algorithm based on *any* number $m$ of pairs ($m$ ranging from 1 to $d$) of function evaluations per iteration can not only find $(\epsilon, \sqrt{\epsilon})$-second order stationary points (cf. the definition later in Definition 1) polynomially fast, but do so using only $\tilde{O}(\mathrm{polylog}(\frac{1}{\delta})d/\epsilon^{2.5})$ function evaluations, with a probability of at least $1 - \delta$. In particular, this proves that using a single two-point zeroth-order estimator at each iteration (with appropriate perturbation) suffices to efficiently escape saddle points in zeroth-order optimization, with high probability. Moreover, for functions that are $(\epsilon, \psi)$ strict-saddle (see Definition 3 for a definition of strict saddle functions), our results become $\tilde{O}(\mathrm{polylog}(\frac{1}{\delta})d/\psi\epsilon^2)$, which is a significant improvement when $\psi \gg \epsilon$; strict saddle functions have been identified as an important class of functions in nonconvex optimization, with several well-known examples such as tensor decomposition (Ge et al., 2015), dictionary learning and phase retrieval (Sun et al., 2015). A comparison of our results with existing zeroth-order and first-order methods is shown in Table 1. We also provide numerical results in Appendix G showing that our proposed two-point algorithm requires fewer total function evaluations to converge than zeroth order methods that use $2d$ function evaluations per iteration, for a nonconvex test function proposed in Du et al. (2017).

To overcome the theoretical challenges that were discussed earlier, we i) first show, via a careful analysis, that zeroth order methods can make function value improvement across iterates with large gradients with high probability, even when only a single two-point estimator (which can have significant variance at large gradients) is used per iteration. ii) Second, near saddle points, we overcome issues caused by the unbounded variance and non-subGaussinity of zeroth-order gradient estimators by developing new technical tools, including novel martingale concentration inequalities involving Gaussian vectors, to tightly bound such terms. In turn, this allows us to show that the noise emanating from the zeroth-order estimators will not overwhelm the effect of the additional isotropic perturbative noise, facilitating escape along negative curvature directions. To the best of our knowledge, both analyses are novel, and may be independent contributions on their own.

*Related Work.* Due to space considerations, we defer a full discussion of related work to Appendix A.

## 2 PROBLEM SETUP

We make the following assumptions on the class of functions $f : \mathbb{R}^d \to \mathbb{R}$ which we consider.

**Assumption 1** (Properties of $f$)**.** *We suppose that $f : \mathbb{R}^d \to \mathbb{R}$ satisfies the following properties:*

1. *$f$ is twice-differentiable and lower bounded, i.e. $f^* := \min_x f(x) > -\infty$.*

2. *$f$ is $L$-Lipschitz, i.e. $\|\nabla f(x) - \nabla f(y)\| \le L\|x - y\| \quad \forall x, y \in \mathbb{R}^d$*

3. *$f$ is $\rho$-Hessian Lipschitz, i.e. $\left\|\nabla^2 f(x) - \nabla^2 f(y)\right\| \le \rho\|x - y\| \quad \forall x, y \in \mathbb{R}^d$.*

In our work, we focus on finding approximate second order stationary points, defined below.

**Definition 1.** A point $x \in \mathbb{R}^d$ is an $(\epsilon, \varphi)$-second order stationary point if

$$\|\nabla f(x)\| < \epsilon, \quad \text{and} \quad \lambda_{\min}(\nabla^2 f(x)) > -\varphi.$$

| | | Iteration Complexity | Fun. evaluations per iter. |
|---|---|---|---|
| First-order | Jin et al. (2017) (deterministic) | $\tilde{O}\left(\frac{1}{\epsilon^2}\right)$ | — |
| | Fang et al. (2019) (SGD) | $\tilde{O}\left(\frac{1}{\epsilon^{3.5}}\right)$ | — |
| Zeroth-order | Bai et al. (2020) | $\tilde{O}\left(\frac{1}{\epsilon^2}\right)$ | $\tilde{\Omega}\left(\frac{d^2}{\epsilon^8}\right)$ |
| | Vlatakis-Gkaragkounis et al. (2019) | $\tilde{O}\left(\frac{1}{\epsilon^2}\right)$ | $\tilde{\Omega}(d)$ |
| | Balasubramanian and Ghadimi (2022) | $\tilde{O}\left(\frac{1}{\epsilon^{3/2}}\right)$ | $\tilde{O}\left(\frac{d}{\epsilon^2} + \frac{d^4}{\epsilon}\right)$ |
| | Lucchi et al. (2021)[†] | $\tilde{O}\left(\frac{1}{\epsilon^2}\right)$ | $\tilde{O}\left(\frac{d}{\epsilon^{2/3}}\right)$ |
| | Zhang et al. (2022) | $\tilde{O}\left(\frac{1}{\epsilon^2}\right)$ | $\tilde{\Omega}(d)$ |
| | Algorithm 1 (this paper)[‡] | $\tilde{O}\left(\frac{d}{\epsilon^2\bar{\psi}m}\right)$ | $2m$ |

Table 1: Selected comparison of convergence results to $(\epsilon, O(\sqrt{\epsilon}))$-second order stationary points in smooth, nonconvex functions; for [†], the convergence is to $(\epsilon, \epsilon^{2/3})$-second order stationary points. For [‡], the term $\bar{\psi}$ in the denominator is (i) $\psi$ when the function $f$ is $(\epsilon, \psi)$-strict saddle for a $\psi > O(\sqrt{\epsilon})$ (see Definition 3 for a definition) and (ii) $O(\sqrt{\epsilon})$ if otherwise.

We define an $(\epsilon, \varphi)$-approximate saddle point as follows.

**Definition 2.** A point $x \in \mathbb{R}^d$ is an $(\epsilon, \varphi)$-approximate saddle point, if

$$\|\nabla f(x)\| < \epsilon, \quad \text{and} \quad \lambda_{\min}(\nabla^2 f(x)) \leq -\varphi.$$

Following past convention (Jin et al., 2019a), we will focus in particular on escaping $(\epsilon, \sqrt{\rho\epsilon})$-saddle points. For notational simplicity, in following text, we refer to $(\epsilon, \sqrt{\rho\epsilon})$-saddle points simply as $\epsilon$-saddle points and $(\epsilon, \sqrt{\rho\epsilon})$-second order stationary points as $\epsilon$-second order stationary points. Beyond the definition of $\epsilon$-approximate saddle points above, it is known that many nonconvex functions with saddle points, such as orthogonal tensor decomposition (Ge et al., 2015), phase retrieval and dictionary learning (Sun et al., 2015), satisfy what is known as a *strict saddle* condition (Ge et al., 2015). For the Hessians of the saddle points of such functions, there is always a strict negative eigenvalue whose magnitude is bounded from below. We provide a precise definition below.

**Definition 3.** A twice-differential function $f(x)$ is $(\epsilon, \psi)$-strict saddle, if for any point $x$, at least one of the following is true:

$$\|\nabla f(x)\| \geq \epsilon \quad \text{or} \quad \lambda_{\min}(\nabla^2 f(x)) \leq -\psi$$

As a corollary of Theorem 1 (to be stated later), for functions $f$ which are $(\epsilon, \psi)$ strict saddle, assuming that $\psi \geq \sqrt{\rho\epsilon}$, the sample complexity of our algorithm scales as $\tilde{\Omega}\left(\frac{d\max\{L^2, L\}(f(x_0) - f^*)}{m\epsilon^2\psi}\right)$, which scales as $\tilde{\Omega}\left(\frac{d}{m\epsilon^2}\right)$ when $\psi$ is of size $\Omega(1)$. Thus, in this setting, for two-point estimators, where $m = 1$, the dependence on $d$ and $\epsilon$ in our sample complexity (as measured by function evaluations) matches that achieved by the algorithms in Vlatakis-Gkaragkounis et al. (2019); Zhang et al. (2022), which have to use $2d$ function evaluations per iteration to estimate the gradient.

In our work, we consider the following batch symmetric two-point zeroth-order estimator.

**Definition 4** ((Batch) two-point zeroth-order estimator with perturbation)**.** We define a $m$-batch two-point zeroth order estimator as follows:

$$g_u^{(m)}(x) := \frac{1}{m}\sum_{i=1}^{m}\frac{f(x + uZ_i) - f(x - uZ_i)}{2u}Z_i, \tag{1}$$

where $Z_i \overset{i.i.d}{\sim} N(0, I)$, and $u > 0$ is a smoothing radius.

Such $2m$ zeroth-order gradient estimators have frequently been studied in zeroth-order optimization works (see e.g. Nesterov and Spokoiny (2017)). To facilitate efficient escape from saddle points, our proposed Algorithm 1 adds isotropic perturbation at each iteration.

---

**Algorithm 1:** Zeroth-order perturbed gradient descent (ZOPGD)

**input :** $x_0$, horizon $T$, step-size $\eta$, smoothing radius $u$, perturbation radius $r$, batch size $m$
**for** *step* $t = 0, \dots, T_1$ **do**

    Sample $Z^{(m)} = \{Z_{t,i}\}_{i=1}^m \sim N(0, I)$ to compute $g_u^{(m)}(x_t)$.

    Update $x_{t+1} = x_t - \eta \left( g_u^{(m)}(x_t) + Y_t \right)$, where $Y_t \sim N(0, \frac{r^2}{d} I)$

---

We now state an informal version of our main result, and follow that with a few remarks.

**Theorem 1** (Main result, informal version of Theorem 2). *Consider running Algorithm 1. Let $\tilde{O}$ hide polylogarithmic terms in $\delta$ and other parameters. Suppose $\delta \in (0, 1/e]$. Suppose $\sqrt{\rho\epsilon} \leq \min\{1, L\}$[1], such that $\bar{\psi} \leq \min\{1, L\}$, where*

$$\bar{\psi} := \begin{cases} \min\{\psi, 1, L\} & \text{if } f(\cdot) \text{ is } (\epsilon, \psi)\text{-strict saddle for any } \psi > \sqrt{\rho\epsilon} \\ \sqrt{\rho\epsilon} & \text{otherwise}. \end{cases} \tag{2}$$

*Suppose $u = \tilde{O}\left( \frac{\min\{\sqrt{\epsilon}, \sqrt{r}\}}{\sqrt{\rho}d} \right)$, $r = \tilde{O}(\epsilon)$, $\eta = \tilde{O}\left( \frac{m\bar{\psi}}{d\max\{L, L^2\}} \right)$, Then, in*

$$T = \tilde{\Omega}\left( \frac{(f(x_0) - f^*)}{\eta\epsilon^2} + \frac{\rho^2(f(x_0) - f^*)}{\eta\bar{\psi}^4} \right)$$

$$= \tilde{\Omega}\left( \frac{d\max\{L, L^2\}(f(x_0) - f^*)}{m\bar{\psi}\epsilon^2} + \frac{d\max\{L, L^2\}\rho^2(f(x_0) - f^*)}{m\bar{\psi}^5} \right)$$

$$= \tilde{\Omega}\left( \frac{d\max\{L, L^2\}\rho^2(f(x_0) - f^*)}{m\bar{\psi}\epsilon^2} \right)$$

*iterations (with each iteration using $2m$ function evaluations), with probability at least $1 - \delta$, at least half the iterates are $\epsilon$-approximate second-order stationary points.*

*Remark* 1. As the choice of $\eta$ in Proposition 4 (Appendix D) and Theorem 2 (Appendix F) respectively imply, the $\tilde{\Omega}\left( \frac{f(x_0) - f^*}{\eta\epsilon^2} \right)$ term in the sample complexity comes from the large gradient iterations (Proposition 4), whereas the $\tilde{\Omega}\left( \frac{\rho^2(f(x_0) - f^*)}{\eta\bar{\psi}^4} \right)$ term comes from the escape saddle point phase.

**Comparison to gradient-based methods.** For first-order escape saddle point algorithms, standard perturbation-based methods (without acceleration) can find a $(\epsilon, O(\sqrt{\epsilon}))$-second-order stationary point using $\tilde{O}(1/\epsilon^2)$ iterations for deterministic GD (Jin et al., 2019a), while for standard SGD the best-known rates are slower at $\tilde{O}(1/\epsilon^{3.5})$ (Fang et al., 2019). In contrast, our sample complexity (as measured by the total number of function evaluations) is $\tilde{O}\left( \frac{d}{\epsilon^2\bar{\psi}} \right)$, where $\bar{\psi}$ is defined in Eq. (2). The extra (linear) dependence on $d$ is typical for zeroth-order algorithms (see e.g. Nesterov and Spokoiny (2017)); intuitively, gradient calculation for $d$-dimensional functions requires $O(d)$ calculations agnostically, so it makes sense that zeroth-order algorithms requires $d$ times more iterations. For general non strict-saddle functions, our dependence on $\epsilon$ sits between that of the deterministic methods and SGD methods, and suggests the benefit of a specialized treatment of zeroth-order methods over considering them simply as a subclass of SGD methods. Moreover, for $(\epsilon, \psi)$- strict-saddle functions where $\psi = \Omega(1)$, our sample complexity becomes $\tilde{O}(\frac{d}{\epsilon^2})$, with an $\epsilon$ dependence that matches that of the best existing sample complexity for non-accelerated first-order escape saddle point methods Jin et al. (2017)

**Comparison to existing zeroth-order methods.** As Table 1 suggests, our sample complexity significantly outperforms that of Bai et al. (2020) and Balasubramanian and Ghadimi (2022), and also

---

[1]In our paper, we focus on the case $\sqrt{\rho\epsilon} \leq L$; otherwise, by the $L$-Lipschitz assumption, $\lambda_{\min}(\nabla^2 f(x)) \geq -L$ for all $x \in \mathbb{R}^d$, which implies $\epsilon$-first order stationary points are also $\epsilon$-second order stationary points.

that in Lucchi et al. (2021), which is a random search method. The sample complexity in Vlatakis-Gkaragkounis et al. (2019); Zhang et al. (2022) outperform our method, with a function evaluation complexity of $\tilde{O}\left(\frac{d}{\epsilon^2}\right)$. However, for for $(\epsilon, \psi)$- strict-saddle functions where $\psi = \Omega(1)$, our sample complexity becomes $\tilde{O}(\frac{d}{\epsilon^2})$, which matches the sample complexity in Vlatakis-Gkaragkounis et al. (2019); Zhang et al. (2022). Moreover, a key limitation of their methods is a requirement to use $\Omega(d)$ function evaluations to estimate the gradient at each iteration, which may not be practical in realistic applications when $d$ is large. In contrast, our method supports any number of function evaluations at each iteration between $1$ to $d$. Moreover, numerically, we found that for a test nonconvex function proposed in Du et al. (2017), our method (with two-point estimators) takes fewer function evaluations to escape saddle points and converge to the global minimum than the methods in Vlatakis-Gkaragkounis et al. (2019); Zhang et al. (2022).

## 3    PROOF STRATEGY AND KEY CHALLENGES IN THE ZEROTH-ORDER SETTING

Broadly speaking, our proof include two major parts, i) characterizing the progress made in iterations when the gradient is large (which we can define to be iterations $t$ where $\|\nabla f(x_t)\| \geq \epsilon$) (Section 3.1), ii) and iterations when we are at an $\epsilon$-approximate saddle point (where progress may be made along the negative eigendirection of the Hessian matrix) (Section 3.2). While the approach is similar to the first-order case (e.g. Jin et al. (2019a)), the zeroth-order setting brings forth several challenges as we explain later in the individual subsections. In the rest of this section, we explain these challenges, sketch out our high-level proof outlines, and provide statements of the main technical results. Note that due to the space limit, the full proof is provided in the Appendix.

### 3.1    SHOWING FUNCTION DECREASE WHEN GRADIENTS ARE LARGE

**Challenge.** Due to the noise in two-point (or $2m$ where $m$ is a small constant) zeroth-order gradient, even when the gradient is large, it may not always be possible to make progress at each iteration, especially when $m < d$ is used in the gradient estimation equation in Eq. (1). While it is tempting to use an expectation-based argument to handle this issue, it is known that expectation-based function decrease arguments are insufficient for escape saddle point purposes (see e.g. Proposition 1 in Ziyin et al. (2021)). We tackle this issue by using high-probability arguments instead; we note that achieving these high-probability bounds is highly nontrivial due to the large variance of the two-point zeroth-order estimator (scaling with $d$ times the squared norm of the gradient). Hence, any single iteration of the zeroth-order method may in fact lead to a function increase rather than decrease.

**High-level proof outline.** **(i)** We first characterize the function value change for our proposed algorithm (Lemma 1). **(ii)** Next, we tackle the issue of the possibility that the function value might increase for any given iteration. The key idea here is that across any small consecutive number of iterations, there will be one iteration where the zeroth-order estimator is sufficiently aligned with the gradient direction (Lemma 14 in Appendix D). **(iii)** Along with a series of other technical results in Appendix D, we then show that the function makes sufficient progress across the duration of the algorithm, with high probability (Proposition 1). To more concretely illustrate the key analytical challenge, we next introduce the following function decrease lemma, proved in Appendix D.

**Lemma 1** (Function decrease for batch zeroth-order optimization). *Suppose at each time $t$, the algorithm performs the update step (with batch-size parameter $1 \leq m \leq d$)*

$$x_{t+1} = x_t - \eta \left( g_u^{(m)}(x_t) + Y_t \right),$$

*where*

$$g_u^{(m)}(x_t) = \frac{1}{m} \sum_{i=1}^{m} \frac{f(x_t + uZ_{t,i}) - f(x_t - uZ_{t,i})}{2u} Z_{t,i},$$

*where each $Z_{t,i}$ is drawn i.i.d from $N(0, I)$, $u > 0$ is the smoothing radius, and $Y_t \sim N(0, \frac{r^2}{d}I)$ with $r > 0$ denoting the perturbation radius.*

*Then, there exist absolute constants $c_1 > 0, C_1 \geq 1$ such that, for any $T \in \mathbb{Z}^+$ and $T \geq \tau > 0$, $\alpha > 0$ and $\delta \in (0, 1/e]$, upon defining $\mathcal{H}_{0,\tau}(\delta)$ to be the event on which the inequality*

$$f(x_\tau) - f(x_0) \leq -\frac{3\eta}{4} \sum_{t=0}^{\tau-1} \frac{1}{m} \sum_{i=1}^{m} \left|Z_{t,i}^\top \nabla f(x_t)\right|^2 + \left(\frac{\eta}{\alpha} + \frac{c_1 L \eta^2 \chi^3 d}{m}\right) \sum_{t=0}^{\tau-1} \|\nabla f(x_t)\|^2$$

$$+ \tau \eta u^4 \rho^2 \cdot c_1 d^3 \left( \log \frac{T}{\delta} \right)^3 + \tau L \eta^2 u^4 \rho^2 \cdot c_1 d^4 \left( \log \frac{T}{\delta} \right)^4$$

$$+ \eta c_1 r^2 (\alpha + \eta L) \log \frac{T}{\delta} + \tau c_1 L \eta^2 r^2 \tag{3}$$

*is satisfied (where $\chi := \log(C_1 d m T / \delta)$), we have*

$$\mathbb{P}(\mathcal{H}_{0,\tau}(\delta)) \geq 1 - \frac{(\tau + 4)\delta}{T}, \qquad \mathbb{P}(\cap_{\tau=1}^{\tau'} \mathcal{H}_{0,\tau}(\delta)) \geq 1 - \frac{5\tau'\delta}{T}$$

*for any $0 \leq \tau' \leq T$.*

Our goal is to show that we can arrive at a contradiction $f(x_T) < \min_x f(x)$ when there is a large number of steps at which $\|\nabla f(x_t)\| \geq \epsilon$ (Proposition 1). As we can see from Eq. (3), this implies that we need to prove a lower bound of the form

$$\sum_{t=0}^{T-1} \frac{1}{m} \sum_{i=1}^{m} \left\| Z_{t,i}^\top \nabla f(x_t) \right\|^2 \geq \Omega \left( \frac{1}{\alpha} + \frac{c_1 L \eta \chi^3 d}{m} \right) \sum_{t=0}^{T-1} \|\nabla f(x_t)\|^2 \tag{4}$$

for some $\alpha$ which is not too large (an example would be picking $\alpha$ such that it only scales logarithmically in the problem parameters). However, it is tricky to prove such a lower-bound in the zeroth-order setting. In particular, for small batch-sizes $m$, $\frac{1}{m} \sum_{i=1}^{m} \left\| Z_{t,i}^\top \nabla f(x_t) \right\|^2$ could be small even as $\|\nabla f(x_t)\|^2$ is large; this is because for each $i \in [m]$, $Z_{t,i}$ could have a negligible component in the $\nabla f(x_t)$ direction. This necessitates a more delicate analysis to prove a bound similar to Eq. (4). Due to space reasons, we defer our more detailed proof approach outline to Appendix D (see the discussion immediately following Lemma 1) The results in Appendix D culminates in the following result which limits the number of large-gradient.

**Proposition 1** (Bound on number of iterates with large gradients, informal version of Proposition 4)**.** *Let $\delta \in (0, 1/e]$ be arbitrary. Letting $\tilde{O}$ hide polylogarithmic dependencies on $\delta$ (and other parameters), consider choosing $u, r, \eta$ and $T$ such that*

$$u = \tilde{O}\left( \frac{\sqrt{\epsilon}}{\sqrt{\rho} d} \right), \qquad r = O(\epsilon), \qquad \eta = \tilde{O}\left( \frac{m}{dL} \right), \qquad T = \tilde{\Omega}\left( \frac{((f(x_0) - f^*) + \epsilon^2/L))}{\eta \epsilon^2} \right).$$

*Then, with probability at least $1 - O(\delta)$, there are at most $T/4$ iterations for which $\|\nabla f(x_t)\| \geq \epsilon$.*

## 3.2 Making progress near saddle points

**Challenge.** The noise in two-point zeroth-order estimators makes the analysis around $\epsilon-$approximate saddle points challenging, because the concentration properties of the (non-subGaussian) noise are hard to characterize. Intuitively, a noisier estimator might facilitate easier escape from saddle point. However, without an appropriate concentration bound, the noise may behave in unpredictable ways, preventing escape from saddle regions. Previous analysis of saddle point escape using stochastic estimators typically requires these estimators to satisfy subGaussian properties (Jin et al., 2019a; Fang et al., 2019), which zeroth-order estimators do not satisfy.

**High-level proof outline. (i)** We first prove a technical result showing that the travelling distance of the iterates can be bounded in terms of the function value decrease (i.e., Improve or Localize, Lemma 2). **(ii)** Next, at any $\epsilon$-saddle point, we consider a coupling argument and define two sequences running near-identical zeroth-order dynamics, differing only in the sign of their perturbative term along the minimum eigendirection of $H$, which denotes the Hessian of the saddle (Lemma 3). Using Lemma 2 in point (i), if we assume for contradiction that the two sequences both "get stuck" and make little function value progress, the dynamics of the difference between the two sequences will remain small as both sequences remain close to the saddle point. **iii)** However, since the perturbation vectors of the two sequences differ in the (most) negative direction of $H$, the norm of the the difference of the two sequences will grow exponentially so long as *a).* the sequences remain close to the saddle point (and thus the Hessian has a negative curvature direction) and *b).* the effect of the zeroth-order stochastic noise can be controlled. This leads to a contradiction, implying that sufficient function decrease must have been made (Proposition 5 in Appendix E.3). **(iv)** To show that the zeroth-order stochastic noise can be controlled, we prove one technical result (Proposition 2), providing a

concentration bound for the product of (possibly unbounded) subGaussian random vectors that scales linearly with the dimension $d$. This enables us to control the effect of the zeroth-order noise near saddle points, and is essential in showing that the eventual sample complexity scales linearly with $d$.

We provide a more detailed proof sketch below, where we elaborate more on our analytical challenges and ideas. We first introduce an informal statement of a key technical result that bounds, with high probability, the travelling distance of the iterates in terms of the function value decrease.

**Lemma 2** (Improve or Localize, informal version of Lemma 23). *Consider the perturbed zeroth-order update Algorithm 1. Let $\delta \in (0, 1/e]$ be arbitrary. Consider any $T_s = \tilde{\Omega}\left(\frac{1}{m}\log(1/\delta)\right)$, and any $t_0 \geq 0$. For any $F > 0$, suppose $f(x_{T_s+t_0}) - f(x_{t_0}) > -F$, i.e. $f(x_{t_0}) - f(x_{T_s+t_0}) < F$. Letting $\tilde{O}$ hide polylogarithmic terms involving $\delta$, suppose*

$$u = \tilde{O}\left(\frac{\min\{\sqrt{\epsilon}, \sqrt{r}\}}{\sqrt{\rho}d}\right), \qquad r = \tilde{O}\left(\min\left\{\epsilon, \frac{F}{\eta T_s}\right\}\right), \qquad \eta = \tilde{O}\left(\frac{m\sqrt{\rho\epsilon}}{dL}\right).$$

*Then, with probability at least $1 - O\left(\frac{T_s\delta}{T}\right)$ (here $T \geq T_s$ denotes the total number of iterations), for each $\tau \in \{0, 1, \ldots, T_s\}$, we have that*

$$\|x_{t_0+\tau} - x_{t_0}\|^2 \leq \phi_{T_s}(\delta, F), \quad \text{where } \phi_{T_s}(\delta, F) = \tilde{O}\left(\max\left\{T_s, \frac{d}{m}\right\}\right)\eta F + \tilde{O}(\eta^2\epsilon^2).$$

Intuitively, the above result shows that if little function value improvement has been made, then the algorithm's iterates have not moved much, such that it remains approximately in a saddle region if it started out in a saddle region. Next, Lemma 3 formally introduces the coupling we have mentioned, setting the stage for the rest of our arguments. For notational convenience, in this section, unless otherwise specified, we will assume that the initial iterate $x_0$ is an $\epsilon$-saddle point.

**Lemma 3.** *Suppose $x_0$ is an $\epsilon$-approximate saddle point. Without loss of generality, suppose that the minimum eigendirection of $H := \nabla^2 f(x_0)$ is the $e_1$ direction, and let $\gamma$ to denote $-\lambda_{\min}(\nabla^2 f(x_0))$ (note $\gamma \geq \sqrt{\rho\epsilon}$). Consider the following coupling mechanism, where we run the zeroth-order gradient dynamics, starting with $x_0$, with two isotropic noise sequences, $Y_t$ and $Y_t'$ respectively, where $(Y_t)_1 = -(Y_t)_1'$, and $(Y_t)_j = (Y_t)_j'$ for all other $j \neq 1$. Suppose that the sequence $\{Z_{t,i}\}_{t\in T, i\in[m]}$ is the same for both sequences. Let $\{x_t\}$ denote the sequence with the $\{Y_t\}$ noise sequence, and let the $\{x_t'\}$ denote the sequence with the $\{Y_t'\}$ noise sequence, where*

$$x_{t+1}' = x_t' - \eta\left(\frac{1}{m}\sum_{i=1}^m\left(Z_{t,i}Z_{t,i}^\top\nabla f(x_t') + \frac{u}{2}Z_{t,i}Z_{t,i}^\top\tilde{H}_{t,i}'Z_{t,i}\right) + Y_t'\right), \quad x_0' = x_0,$$

*and $\tilde{H}_{t,i}' := \frac{H_{t,i,+}' - H_{t,i,-}'}{2}$, with $H_{t,i,+}' = \nabla^2 f(x_t' + \alpha_{t,i,+}'uZ_i')$ for some $\alpha_{t,i,+}' \in [0,1]$, and $H_{t,i,-}' = \nabla^2 f(x_t - \alpha_{t,i,-}'uZ_i')$ for some $\alpha_{t,i,-}' \in [0,1]$. Then, for any $t \geq 0$,*

$$\hat{x}_{t+1} := x_{t+1} - x_{t+1}' = \underbrace{-\eta\sum_{\tau=0}^t(I - \eta H)^{t-\tau}\hat{\xi}_{g_0}(\tau)}_{W_{g_0}(t+1)} \underbrace{-\eta\sum_{\tau=0}^t(I - \eta H)^{t-\tau}(\bar{H}_\tau - H)\hat{x}_\tau}_{W_H(t+1)}$$

$$\underbrace{-\eta\sum_{\tau=0}^t(I - \eta H)^{t-\tau}\hat{\xi}_u(\tau)}_{W_u(t+1)} \underbrace{-\eta\sum_{\tau=0}^t(I - \eta H)^{t-\tau}\hat{Y}_\tau}_{W_p(t+1)}$$

*where*

$$\xi_{g_0}(t) = \frac{1}{m}\sum_{i=1}^m(Z_{t,i}Z_{t,i}^\top - I)\nabla f(x_t), \; \xi_{g_0}'(t) = \frac{1}{m}\sum_{i=1}^m(Z_{t,i}(Z_{t,i})^\top - I)\nabla f(x_t'), \; \hat{\xi}_{g_0}(t) = \xi_{g_0}(t) - \xi_{g_0}'(t),$$

$$\xi_u(t) = \frac{1}{m}\sum_{i=1}^m\frac{u}{2}Z_{t,i}Z_{t,i}\tilde{H}_{t,i}Z_{t,i}, \quad \xi_u'(t) = \frac{1}{m}\sum_{i=1}^m\frac{u}{2}Z_{t,i}Z_{t,i}\tilde{H}_{t,i}'Z_{t,i}, \quad \hat{\xi}_u(t) = \xi_u(t) - \xi_u'(t),$$

$$\hat{Y}_t = Y_t - Y_t', \quad \bar{H}_t = \int_0^1\nabla^2 f(ax_t + (1-a)x_t')da.$$

Our goal is to show that the dominating term in the evolution of the difference dynamics comes from the $W_p$ term involving the additional perturbation. To this end, we need to bound the remaining terms, $W_{g_0}, W_H, W_u$. A key technical challenge is to find a precise concentration bound for the $W_{g_0}(t+1)$ term, where

$$W_{g_0}(t+1) = -\eta \sum_{\tau=0}^{t} (I - \eta H)^{t-\tau} \left( \frac{1}{m} \sum_{i=1}^{m} (Z_{\tau,i} Z_{\tau,i}^\top - I)(\nabla f(x_\tau) - \nabla f(x_\tau')) \right).$$

For the simplicity of discussion, we assume for the time being that $m = 1$, and drop the $i$ index in the subscript of $Z_{\tau,i}$. Since $\mathbb{E}[Z_\tau Z_\tau^\top] = I$, heuristically, assuming that $Z_\tau Z_\tau^\top - I$ satisfies "nice" concentration properties, utilizing the independence of the $Z_\tau$'s across time and the fact that $(I - \eta H) \preceq (1 + \eta\gamma)I$, we would like to show that with high probability,

$$\|W_{g_0}(t)\| \lesssim \eta \sqrt{ \sum_{\tau=0}^{t-1} (1 + \eta\gamma)^{2(t-1-\tau)} \mathbb{E} \left[ \|(Z_\tau Z_\tau - I)(\nabla f(x_\tau) - \nabla f(x_\tau'))\|^2 \mid \mathcal{F}_{\tau-1} \right] } \quad (5)$$

where $\mathcal{F}_{\tau-1}$ is a sigma-algebra containing all randomness up to and including iteration $\tau - 1$, such that $x_\tau$ and $x_\tau'$ are both in $\mathcal{F}_{\tau-1}$, but $Z_\tau$ is not. Then, assuming that Eq. (5) holds, since

$$\mathbb{E} \left[ \|(Z_\tau Z_\tau - I)(\nabla f(x_\tau) - \nabla f(x_\tau'))\|^2 \mid \mathcal{F}_{\tau-1} \right] = O(d) \|\nabla f(x_\tau) - \nabla f(x_\tau')\|^2,$$

it follows that

$$\|W_{g_0}(t)\| \leq \eta \sqrt{ O(d) \sum_{\tau=0}^{t-1} (1 + \eta\gamma)^{2(t-1-\tau)} \|\nabla f(x_\tau) - \nabla f(x_\tau')\|^2 }$$

With this bound on $\|W_{g_0}(t)\|$, we eventually prove in Proposition 5 in Appendix E.3 that our algorithm escapes any $\epsilon-$saddle point with constant probability and that the $O(d)$ term appearing in the square root term above will eventually lead to an $O(d)$ dependence in the sample complexity[2]. We note that the $O(d)$ dimension dependence matches that of the best-known existing upper bound for finding first-order stationary points in smooth nonconvex zeroth-order optimization (Nesterov and Spokoiny, 2017), and has been conjectured to be the best possible dimension dependence for general smooth nonconvex zeroth-order optimization (Balasubramanian and Ghadimi, 2022).

**Key technical challenge** The key challenge in the above argument is to show that an equation in the form of Eq. (5) could in fact hold. At first glance, that an inequality such as Eq. (5) should hold is rather non-obvious — this is because while the variable $(Z_\tau Z_\tau - I)(\nabla f(x_\tau) - \nabla f(x_\tau')) \mid \mathcal{F}_{\tau-1}$ is mean-zero, it is subExponential rather than subGaussian. In fact, even in the subGaussian case, given a sequence of random vectors $\boldsymbol{x}_0, \ldots, \boldsymbol{x}_{t-1}$, such that each $\mathbb{E}[\boldsymbol{x}_\tau \mid \mathcal{F}_{\tau-1}] = 0$, and that each $\boldsymbol{x}_\tau \mid \mathcal{F}_{\tau-1}$ is norm-subGaussian with parameter $\sigma_\tau \in \mathcal{F}_{\tau-1}$ (which is an appropriate generalization of subGaussianity for vectors, proposed in Jin et al. (2019b)), proving a concentration inequality of the form $\left\| \sum_{\tau=0}^{t-1} \boldsymbol{x}_\tau \right\| \approx \tilde{O} \left( \sqrt{ \sum_{\tau=0}^{t-1} \sigma_\tau^2 } \right)$ is a very delicate matter. In our case, the analogue of $\boldsymbol{x}_\tau$ is $(I - \eta H)^{t-1-\tau}(Z_\tau Z_\tau - I)(\nabla f(x_\tau) - \nabla f(x_\tau'))$, while the analogue of $\sigma_\tau^2$ is $(1 + \eta\gamma)^{2(t-1-\tau)} \mathbb{E} \left[ \|(Z_\tau Z_\tau - I)(\nabla f(x_\tau) - \nabla f(x_\tau'))\|^2 \mid \mathcal{F}_{\tau-1} \right]$. Existing techniques (cf. Tropp et al. (2015); Jin et al. (2019b)) rely crucially on subGaussian properties that allow for each $\tau$ the moment-generating function $\mathbb{E}[e^{\theta \boldsymbol{Y}_\tau} \mid \mathcal{F}_{\tau-1}]$ to be defined for any fixed (and non-random) $\theta > 0$, where $\boldsymbol{Y}_\tau$ takes the form

$$\boldsymbol{Y}_\tau = \begin{bmatrix} 0 & \boldsymbol{x}_\tau^\top \\ \boldsymbol{x}_\tau & 0 \end{bmatrix},$$

such that $\mathbb{E}[\boldsymbol{Y}_\tau \mid \mathcal{F}_{\tau-1}] = 0$ (since $\mathbb{E}[\boldsymbol{x}_\tau \mid \mathcal{F}_{\tau-1}] = 0$), and the eigenvalues of $\boldsymbol{Y}_\tau$ are $\pm\|\boldsymbol{x}_\tau\|$. In the case when $\boldsymbol{x}_\tau$ is merely subExponential, the Moment Generating Function (MGF), $\mathbb{E}[e^{\theta \boldsymbol{Y}_\tau} \mid \mathcal{F}_{\tau-1}]$, will no longer be well-defined at any fixed (and non-random) $\theta > 0$. This poses a challenge in our setting, since $\boldsymbol{x}_\tau$ takes the form $(I - \eta H)^{t-1-\tau}(Z_\tau Z_\tau^\top - I)(\nabla f(x_\tau) - \nabla f(x_\tau'))$, which is subExponential rather than subGaussian. While it may be possible to force $(I - \eta H)^{t-1-\tau}(Z_\tau Z_\tau^\top - I)(\nabla f(x_\tau) - \nabla f(x_\tau'))$ to be sub-Gaussian, say by normalizing $Z_\tau$ to have norm $\sqrt{d}$ (note any

---

[2]For general $1 \leq m \leq d$, there will also be an $O(1/m)$ dependence in the sample complexity.

bounded random vector is also subGaussian), such that $\left\|(Z_\tau Z_\tau^\top - I)g\right\|^2 \le d^2\|g\|^2$ for any vector $g \in \mathbb{R}^d$, a careful examination of the argument in Proposition 5 would show that this results in a $O(d^2)$ rather than $O(d)$ dependence in the sample complexity, incurring a heavy price on the overall sample complexity (extra factor of $d$) if $d$ is large.

**Our solution**    To overcome the issue, we build on the following observation: with high probability, for any vector $g \in \mathbb{R}^d$, $\left|Z_\tau^\top g\right|$ is bounded within some log factor of $\|g\|$. On the event $\{|Z_\tau^\top g| = \tilde{O}(\|g\|)\}$, the variable

$$(Z_\tau Z_\tau^\top - I)g = Z_\tau(Z_\tau^\top g) - g \approx Z_\tau\|g\| - g$$

behaves approximately like a subGaussian random vector since $Z_\tau \sim N(0, I_d)$. Based on this intuition, after some careful analysis, we can show that $(Z_\tau Z_\tau^\top - I)(\nabla f(x_\tau) - \nabla f(x_\tau')) \mid \mathcal{F}_{\tau-1}$ is subGaussian on the event that $\left|Z_\tau^\top \nabla f(x_\tau)\right|$ is bounded within some log factor of $\|\nabla f(x_\tau)\|$, which happens with high probability. This then allows us to show that on this event, the corresponding MGF is well-defined for all fixed $\theta > 0$, enabling us to prove a concentration inequality of the form Eq. (5). This intuition is crystallized in the following proposition, which proves a more general bound than what we strictly need. For notational simplicity, we introduce the function $\mathrm{lr}(x) := \log\left(x\log(x)\right)$.

**Proposition 2.** *Let $\mathcal{F}_t$, $t \ge -1$ be a filtration. Let $(Z_t)_{t\ge 0}$ be a sequence of random vectors following the distribution $N(0, I)$ such that $Z_t \in \mathcal{F}_t$ and is independent of $\mathcal{F}_{t-1}$, and let $(v_t)_{t\ge 0}$ be a sequence of random vectors such that $v_t \in \mathcal{F}_{t-1}$. For each $\tau \ge 0$, let*

$$W_\tau = \sum_{t=0}^{\tau-1} M_t(Z_t Z_t^\top - I)v_t,$$

*where each $M_t$ is a deterministic matrix of appropriate dimension. Then, there exist some absolute constants $c', C > 0$ such that for any $\tau \in \mathbb{Z}^+$ and $\delta \in (0, 1/e]$, the following statements hold:*

*1. For any $\theta > 0$, with probability at least $1 - \delta$, we have*

$$\|W_\tau\| \le c'\theta \sum_{t=0}^{\tau-1} \|M_t\|_2^2 d(\mathrm{lr}(C\tau/\delta))^2\|v_t\|^2 + \frac{1}{\theta}\log(Cd\tau/\delta).$$

*2. For any $B > b > 0$, with probability at least $1 - \delta$,*

*either*    $\sum_{t=0}^{\tau-1} \|M_t\|_2^2 d(\mathrm{lr}(C\tau/\delta))^2\|v_t\|^2 \ge B$

*or*    $\|W_\tau\| \le c'\sqrt{\max\left\{\sum_{t=0}^{\tau-1}\|M_t\|_2^2 d(\mathrm{lr}(C\tau/\delta))^2\|v_t\|^2, b\right\}}\left(\log(C\tau d/\delta) + \log(\log(B/b) + 1)\right)$

*Moreover, as is clear from the bounds above, we may pick $C \ge 1$ such that $\log\left(\frac{C}{\delta}\right) \ge 1, \forall \delta \in (0, \frac{1}{e}]$.*

With this result, along with a series of other technical results in Appendix E.3, we can show that the algorithm makes a function decrease of $F$ with $\Omega(1)$ probability near an $\epsilon$-saddle point (Proposition 5in Appendix E.3). Armed with Proposition 5, as well as Proposition 1, the main result in Theorem 1 then follows. The complete detailed analysis can be found in Appendix E (Escaping saddle point) and Appendix F (main result).

## 4    CONCLUSION

In this paper, we proved that using two function evaluations per iteration suffices to escape saddle points and reach approximate second order stationary points efficiently in zeroth-order optimization. Along the way, we gave the first analysis of high-probability function change using two(or more)-point zeroth-order gradient estimators, as well as a novel concentration bound for sums of subExponential (but not subGaussian) vectors which are each the products of Gaussian vectors. These technical contributions may be of independent interest to researchers working in zeroth-order optimization as well as general stochastic optimization. There are a few limits of the current results which lead to several interesting future directions, such as extension to noisy function evaluations, as well as studying if some zeroth-order estimators such as asymmetric two-point estimators (Nesterov and Spokoiny, 2017) or single-point estimators (Flaxman et al., 2005) could actually escape saddle points without additional perturbation noise.

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

# A   RELATED WORK

**Two-point methods in zeroth-order optimization.**   Two-point (or in general $2m$-point, where $1 \leq m < d$ with $d$ being the problem dimension) estimators, which approximate the gradient using two (or $2m$) function evaluations per iteration, have been widely studied by researchers in the zeroth-order optimization literature, in convex (Nesterov and Spokoiny, 2017; Duchi et al., 2015; Shamir, 2017), nonconvex (Nesterov and Spokoiny, 2017), online (Shamir, 2017), as well as distributed settings (Tang et al., 2019). A key reason for doing so is that for applications of zeroth-order optimization arising in robotics (Li et al., 2022), wind farms (Tang et al., 2020a), power systems (Chen et al., 2020), online (time-varying) optimization (Shamir, 2017), learning-based control (Malik et al., 2019; Li et al., 2021), and improving adversarial robustness to black-box attacks in deep neural networks (Chen et al., 2017), it may be costly or impractical to wait for $\Omega(d)$ (where $d$ denotes the problem dimension) function evaluations per iteration to make a step. This is especially true for high-dimensional and/or time-varying problems. Indeed, for high-dimensional problems, two-point estimators can make swift progress even in the initial stage compared to $2d$-point estimator, and can reach a higher-quality solution if computation is limited (Tang et al., 2020b; Chen et al., 2017). For instance, consider the work in (Chen et al., 2017), which studies the use of zeroth-order estimators to perform black-box attacks on deep neural networks, in order to identify (and then defend against) adversarial images that may lead to misclassification. In the paper, the authors employed two-point zeroth-order estimators, due to the high computational cost of using $2d$ function evaluations per iteration for hundreds of iterations (here $d$ is the dimension of an image, which in this case is over 20000). The authors showed empirically that their two-point estimators worked well; however there over no accompanying theoretical results.

For online or time-varying environments, two-points estimators also often preferable. Since zeroth-order methods are often used in physical systems whose environment drifts or changes over time, this leads naturally to a *time-varying* or *online* optimization. For these problems, $2d$-point estimators will not produce a good estimation because the underlying function can drift to a very different problem while waiting for the $2d$ function evaluations. Indeed, the fewer function evaluations an optimization procedure needs, the faster it can catch up with the time-varying environment. In fact, for online optimization, it has been shown that two points estimator is optimal for convex Lipschitz functions (Shamir, 2017). Thus, two-point estimators are a natural fit for time-varying online optimization problems.

**Saddle point escape with access to deterministic gradient.** While standard gradient descent can escape saddle points asymptotically (Lee et al., 2019; Panageas et al., 2019), it is known that standard gradient descent may take exponential time to escape saddle points (Du et al., 2017). Hence, when access to deterministic gradient is available, research has centered on escaping saddle points with adding perturbation (Jin et al., 2017), momentum/acceleration based methods (Jin et al., 2018; Sun et al., 2019a; Staib et al., 2019), or gradient-based robust Hessian power/curvature exploitation methods (Zhang and Li, 2021; Adolphs et al., 2019). In addition, there has also been work on escaping saddle points devoted to specific optimization settings, such as constrained optimization (Mokhtari et al., 2018; Avdiukhin et al., 2019), optimization of weakly convex functions (Huang, 2021), bilevel optimization (Huang et al., 2022), as well as on general manifolds (Sun et al., 2019b; Criscitiello and Boumal, 2019; Han and Gao, 2020).

**Saddle point escape in stochastic gradient descent (SGD).** In practice, only stochastic gradient estimators are available in many problems. While SGD may converge to local maxima in worst-case scenarios (Ziyin et al., 2021), under assumptions such as bounded variance or subGaussian noise, there have been many works that have studied the problem of saddle point escape in SGD (Ge et al., 2015; Daneshmand et al., 2018; Xu et al., 2018; Jin et al., 2019a; Vlaski and Sayed, 2021b). The best existing rate (without considering momentum/variance reduction techniques) appears to belong to that of Fang et al. (2019), which converges to $\epsilon$-second order stationary points using $\tilde{O}(1/\epsilon^{3.5})$ stochastic gradients. While zeroth-order gradient estimators may also be viewed as stochastic gradients, they typically do not satisfy the bounded/subGaussian noise assumptions that are assumed in these works, making a direct comparison inappropriate. Escaping saddle point via momentum methods in SGD has also been studied (Wang et al., 2021; Antonakopoulos et al., 2022); while we do not consider incorporating momentum in our works, this may be interesting future work. A number of papers has also considered the specialized setting of escaping saddle points in nonconvex finite-sum optimization (Reddi et al., 2018; Liang et al., 2021), with many considering the case where variance-reduction is

used (Ge et al., 2019; Li, 2019). While the finite-sum problem is quite different from our problem, the variance reduction approach considered in these works may be a relevant future direction. The saddle point escape problem has also been studied in other specific settings such as compressed optimization (Avdiukhin and Yaroslavtsev, 2021), distributed optimization (Vlaski and Sayed, 2021a), or in the overparameterization case (Roy et al., 2020).

**Saddle point escape with zeroth-order information.** The problem of escaping saddle points in zeroth-order optimization has been studied less often, and we have already listed all known works comparable to our work in the introduction (Bai et al., 2020; Vlatakis-Gkaragkounis et al., 2019; Balasubramanian and Ghadimi, 2022); a more detailed comparison of these works with our results has been provided in the discussion following the statement of our main result Theorem 1. We would like to mention that Roy et al. (2020) also includes a convergence result of $\tilde{O}\left(\frac{d^{1.5}}{\epsilon^{4.5}}\right)$ for the case with noisy function evaluations, which is incomparable to our existing work which focuses on the case with exact function evaluation. In addition, Roy et al. (2020) also makes a subGaussian assumption on the estimator noise, which zeroth-order estimators in our paper do not satisfy. Nonetheless, considering the extension to noisy function evaluations will make for important future work.

**Zeroth-order optimization.** Our work rests on a line of research in zeroth-order optimization which focuses on constructing gradient estimators using zeroth-order function values (Flaxman et al., 2005; Duchi et al., 2015; Nesterov and Spokoiny, 2017; Shamir, 2017; Larson et al., 2019). As we have discussed, for smooth nonconvex functions, it is known that two-point zeroth-order estimators suffice to find first-order $\epsilon$-stationary points using $\tilde{O}(d/\epsilon^2)$ function evaluations (Nesterov and Spokoiny, 2017). Our work studies the more complicated problem of reaching $\epsilon$-second order stationary points, attaining a rate of $\tilde{O}(d/\epsilon^{2.5})$.

## B    PROOF ROADMAP

We begin by introducing several key concentration inequalities in Appendix C which we will frequently use in our proofs. We then describe in detail (and prove) the sequence of results that lead up to Proposition 4 in Appendix D, showing that there cannot be too many iterations with large gradients. Next, we describe the saddle point argument in detail, and prove Proposition 5 in Appendix E.3. Finally, we combine these results and prove our main result Theorem 2 (whose informal version is Theorem 1) in Appendix F

Throughout our proofs, absolute constants, as denoted by e.g. $(c, c', C)$, may change from line to line. However, within the same proof, for clarity, we try to index different constants differently. We assume $d \geq 2$ and $m \leq d$.

**Notations.**    We shall denote the conditional expectation and conditional probability by $\mathbb{E}_{\mathcal{F}}[\cdot] = \mathbb{E}[\cdot \mid \mathcal{F}]$ and $\mathbb{P}_{\mathcal{F}}(\cdot) = \mathbb{P}(\cdot \mid \mathcal{F})$ where $\mathcal{F}$ is a sigma-algebra.

## C    CONCENTRATION INEQUALITIES

This section serves to introduce several probabilistic results which will be useful for our main proofs in subsequent sections. We first introduce subGaussian, subExponential and norm-subGaussian random vectors in Appendix C.1. Next, in Appendix C.2, we provide concentration bounds for norm-subGaussian and subExponential random vectors. We then prove a novel concentration inequality involving products of subGaussian random vectors in Appendix C.3. We conclude by stating some concentration bounds for Appendix C.4 random variables.

### C.1    SUBGAUSSIAN, SUBEXPONENTIAL AND NORM-SUBGAUSSIAN RANDOM VECTORS

We first define subGaussian and subExponential random vectors. A detailed reference for these concepts can be found in Vershynin (2018).

**Definition 5** (subGaussian and subExponential random vectors). A random vector $\boldsymbol{x} \in \mathbb{R}^d$ is $\sigma$-subGaussian (SG($\sigma$)), if there exists $\sigma > 0$ such that for any unit vector $g \in \mathbb{S}^{d-1}$,

$$\mathbb{E}\left[\exp(\lambda \langle g, \boldsymbol{x} - \mathbb{E}[\boldsymbol{x}] \rangle)\right] \leq \exp(\lambda^2 \sigma^2 / 2) \quad \forall \lambda \in \mathbb{R}.$$

Meanwhile, a random vector $\boldsymbol{x} \in \mathbb{R}^d$ is $\sigma$-subExponential (SE($\sigma$)), if there exists $\sigma > 0$ such that for any unit vector $g \in \mathbb{S}^{d-1}$,

$$\mathbb{E}\left[\exp(\lambda\langle g, \boldsymbol{x} - \mathbb{E}[\boldsymbol{x}]\rangle)\right] \leq \exp(\lambda^2\sigma^2/2) \quad \forall|\lambda| \leq \frac{1}{\sigma}$$

An alternative concentration property for random vectors revolving around its norm, known as norm-subGaussianity (Jin et al., 2019b), is also relevant.

**Definition 6** (norm-subGaussian random vectors). A random vector $\boldsymbol{x} \in \mathbb{R}^d$ is $\sigma$-norm-subGaussian (nSG($\sigma$)), there exists $\sigma > 0$ such that

$$\mathbb{P}(\|\boldsymbol{x} - \mathbb{E}\boldsymbol{x}\| \geq s) \leq 2e^{-\frac{s^2}{2\sigma^2}} \quad \forall s \geq 0.$$

We recall the following result which provides several examples of nSG random vectors. In particular, it tells us a random vector $\boldsymbol{x} \in \mathbb{R}^d$ that is $(\sigma/\sqrt{d})-$subGaussian is also $\sigma$-subGaussian.

**Lemma 4** (Lemma 1 in Jin et al. (2019b)). *There exists absolute constant $c$ such that the following random vectors are all nSG(c$\sigma$).*

1. *A bounded random vector $\boldsymbol{x} \in \mathbb{R}^d$ so that $\|\boldsymbol{x}\| \leq \sigma$.*

2. *A random vector $\boldsymbol{x} \in \mathbb{R}^d$, where $\boldsymbol{x} = \xi\boldsymbol{e}_1$ and the random variable $\xi \in \mathbb{R}$ is $\sigma$-subGaussian.*

3. *A random vector $\boldsymbol{x} \in \mathbb{R}^d$ that is $(\sigma/\sqrt{d})-$subGaussian*

In addition, if $\boldsymbol{x} \in \mathbb{R}^d$ is zero-mean nSG($\sigma$), its component along a single direction is also subGaussian.

**Lemma 5.** *Suppose $\boldsymbol{x} \in \mathbb{R}^d$ is zero-mean nSG($\sigma$). Then, for any fixed vector $\boldsymbol{v} \in \mathbb{R}^d$, $\langle \boldsymbol{v}, \boldsymbol{x} \rangle$ is zero-mean $\|\boldsymbol{v}\|\sigma$-subGaussian.*

*Proof.* Without loss of generality, we assume that $\boldsymbol{v} \in \mathbb{S}^{d-1}$ is a unit vector. That $\langle \boldsymbol{v}, \boldsymbol{x} \rangle$ is zero-mean follows directly from $\boldsymbol{x}$ being zero-mean and $\boldsymbol{v}$ being fixed. Meanwhile, since $|\langle \boldsymbol{v}, \boldsymbol{x} \rangle| \leq \|\boldsymbol{v}\|\|\boldsymbol{x}\| = \|\boldsymbol{x}\|$, for any $s \geq 0$, it follows that

$$\mathbb{P}(|\langle \boldsymbol{v}, \boldsymbol{x} \rangle| \geq s) \leq \mathbb{P}(\|\boldsymbol{x}\| \geq s) \leq 2e^{-\frac{s^2}{2\sigma^2}},$$

where the last inequality follows from the fact that $\boldsymbol{x}$ is zero-mean and also nSG($\sigma$). Hence $\langle \boldsymbol{v}, \boldsymbol{x} \rangle$ is zero-mean SG($\sigma$), as desired. $\qquad\square$

### C.2 CONCENTRATION BOUNDS FOR NORM-SUBGAUSSIAN AND SUBEXPONENTIAL RANDOM VECTORS

We begin by giving some concentration bounds for norm-subGaussian random vectors. To do so, we introduce the following condition.

**Condition 1.** *Consider random vectors $\boldsymbol{x}_1, \ldots, \boldsymbol{x}_n \in \mathbb{R}^d$, and corresponding filtrations $\mathcal{F}_i$ generated by $(\boldsymbol{x}_1, \ldots, \boldsymbol{x}_i)$. We assume $\boldsymbol{x}_i \mid \mathcal{F}_{i-1}$ is zero-mean, nSG($\sigma_i$), with $\sigma_i \in \mathcal{F}_{i-1}$, i.e,*

$$\mathbb{E}\left[\boldsymbol{x}_i \mid \mathcal{F}_{i-1}\right] = 0,$$

*and*

$$\mathbb{P}\left(\|\boldsymbol{x}_i\| \geq s \mid \mathcal{F}_{i-1}\right) \leq 2e^{-\frac{s^2}{2\sigma_i^2}} \quad \forall s \geq 0,$$

*where $\sigma_i$ is a measurable function of $(\boldsymbol{x}_1, \ldots, \boldsymbol{x}_{i-1})$ for each $i$.*

For norm subGaussian random vectors satisfying Condition 1, we first have the following bound.

**Lemma 6.** *Suppose $(\boldsymbol{x}_1, \ldots, \boldsymbol{x}_n) \in \mathbb{R}^d$ satisfy Condition 1, i.e. each $\boldsymbol{x}_i \mid \mathcal{F}_{i-1}$ is mean-zero, nSG($\sigma_i$) with $\sigma_i \in \mathcal{F}_{i-1}$. Let $\{\boldsymbol{u}_i\}$ denote a sequence of random vectors such that $\boldsymbol{u}_i \in \mathcal{F}_{i-1}$ for every $i \in [n]$. Then, there exists an absolute constant $c$, such that for any $\delta \in (0, 1)$ and $\lambda > 0$, with probability at least $1 - \delta$,*

$$\sum_{i=1}^n \langle \boldsymbol{u}_i, \boldsymbol{x}_i \rangle \leq c\lambda \sum_{i=1}^n \|\boldsymbol{u}_i\|^2 \sigma_i^2 + \frac{1}{\lambda}\log(1/\delta).$$

*Proof.* We note that if $\boldsymbol{x}_i$ is mean-zero and nSG($\sigma_i$), then by Lemma 5, $\langle \boldsymbol{u}_i, \boldsymbol{x}_i \rangle \mid \mathcal{F}_{i-1}$ is zero-mean and $\|\boldsymbol{u}_i\|\sigma_i$-subGaussian. The rest of the proof follows from the proof of Lemma 39 in Jin et al. (2019a) (key idea is exponentiate and then apply Markov's inequality). For completeness, we restate the proof here. Observe that for any $i$, since $\langle \boldsymbol{u}_i, \boldsymbol{x}_i \rangle$ is $\|\boldsymbol{u}_i\|\sigma_i$-subGaussian, for any $\lambda > 0$, we have that

$$\mathbb{E}\left[\exp(\lambda\langle \boldsymbol{u}_i, \boldsymbol{x}_i \rangle) \mid \mathcal{F}_{i-1}\right] \leq \exp(\lambda^2 \|\boldsymbol{u}_i\|^2 \sigma_i^2 / 2)$$

For any $\lambda > 0$ and $s \geq 0$, observe that

$$\mathbb{P}\left( \sum_{i=1}^{n} \lambda\langle \boldsymbol{u}_i, \boldsymbol{x}_i \rangle - \lambda^2 \|\boldsymbol{u}_i\|^2 \sigma_i^2 / 2 \geq s \right)$$

$$= \mathbb{P}\left( \exp\left( \lambda \sum_{i=1}^{n} \langle \boldsymbol{u}_i, \boldsymbol{x}_i \rangle - \lambda^2 \|\boldsymbol{u}_i\|^2 \sigma_i^2 / 2 \right) \geq \exp(\lambda s) \right)$$

$$\leq \mathbb{E}\left[ \exp\left( \lambda \sum_{i=1}^{n} \langle \boldsymbol{u}_i, \boldsymbol{x}_i \rangle - \lambda^2 \|\boldsymbol{u}_i\|^2 \sigma_i^2 / 2 \right) \right] \exp(-\lambda s)$$

$$= \mathbb{E}\left[ \mathbb{E}\left[ \exp\left( \lambda \sum_{i=1}^{n} \langle \boldsymbol{u}_i, \boldsymbol{x}_i \rangle - \lambda^2 \|\boldsymbol{u}_i\|^2 \sigma_i^2 / 2 \right) \Big| \mathcal{F}_{n-1} \right] \right] \exp(-\lambda s)$$

$$= \mathbb{E}\left[ \exp\left( \lambda \sum_{i=1}^{n-1} \langle \boldsymbol{u}_i, \boldsymbol{x}_i \rangle - \lambda^2 \|\boldsymbol{u}_i\|^2 \sigma_i^2 / 2 \right) \mathbb{E}\left[ \exp\left( \lambda\langle \boldsymbol{u}_n, \boldsymbol{x}_n \rangle - \lambda^2 \|\boldsymbol{u}_n\|^2 \sigma_n^2 / 2 \right) \Big| \mathcal{F}_{n-1} \right] \right] \exp(-\lambda s)$$

$$\overset{(i)}{\leq} \mathbb{E}\left[ \exp\left( \lambda \sum_{i=1}^{n-1} \langle \boldsymbol{u}_i, \boldsymbol{x}_i \rangle - \lambda^2 \|\boldsymbol{u}_i\|^2 \sigma_i^2 / 2 \right) \right] \exp(-\lambda s) \leq \cdots \leq \exp(-\lambda s)$$

Above, (i) follows from the fact that $\langle \boldsymbol{u}_i, \boldsymbol{x}_i \rangle \mid \mathcal{F}_{i-1}$ is zero-mean and $\|\boldsymbol{u}_i\|\sigma_i$-subGaussian for each $i \in [n]$. The final result then follows by picking $c = \frac{1}{2}$ and $s = \log(1/\delta)$. $\qquad\square$

Assuming Condition 1, the following concentration result also holds for a sequence of nSG random vectors.

**Lemma 7** (Lemma 6, Corollary 7 and Corollary 8 in Jin et al. (2019b) combined)**.** *Suppose* $(\boldsymbol{x}_1, \ldots, \boldsymbol{x}_n) \in \mathbb{R}^d$ *satisfy Condition 1. Then, there exists an absolute constant $c$ such that for any fixed $\delta \in (0, 1)$, $\theta > 0$, with probability at least $1 - \delta$,*

$$\left\| \sum_{i=1}^{n} \boldsymbol{x}_i \right\| \leq c\theta \sum_{i=1}^{n} \sigma_i^2 + \frac{1}{\theta} \log(2d/\delta).$$

*Moreover, there are two corollaries.*

1. *((Jin et al., 2019b, Corollary 7)) When $\{\sigma_i\}$ is deterministic, there exists an absolute constant $c$ such that for any fixed $\delta \in (0, 1)$, with probability at least $1 - \delta$.*

$$\left\| \sum_{i=1}^{n} \boldsymbol{x}_i \right\| \leq c\sqrt{\log(2d/\delta) \sum_{i=1}^{n} \sigma_i^2}$$

2. *((Jin et al., 2019b, Corollary 8)) Suppose that the $\{\sigma_i\}$ sequence is random. Then, there exists an absolute constant $c$ such that for any fixed $\delta \in (0, 1)$ and $B > b > 0$, with probability at least $1 - \delta$:*

$$\textit{either } \sum_{i=1}^{n} \sigma_i^2 \geq B \quad \textit{or } \left\| \sum_{i=1}^{n} \boldsymbol{x}_i \right\| \leq c\sqrt{\max\left\{ \sum_{i=1}^{n} \sigma_i^2, b \right\} \cdot (\log(2d/\delta) + \log(\log(B/b)))}$$

We state here a Bernstein-type concentration inequality for sub-exponential random variables, which we also need.

**Lemma 8** (Bernstein concentration inequality). *Consider a sequence of independently distributed $\sigma$-subexponential variables $\boldsymbol{x}_1, \ldots, \boldsymbol{x}_n \in \mathbb{R}$, with mean $\mathbb{E}[\boldsymbol{x}_i] \leq c'\sigma$ for some $c' > 0$ and each $i \in [n]$. Then, there exists an absolute constant $C > 0$, such that for any $\delta \in (0, 1)$, with probability at least $1 - \delta$,*

$$\sum_{i=1}^{n} \boldsymbol{x}_i \leq C\sigma(n + \log(1/\delta)). \tag{6}$$

*Proof.* The result of Eq. (6) follows by applying Bernstein's inequality to $\sum_{i=1}^{n} \boldsymbol{x}_i - \mathbb{E}[\boldsymbol{x}_i]$ (so each summand is mean-zero). Per Bernstein's inequality, (cf. Theorem 2.8.1 in Vershynin (2018)), there exists an absolute constant $c > 0$ such that for any $s \geq 0$,

$$\mathbb{P}\left(\sum_{i=1}^{n}(\boldsymbol{x}_i - \mathbb{E}[\boldsymbol{x}_i]) \geq s\right) \leq \exp\left(-c\min\left\{\frac{s^2}{n\sigma^2}, \frac{s}{\sigma}\right\}\right).$$

Pick $s = \sigma\left(n + \frac{\log(1/\delta)}{c}\right)$. Then,

$$\min\left\{\frac{s^2}{n\sigma^2}, \frac{s}{\sigma}\right\} = \min\left\{n + 2\frac{\log(1/\delta)}{c} + \frac{(\log(1/\delta))^2}{c^2 n}, n + \frac{\log(1/\delta)}{c}\right\} = n + \frac{\log(1/\delta)}{c}.$$

Continuing, we have that

$$\mathbb{P}\left(\sum_{i=1}^{n}(\boldsymbol{x}_i - \mathbb{E}[\boldsymbol{x}_i]) \geq s\right) \leq \exp\left(-c\min\left\{\frac{s^2}{n\sigma^2}, \frac{s}{\sigma}\right\}\right) \leq \exp\left(-c\left(n + \frac{\log(1/\delta)}{c}\right)\right) \leq \delta.$$

Thus, it follows that with probability at least $1 - \delta$,

$$\sum_{i=1}^{n}(\boldsymbol{x}_i - \mathbb{E}[\boldsymbol{x}_i]) \leq \sigma\left(n + \frac{\log(1/\delta)}{c}\right) \implies \sum_{i=1}^{n} \boldsymbol{x}_i \leq \sigma\left(n + \frac{\log(1/\delta)}{c}\right) + nc'\sigma,$$

where implication holds since by assumption, $\mathbb{E}[\boldsymbol{x}_i] \leq c'\sigma$ for some $c' > 0$. Then, by setting $C = \max\{1 + c', 1/c\}$, the desired result follows. $\square$

### C.3 A NOVEL CONCENTRATION INEQUALITY FOR THE ZEROTH-ORDER SETTING

In the zeroth-order setting, we will frequently have to bound the norms of terms of the form

$$W_\tau = \sum_{t=0}^{\tau-1} M_t(Z_t Z_t^\top - I)v_t, \tag{7}$$

where $M_t$ is a known and fixed quantity, while $Z_t$ is random, and $v_t$ depends on $x_0$ and the history of previous $\{Z_j\}_{j=0}^{t-1}$'s, and is hence $\mathcal{F}_{t-1}$-measurable. For our purposes, it suffices to consider $Z_t \sim N(0, I)$.

To see why such a bound will be useful, as mentioned in the main text and as we will see again later in the full proofs, in the analysis of escaping saddle points, we need to bound a term of the form

$$W_{g_0}(\tau) = \eta \sum_{t=0}^{\tau-1}(I - \eta H)^{\tau-1-t}(Z_t Z_t^\top - I)(\nabla f(x_t) - \nabla f(x_t')),$$

where $H = \nabla^2 f(x_0)$ (assuming that $x_0$ is an $\epsilon$-saddle point), and $x_t$ and $x_t'$ are two coupled sequences. Comparing with Eq. (7), we see that for the equation above, we can define $M_t = \eta(I - \eta H)^{\tau-1-t}$ (a fixed and known quantity) and $v_t = \nabla f(x_t) - \nabla f(x_t')$ (clearly, $\nabla f(x_t) - \nabla f(x_t')$ is $\mathcal{F}_{t-1}$-measurable). This motivates why we wish to bound terms of the form Eq. (7).

Observe that each $(Z_t Z_t^\top - I)v_t \mid \mathcal{F}_{t-1}$ term is subExponential rather than subGaussian. While it is possible to define norm-subExponential vectors in analogous way to norm-subGaussian vectors, the corresponding moment generating function (MGF) for subExponential random variables is not defined on the entirety of $\mathbb{R}$. When bounding a sum in the form of $\sum_{t=0}^{\tau-1}(Z_t Z_t^\top - I)v_t$, this creates a subtle but challenging technical issue.

Following the intuition outlined in the main text, we bypass this difficulty by proving the following result. For notational simplicity, we introduce the function

$$\mathrm{lr}(x) := \log\left(x\log(x)\right). \tag{8}$$

We now recall Proposition 2 which we first introduced in the main text.

**Proposition 2.** *Let $\mathcal{F}_t$, $t \geq -1$ be a filtration. Let $(Z_t)_{t\geq 0}$ be a sequence of random vectors following the distribution $N(0, I)$ such that $Z_t \in \mathcal{F}_t$ and is independent of $\mathcal{F}_{t-1}$, and let $(v_t)_{t\geq 0}$ be a sequence of random vectors such that $v_t \in \mathcal{F}_{t-1}$. For each $\tau \geq 0$, let*

$$W_\tau = \sum_{t=0}^{\tau-1} M_t(Z_t Z_t^\top - I)v_t,$$

*where each $M_t$ is a deterministic matrix of appropriate dimension. Then, there exist some absolute constants $c', C > 0$ such that for any $\tau \in \mathbb{Z}^+$ and $\delta \in (0, 1/e]$, the following statements hold:*

1. *For any $\theta > 0$, with probability at least $1 - \delta$, we have*

$$\|W_\tau\| \leq c'\theta \sum_{t=0}^{\tau-1} \|M_t\|_2^2 d(\mathrm{lr}(C\tau/\delta))^2 \|v_t\|^2 + \frac{1}{\theta}\log(Cd\tau/\delta).$$

2. *For any $B > b > 0$, with probability at least $1 - \delta$,*

$$\text{either} \quad \sum_{t=0}^{\tau-1} \|M_t\|_2^2 d(\mathrm{lr}(C\tau/\delta))^2 \|v_t\|^2 \geq B$$

$$\text{or} \quad \|W_\tau\| \leq c'\sqrt{\max\left\{\sum_{t=0}^{\tau-1} \|M_t\|_2^2 d(\mathrm{lr}(C\tau/\delta))^2 \|v_t\|^2, b\right\} (\log(C\tau d/\delta) + \log(\log(B/b) + 1))}$$

*Moreover, as is clear from the bounds above, we may pick $C \geq 1$ such that $\log\left(\frac{C}{\delta}\right) \geq 1, \forall \delta \in (0, \frac{1}{e}]$.*

*Proof.* We will focus on proving the first point, since the second follows as a natural corollary of our proof of the first part and the proof of (Jin et al., 2019b, Corollary 8). For simplicity, we shall assume $v_t \neq 0$ in the intermediate steps; extension to the general case is straightforward.

First of all, for $0 \leq \alpha < 1$, let

$$g(\alpha; \delta) = \sqrt{\frac{2}{\pi}} \int_\alpha^{\sqrt{2\,\mathrm{lr}(1/\delta)}} (x^2 - 1)e^{-x^2/2}\,dx = \sqrt{\frac{2}{\pi}}\left(\alpha e^{-\alpha^2/2} - \frac{\delta\sqrt{2\,\mathrm{lr}(1/\delta)}}{\log(1/\delta)}\right).$$

It's not hard to see that for a fixed $\delta \in (0, 1/e]$, $g(\alpha; \delta)$ is continuous and strictly increasing over $\alpha \in [0, 1)$. Then, since $\frac{\log x}{x} + 1 \leq x$ for $x \geq 1$, by plugging in $x = \log(1/\delta)$, we get

$$\frac{\mathrm{lr}(1/\delta)}{(\log(1/\delta))^2} = \frac{\log\log(1/\delta) + \log(1/\delta)}{(\log(1/\delta))^2} = \frac{1}{\log(1/\delta)}\left(\frac{\log\log(1/\delta)}{\log(1/\delta)} + 1\right) \leq 1,$$

which leads to

$$g(2\delta; \delta) = \sqrt{\frac{2}{\pi}}\left(2\delta e^{-2\delta^2} - \frac{\delta\sqrt{2\,\mathrm{lr}(1/\delta)}}{\log(1/\delta)}\right) \geq \sqrt{\frac{2}{\pi}}\left(2e^{-2/e^2}\delta - \sqrt{2}\delta\right) > 0$$

for $\delta \in (0, 1/e]$. Furthermore, we obviously have $g(0; \delta) < 0$. Therefore $g(\alpha; \delta) = 0$ has a unique solution in $(0, 2\delta)$, which we denote by $\alpha(\delta)$.[3] These results imply that, for a random variable $Z$ following the standard normal distribution, we have

$$\mathbb{E}\left[(Z^2 - 1)\mathbb{1}_{\alpha(\delta) \leq |Z| \leq \sqrt{2\,\mathrm{lr}(1/\delta)}}\right] = \sqrt{\frac{2}{\pi}}\int_{\alpha(\delta)}^{\sqrt{2\,\mathrm{lr}(1/\delta)}}(x^2 - 1)e^{-x^2/2}\,dx = g(h(\delta); \delta) = 0$$

---

[3]By letting $W_0(x)$ denote the the principal branch of the Lambert $W$ function, it can be shown that

$$\alpha(\delta) = \sqrt{-W_0\left(-\frac{2\delta^2\,\mathrm{lr}(1/\delta)}{(\log(1/\delta))^2}\right)}.$$

and

$$\mathbb{P}(\alpha(\delta) \leq |Z| \leq \sqrt{2\operatorname{lr}(1/\delta)}) \geq 1 - 2\left(\frac{1}{\sqrt{2\pi}}\int_{\sqrt{2\operatorname{lr}(1/\delta)}}^{\infty} e^{-x^2/2}\,dx + \frac{1}{\sqrt{2\pi}}\int_0^{\alpha(\delta)} e^{-x^2/2}\,dx\right)$$

$$\geq 1 - 2\left(\frac{1}{2}\exp\left(-\frac{2\operatorname{lr}(1/\delta)}{2}\right) + \frac{\alpha(\delta)}{\sqrt{2\pi}}\right) = 1 - 2\left(\frac{\delta}{2\log(1/\delta)} + \frac{\alpha(\delta)}{\sqrt{2\pi}}\right)$$

$$\geq 1 - 2\left(\frac{\delta}{2} + \frac{2}{\sqrt{2\pi}}\delta\right) \geq 1 - C\delta$$

for any $\delta \in (0, 1/e]$, where we define the absolute constant $C := 2(1/2 + 2/\sqrt{2\pi})$.

Now we let $A_t$ denote the event

$$A_t = \left\{\alpha(\delta) \leq \frac{|Z_t^\top v_t|}{\|v_t\|} \leq \sqrt{2\operatorname{lr}(1/\delta)}\right\}.$$

Since $Z_t^\top v_t/\|v_t\|$ conditioned on $\mathcal{F}_{t-1}$ follows the standard normal distribution, we have

$$\mathbb{P}_{\mathcal{F}_{t-1}}(A_t) \geq 1 - C\delta, \tag{9}$$

and

$$\mathbb{E}_{\mathcal{F}_{t-1}}\left[v_t^\top \left(Z_t Z_t^\top - I\right) v_t \mathbb{1}_{A_t}\right] = 0.$$

Moreover, for any random vector $u \in \mathcal{F}_{t-1}$ that is orthogonal to $v_t$, we have

$$\mathbb{E}_{\mathcal{F}_{t-1}}\left[u^\top \left(Z_t Z_t^\top - I\right) v_t \mathbb{1}_{A_t}\right] = \mathbb{E}_{\mathcal{F}_{t-1}}\left[u^\top Z_t\right] \cdot \mathbb{E}_{\mathcal{F}_{t-1}}\left[Z_t^\top v_t \mathbb{1}_{A_t}\right] = 0,$$

where we used the fact that $Z_t^\top u$ is independent of $Z_t^\top v_t$ conditioned on $\mathcal{F}_{t-1}$. Therefore

$$\mathbb{E}_{\mathcal{F}_{t-1}}\left[(Z_t Z_t^\top - I)v_t \mathbb{1}_{A_t}\right] = 0.$$

Consider defining then the random variable $Q_t$ by

$$Q_t := (Z_t Z_t^\top - I)v_t \cdot \mathbb{1}_{A_t}.$$

We now show that $Q_t \mid \mathcal{F}_{t-1}$ is norm-subGaussian. Let $u \in \mathbb{R}^d$ with $\|u\| = 1$ be arbitrary. We have

$$u^\top Q_t = u^\top (Z_t Z_t^\top - I)v_t \cdot \mathbb{1}_{A_t}$$

$$= u^\top \left(\frac{v_t v_t^\top}{\|v_t\|^2} + I - \frac{v_t v_t^\top}{\|v_t\|^2}\right)(Z_t Z_t^\top - I)v_t \cdot \mathbb{1}_{A_t}$$

$$= u^\top v_t \left(\frac{|Z_t^\top v_t|^2}{\|v_t\|^2} - 1\right) \cdot \mathbb{1}_{A_t} + u^\top \left(I - \frac{v_t v_t^\top}{\|v_t\|^2}\right)(Z_t Z_t^\top - I)v_t \cdot \mathbb{1}_{A_t}$$

$$= u^\top v_t \left(\frac{|Z_t^\top v_t|^2}{\|v_t\|^2} - 1\right) \cdot \mathbb{1}_{A_t} + u_\perp^\top Z_t Z_t^\top v_t \cdot \mathbb{1}_{A_t},$$

where we denote $u_\perp = \left(I - \frac{v_t v_t^\top}{\|v_t\|^2}\right)u$. Since

$$\left|u^\top v_t \left(\frac{|Z_t^\top v_t|^2}{\|v_t\|^2} - 1\right) \cdot \mathbb{1}_{A_t}\right| \leq |u^\top v_t|(2\operatorname{lr}(1/\delta) - 1),$$

we see that $u^\top v_t \left(\frac{|Z_t^\top v_t|^2}{\|v_t\|^2} - 1\right) \cdot \mathbb{1}_{A_t}$ conditioned on $\mathcal{F}_{t-1}$ is $|u^\top v_t|(2\operatorname{lr}(1/\delta) - 1)$-subGaussian. Furthermore, since $|u_\perp^\top Z_t Z_t^\top v_t \cdot \mathbb{1}_{A_t}| \leq |Z_t^\top u_\perp|\sqrt{2\operatorname{lr}(1/\delta)}\|v_t\|$, we have

$$\mathbb{P}_{\mathcal{F}_{t-1}}\left(|u_\perp^\top Z_t Z_t^\top v_t \cdot \mathbb{1}_{A_t}| \geq s\right) \leq \mathbb{P}_{\mathcal{F}_{t-1}}\left(|Z_t^\top u_\perp|\sqrt{2\operatorname{lr}(1/\delta)}\|v_t\| \geq s\right),$$

and since $Z_t u_\perp/\|u_\perp\| \mid \mathcal{F}_{t-1}$ follows the standard normal distribution, we see that $u_\perp^\top Z_t Z_t^\top v_t \cdot \mathbb{1}_{A_t}$ is a $\sqrt{2\operatorname{lr}(1/\delta)}\|u_\perp\|\|v_t\|$-subGaussian variable. Note that $u^\top Q_t$ is just the sum of $u^\top v_t \left(\frac{|Z_t^\top v_t|^2}{\|v_t\|^2} - 1\right) \cdot \mathbb{1}_{A_t}$ and $u_\perp^\top Z_t Z_t^\top v_t \cdot \mathbb{1}_{A_t}$, we can conclude that $u^\top Q_t$ is subGaussian with parameter

$$(2\operatorname{lr}(1/\delta) - 1)|u^\top v_t| + \sqrt{2\operatorname{lr}(1/\delta)}\|u_\perp\|\|v_t\|$$

$$\leq 2\operatorname{lr}(1/\delta)(|u^\top v_t| + \|u_\perp\|\|v_t\|) \leq 2\sqrt{2}\operatorname{lr}(1/\delta)\sqrt{|u^\top v_t|^2 + \|u_\perp\|^2\|v_t\|^2}$$
$$= 2\sqrt{2}\operatorname{lr}(1/\delta)\|v_t\|,$$

whenever $\delta \in (0, 1/e]$. Consequently, by (Jin et al., 2019b, Lemma 1), we see that $Q_t \mid \mathcal{F}_{t-1}$ is $8\operatorname{lr}(1/\delta)\sqrt{d}\|v_t\|$-norm-subGaussian.

It follows easily that $M_t Q_t \mid \mathcal{F}_{t-1}$ is mean-zero and $8\operatorname{lr}(1/\delta)\|M_t\|_2\|v_t\|\sqrt{d}$-norm-subGaussian. Hence, by (Jin et al., 2019a, Lemma 6), we know that there exists an absolute constant $c > 0$ such that for any $\theta > 0$ and $\delta > 0$, we have that with probability at least $1 - \delta$,

$$\left\|\sum_{t=0}^{\tau-1} M_t Q_t\right\| \leq c\theta \sum_{t=0}^{\tau-1} d(\operatorname{lr}(1/\delta))^2\|M_t\|_2^2\|v_t\|^2 + \frac{1}{\theta}\log(2d/\delta).$$

Now, consider denoting the event

$$A := \bigcup_{t=0}^{\tau-1} A_t = \left\{|Z_t^\top v_t| \in \left(\alpha(\delta)\|v_t\|, \sqrt{2\operatorname{lr}(1/\delta)})\|v_t\|\right), \ \forall t \in \{0, \ldots, \tau-1\}\right\}$$

By the union bound and Eq. (9), we note that

$$\mathbb{P}(A) \geq 1 - \tau C\delta.$$

Moreover, note that on the event $A$, $\sum_{t=0}^{\tau-1} M_t Q_t = \sum_{t=0}^{\tau-1} M_t(Z_t Z_t^\top - I)v_t$. Hence,

$$\mathbb{P}\left(\left\|\sum_{t=0}^{\tau-1} M_t(Z_t Z_t^\top - I)v_t\right\| \leq c\theta \sum_{t=0}^{\tau-1} d(\operatorname{lr}(1/\delta))^2\|M_t\|_2^2\|v_t\|^2 + \frac{1}{\theta}\log(2d/\delta)\right)$$
$$\geq \mathbb{P}\left(\left\|\sum_{t=0}^{\tau-1} M_t Y_t\right\| \leq c\theta \sum_{t=0}^{\tau-1} d(\operatorname{lr}(1/\delta))^2\|M_t\|_2^2\|v_t\|^2 + \frac{1}{\theta}\log(2d/\delta), \text{ and } A \text{ happens}\right)$$
$$\geq 1 - \left(\mathbb{P}\left(\left\|\sum_{t=0}^{\tau-1} M_t Y_t\right\| \geq c\theta \sum_{t=0}^{\tau-1} d(\operatorname{lr}(1/\delta))^2\|M_t\|_2^2\|v_t\|^2 + \frac{1}{\theta}\log(2d/\delta)\right) + \mathbb{P}(A^c)\right)$$
$$\geq 1 - (\delta + \tau C\delta).$$

Now, by rescaling $\delta$ to $\delta/(C\tau + 1)$, we get the desired result. Note this $C$ is different from the $C$ in the statement of the lemma by an absolute multiplicative factor. $\qquad\square$

## C.4 SUB-WEIBULL RANDOM VARIABLES

In our work, we occasionally require bounding sums of heavy-tailed distribution, e.g. higher powers of $\|Z\|$ where $Z \sim N(0, I)$. To this end, we consider the following definition of sub-Weibull random variables.

**Definition 7.** We say that a random variable $X \in \mathbb{R}$ is *sub-Weibull(K, $\alpha$)* for some $K, \alpha > 0$,

$$\mathbb{P}(|X| \geq s) \leq 2\exp(-(s/K)^{1/\alpha}) \quad \forall s \geq 0.$$

For instance, the standard normal distribution is sub-Weibull$(1, \frac{1}{2})$. From the way we define the tail parameter $\alpha$, the larger the $\alpha$, the heavier the tail of the distribution.

In our work, we need to show that the sum of sub-Weibull random variables is again sub-Weibull, which is ensured by the following result

**Lemma 9.** *Suppose $X$ and $Y$ are sub-Weibull($K_X, \alpha$) and sub-Weibull($K_Y, \alpha$) respectively. Then, $XY$ is sub-Weibull($C(K_X \cdot K_Y), 2\alpha$) and $X + Y$ is sub-Weibull($C(K_X + K_Y), \alpha$) for some absolute constant $C > 0$.*

A helpful result is the following, which bounds the sum of identically distributed sub-Weibull random variables.

**Lemma 10** (Corollary 3.1 in Vladimirova et al. (2020)). *Suppose $X_1, \ldots, X_n$ are identically distributed $(K', \alpha)$ sub-Weibull random variables. Then, for some absolute constant $c > 0$, for all $s \geq ncK'$, we have*

$$\mathbb{P}\left(\left|\sum_{i=1}^n X_i\right| \geq s\right) \leq \exp\left(-\left(\frac{s}{ncK'}\right)^{1/\alpha}\right)$$

In our work, we frequently need to bound sums of the $k$-th power of the norm of a standard $d$-dimensional Gaussian. We do so using Lemma 10.

**Lemma 11.** *Suppose $X_i \overset{i.i.d}{\sim} N(0, I_d)$ for $i \in [n]$. Then, for any $k \in \mathbb{Z}^+$, there exists absolute constants $c, C > 0$ such that for any $\delta \in (0, 1)$, with probability at least $1 - \delta$,*

$$\left|\sum_{i=1}^n \|X_i\|^{2k}\right| \leq nCc^k d^k (1 + (\log(1/\delta))^k).$$

*In particular, for any $\delta \in (0, 1/e)$ such that $\log(1/\delta) \geq 1$, it follows that*

$$\left|\sum_{i=1}^n \|X_i\|^{2k}\right| \leq 2nCc^k d^k (\log(1/\delta))^k.$$

*Proof.* First, observe that for any $j \in [d]$, $(X_i)_j^2$, being subExponential, is $(1, 1)$-subWeibull. Then, by Lemma 9, $\|X_i\|^2 = \sum_{j=1}^d (X_i)_j^2$ is $(cd, 1)$ for some absolute constant $c$. Now, it follows from definition of sub-Weibullness in Definition 7 that $\|X_i\|^{2k}$ is $(c^k d^k, k)$-subWeibull. Hence, applying Lemma 10, we have that there exists absolute constant $C > 0$ such that for any $s \geq nCc^k d^k$,

$$\mathbb{P}\left(\left|\sum_{i=1}^n \|X_i\|^{2k}\right| \geq s\right) \leq \exp\left(-\left(\frac{s}{nCc^k d^k}\right)^{1/k}\right)$$

Choosing $s = (1 + (\log(1/\delta))^k)nCc^k d^k$, we arrive then at the desired result. □

## C.5 SUPERMARTINGALE CONCENTRATION INEQUALITIES

We first state and prove a supermartingale-type concentration inequality of the form we later require.

**Lemma 12.** *Consider a filtration of sigma-algebras $\mathcal{F}_0 \subset \mathcal{F}_1 \subset \cdots \subset \mathcal{F}_{n-1} \subset \mathcal{F}_n$ and a sequence of random variables $X_1, \ldots, X_n$ such that $X_i \in \mathcal{F}_i$. Suppose that*

$$\mathbb{P}_{\mathcal{F}_{i-1}}(X_i \leq a) = 1 \qquad \text{and} \qquad \mathbb{P}_{\mathcal{F}_{i-1}}(X_i \leq -b) \geq p \tag{10}$$

*for some $a, b > 0$ and $0 < p \leq \frac{1}{2}$. Then, for any $0 < \mu \leq b$ such that $|-b + \mu| \geq \frac{1-p}{p}(a + \mu)$, we have*

$$\mathbb{P}\left(\sum_{i=1}^n X_i \geq -n\mu + s\right) \leq \exp\left(-\frac{s^2}{4n(b-\mu)^2}\right), \qquad \forall s > 0.$$

*Proof.* Observe that by Markov's inequality, for any $\lambda > 0$,

$$\mathbb{P}\left(\sum_{i=1}^n X_i \geq -n\mu + s\right) = \mathbb{P}\left(\exp\left(\lambda \sum_{i=1}^n (X_i + \mu)\right) \geq \exp(\lambda s)\right) \leq \frac{\mathbb{E}\left[\exp\left(\lambda \sum_{i=1}^n (X_i + \mu)\right)\right]}{\exp(\lambda s)}.$$

Now, observe that

$$\mathbb{E}\left[\exp\left(\lambda \sum_{i=1}^n (X_i + \mu)\right)\right] = \mathbb{E}\left[\mathbb{E}_{\mathcal{F}_{n-1}}\left[\exp\left(\lambda \sum_{i=1}^n (X_i + \mu)\right)\right]\right]$$

$$= \mathbb{E}\left[\exp\left(\lambda \sum_{i=1}^{n-1} (X_i + \mu)\right) \mathbb{E}_{\mathcal{F}_{n-1}}[\exp(\lambda(X_n + \mu))]\right]. \tag{11}$$

Let us now compute $\mathbb{E}_{\mathcal{F}_{n-1}}[\exp(\lambda(X_n + \mu))]$:

$$\mathbb{E}_{\mathcal{F}_{n-1}}[\exp(\lambda(X_n + \mu))]$$

$$= \int_{(-\infty, -b]} \exp(\lambda(x + \mu)) \, \mathbb{P}_{\mathcal{F}_{n-1}}(X_n \in dx) + \int_{(-b, a]} \exp(\lambda(x + \mu)) \, \mathbb{P}_{\mathcal{F}_{n-1}}(X_n \in dx)$$

$$\leq \mathbb{P}_{\mathcal{F}_{n-1}}(X_n \leq -b) \exp(\lambda(-b + \mu)) + \mathbb{P}_{\mathcal{F}_{n-1}}(-b < X_n \leq a) \exp(\lambda(a + \mu))$$

$$\leq p \exp(\lambda(-b + \mu)) + (1 - p) \exp(\lambda(a + \mu)).$$

Then observe that by our choice of $\mu$, $-b + \mu < 0$, and that $|-b + \mu| \geq (a + \mu)\frac{1-p}{p}$. Since we assumed $p \leq \frac{1}{2}$, this means that $\frac{1-p}{p} \geq 1$ and so for any $k \geq 1$,

$$|-b + \mu| \geq (a + \mu)\frac{1 - p}{p} \implies |-b + \mu| \geq (a + \mu)\left(\frac{1 - p}{p}\right)^{1/k} \implies p|-b + \mu|^k \geq (1 - p)(a + \mu)^k.$$

Consequently, by Taylor expansion,

$$p \exp(\lambda(-b + \mu)) + (1 - p) \exp(\lambda(a + \mu))$$

$$= 1 + \sum_{k=1}^{\infty} \frac{\lambda^k (p(-b + \mu)^k + (1 - p)(a + \mu)^k)}{k!} \leq 1 + \sum_{k=1}^{\infty} \frac{\lambda^k (p(-b + \mu)^k + p|-b + \mu|^k)}{k!}$$

$$= 1 + \sum_{k=1}^{\infty} \frac{\lambda^{2k} \cdot 2p|-b + \mu|^{2k}}{(2k)!} \leq 1 + \sum_{k=1}^{\infty} \frac{\lambda^{2k}|-b + \mu|^{2k}}{(k)!}$$

$$= \exp(\lambda^2(-b + \mu)^2),$$

which leads to

$$\mathbb{E}_{\mathcal{F}_{n-1}}[\exp(\lambda(X_n + \mu))] \leq \exp(\lambda^2(-b + \mu)^2).$$

Now, continuing from Eq. (11), we have that

$$\mathbb{E}\left[\exp\left(\lambda \sum_{i=1}^{n}(X_i + \mu)\right)\right] \leq \mathbb{E}\left[\exp\left(\lambda \sum_{i=1}^{n-1}(X_i + \mu)\right) \mathbb{E}_{\mathcal{F}_{n-1}} \mathbf{1}\left[\exp(\lambda(X_n + \mu))\right]\right]$$

$$\leq \mathbb{E}\left[\exp\left(\lambda \sum_{i=1}^{n-1}(X_i + \mu)\right) \exp(\lambda^2(b - \mu)^2)\right]$$

$$\leq \dots$$

$$\leq \exp(n\lambda^2(b - \mu)^2).$$

Thus, for any $\lambda > 0$ and $s \geq 0$,

$$\mathbb{P}\left(\sum_{i=1}^{n} X_i \geq -n\mu + s\right) \leq \frac{\mathbb{E}\left[\exp(\lambda(\sum_{i=1}^{n}(X_i + \mu)))\right]}{\exp(\lambda s)}$$

$$\leq \exp(n\lambda^2(b - \mu)^2 - \lambda s)$$

By finding the minimizing $\lambda$, we find that

$$\mathbb{P}\left(\sum_{i=1}^{n} X_i \geq -n\mu + s\right) \leq \exp\left(-\frac{s^2}{4n(b - \mu)^2}\right),$$

which completes the proof. $\qquad\square$

We will later require a weakened form of a supermartingale concentration inequality, as stated and proven below.

**Proposition 3** (Weakened supermartingale concentration inequality). *Consider a filtration of sigma-algebras $\mathcal{F}_0 \subset \mathcal{F}_1 \cdots \subset \mathcal{F}_n$ and a sequence of random variables $X_1, \dots, X_n$ such that $X_i \in \mathcal{F}_i$. Consider for each $i \in \{1, \dots, n\}$ a bad set $B_i$ where $\mathbb{1}_{B_i} \in \mathcal{F}_{i-1}$, and suppose*

$$\mathbb{P}_{\mathcal{F}_{i-1}}(X_i \mathbb{1}_{B_i^c} \leq a) = 1 \qquad and \qquad \mathbb{P}_{\mathcal{F}_{i-1}}(X_i \mathbb{1}_{B_i^c} \leq -b) \geq p$$

*for some $a, b > 0$ and $0 \leq p \leq 1/2$. Then, for any $0 < \mu \leq b$ such that $|-b + \mu| \geq \frac{1-p}{p}(a + \mu)$, we have*

$$\mathbb{P}\left(\sum_{i=1}^{n} X_i \geq -n\mu + s\right) \leq \exp\left(-\frac{s^2}{4n(b-\mu)^2}\right) + \sum_{i=1}^{n} \mathbb{P}(X_i \in B_i), \qquad \forall s > 0.$$

*Proof.* We define $Q_i := X_i \mathbb{1}_{B_i^c}$. We can then apply Lemma 12 and get

$$\mathbb{P}\left(\sum_{i=1}^{n} Q_i \geq -n\mu + s\right) \leq \exp\left(-\frac{s^2}{4n(b-\mu)^2}\right).$$

Since $\mathbb{P}(X_i \neq Q_i \text{ for some } i \in [n]) \leq \sum_i \mathbb{P}(X_i \in B_i)$, it follows that

$$\mathbb{P}\left(\sum_{i=1}^{n} X_i \geq -n\mu + s\right) \leq \exp\left(-\frac{s^2}{4n(b-\mu)^2}\right) + \sum_{i=1}^{n} \mathbb{P}(X_i \in B_i),$$

which completes the proof. $\qquad\square$

## D    FUNCTION DECREASE IN LARGE GRADIENT REGIME

In this section, we show that sufficient function decrease can be made across the iterations with large gradients. We first restate and prove the function decrease lemma (Lemma 1), first introduced in the main text. We then provide a detailed roadmap of our proof in the subsequent discussion following the proof of Lemma 1.

**Lemma 1** (Function decrease for batch zeroth-order optimization). *Suppose at each time $t$, the algorithm performs the update step (with batch-size parameter $1 \leq m \leq d$)*

$$x_{t+1} = x_t - \eta\left(g_u^{(m)}(x_t) + Y_t\right),$$

*where*

$$g_u^{(m)}(x_t) = \frac{1}{m}\sum_{i=1}^{m} \frac{f(x_t + uZ_{t,i}) - f(x_t - uZ_{t,i})}{2u} Z_{t,i},$$

*where each $Z_{t,i}$ is drawn i.i.d from $N(0, I)$, $u > 0$ is the smoothing radius, and $Y_t \sim N(0, \frac{r^2}{d}I)$ with $r > 0$ denoting the perturbation radius.*

*Then, there exist absolute constants $c_1 > 0, C_1 \geq 1$ such that, for any $T \in \mathbb{Z}^+$ and $T \geq \tau > 0$, $\alpha > 0$ and $\delta \in (0, 1/e]$, upon defining $\mathcal{H}_{0,\tau}(\delta)$ to be the event on which the inequality*

$$\begin{aligned}
f(x_\tau) - f(x_0) \leq & -\frac{3\eta}{4}\sum_{t=0}^{\tau-1}\frac{1}{m}\sum_{i=1}^{m}\left|Z_{t,i}^\top \nabla f(x_t)\right|^2 + \left(\frac{\eta}{\alpha} + \frac{c_1 L\eta^2\chi^3 d}{m}\right)\sum_{t=0}^{\tau-1}\|\nabla f(x_t)\|^2 \\
& + \tau\eta u^4\rho^2 \cdot c_1 d^3\left(\log\frac{T}{\delta}\right)^3 + \tau L\eta^2 u^4\rho^2 \cdot c_1 d^4\left(\log\frac{T}{\delta}\right)^4 \\
& + \eta c_1 r^2(\alpha + \eta L)\log\frac{T}{\delta} + \tau c_1 L\eta^2 r^2
\end{aligned} \tag{3}$$

*is satisfied (where $\chi := \log(C_1 dmT/\delta)$), we have*

$$\mathbb{P}(\mathcal{H}_{0,\tau}(\delta)) \geq 1 - \frac{(\tau+4)\delta}{T}, \qquad \mathbb{P}(\cap_{\tau=1}^{\tau'}\mathcal{H}_{0,\tau}(\delta)) \geq 1 - \frac{5\tau'\delta}{T}$$

*for any $0 \leq \tau' \leq T$.*

*Proof.* First, for each $t \in \{-1, \ldots, \tau\}$, we define $\mathcal{F}_t$ to be the sigma-algebra generated by

$$x_0, \quad (\{Z_{0,i}\}_{i=1}^m, \ldots, \{Z_{t,i}\}_{i=1}^m), \quad (Y_0, \ldots, Y_t).$$

Note that $\mathcal{F}_{-1}$ is the sigma-algebra generated only by $x_0$.

By Taylor expansion, for any $x, y \in \mathbb{R}^d$, there exists $\alpha \in [0, 1]$ such that $f(x + y) = f(x) + \langle \nabla f(x), y \rangle + \frac{1}{2} y^\top \nabla^2 f(x + \alpha y) \, y$. Therefore

$$\frac{f(x_t + uZ_{t,i}) - f(x_t - uZ_{t,i})}{2u} = \langle \nabla f(x), Z_{t,i} \rangle + \frac{u}{2} Z_{t,i}^\top \tilde{H}_{t,i} Z_{t,i}$$

with

$$\tilde{H}_{t,i} = \frac{\nabla^2 f(x + \alpha_{i,+} u Z_{t,i}) - \nabla^2 f(x - \alpha_{i,-} u Z_{t,i})}{2}$$

for some $\alpha_{i,\pm} \in [0, 1]$, and

$$x_{t+1} = x_t - \eta \left( \frac{1}{m} \sum_{i=1}^m \left( Z_{t,i} Z_{t,i}^\top \nabla f(x_t) + \frac{u}{2} Z_{t,i} Z_{t,i}^\top \tilde{H}_{t,i} Z_{t,i} \right) + Y_t \right) \tag{12}$$

By the $\rho$-Hessian Lipschitz property of $f$, it follows that $\left\| \tilde{H}_{t,i} \right\| \leq \rho u \|Z_{t,i}\|$

Observe that

$$
\begin{aligned}
f(x_{t+1}) &\overset{(i)}{\leq} f(x_t) + \langle x_{t+1} - x_t, \nabla f(x_t) \rangle + \frac{L}{2} \|x_{t+1} - x_t\|^2 \\
&\overset{(ii)}{=} f(x_t) - \eta \frac{1}{m} \sum_{i=1}^m \left| Z_{t,i}^\top \nabla f(x_t) \right|^2 - \eta \frac{1}{m} \sum_{i=1}^m \frac{u}{2} Z_{t,i}^\top \nabla f(x_t) \cdot Z_{t,i}^\top \tilde{H}_{t,i} Z_{t,i} - \eta \langle \nabla f(x_t), Y_t \rangle \\
&\quad + \frac{L\eta^2}{2} \left\| \frac{1}{m} \sum_{i=1}^m \left( Z_{t,i} Z_{t,i}^\top \nabla f(x_t) + \frac{u}{2} Z_{t,i} Z_{t,i}^\top \tilde{H}_{t,i} Z_{t,i} \right) + Y_t \right\|^2 \\
&\overset{(iii)}{\leq} f(x_t) - \frac{\eta}{m} \sum_{i=1}^m \left| Z_{t,i}^\top \nabla f(x_t) \right|^2 + \frac{\eta}{m} \sum_{i=1}^m \left( \frac{\left| Z_{t,i}^\top \nabla f(x_t) \right|^2}{4} + \frac{u^2 \left| Z_{t,i}^\top \tilde{H}_{t,i} Z_{t,i} \right|^2}{4} \right) - \eta \langle \nabla f(x_t), Y_t \rangle \\
&\quad + \frac{L\eta^2}{2} \left( 2 \left\| \frac{1}{m} \sum_{i=1}^m Z_{t,i} Z_{t,i}^\top \nabla f(x_t) \right\|^2 + u^2 \left\| \frac{1}{m} \sum_{i=1}^m Z_{t,i} Z_{t,i}^\top \tilde{H}_{t,i} Z_{t,i} \right\|^2 + 4 \|Y_t\|^2 \right) \\
&\overset{(iv)}{\leq} f(x_t) - \frac{3\eta}{4m} \sum_{i=1}^m \left| Z_{t,i}^\top \nabla f(x_t) \right|^2 + \frac{\eta u^2}{m} \sum_{i=1}^m \frac{u^2 \rho^2 \|Z_{t,i}\|^6}{4} - \eta \langle \nabla f(x_t), Y_t \rangle \\
&\quad + \frac{L\eta^2}{2} \left( 2 \left\| \frac{1}{m} \sum_{i=1}^m Z_{t,i} Z_{t,i}^\top \nabla f(x_t) \right\|^2 + \frac{u^2}{m} \sum_{i=1}^m u^2 \rho^2 \|Z_{t,i}\|^8 + 4 \|Y_t\|^2 \right) \\
&\leq f(x_t) - \frac{3\eta}{4m} \sum_{i=1}^m \left| Z_{t,i}^\top \nabla f(x_t) \right|^2 + \frac{\eta u^4 \rho^2}{4m} \sum_{i=1}^m \|Z_{t,i}\|^6 + \frac{L\eta^2 u^4 \rho^2}{2m} \sum_{i=1}^m \|Z_{t,i}\|^8 - \eta \langle \nabla f(x_t), Y_t \rangle \\
&\quad + \frac{L\eta^2}{2} \left( 2 \left\| \frac{1}{m} \sum_{i=1}^m Z_{t,i} Z_{t,i}^\top \nabla f(x_t) \right\|^2 + 4 \|Y_t\|^2 \right) \tag{13}
\end{aligned}
$$

Above, to derive (i), we used the $L$-smoothness of $f$. To derive (ii), we used the expression for $(x_{t+1} - x_t)$ shown in Eq. (12). To derive (iii), we used the fact that $ab \leq (a^2 + b^2)/2$ for any $a, b \in \mathbb{R}_{\geq 0}$, as well as two applications of the fact that $\|a + b\|^2 \leq 2(\|a\|^2 + \|b\|^2)$ for any two vectors $a, b \in \mathbb{R}^d$. To derive (iv), we used the fact that $\left\| \tilde{H}_{t,i} \right\| \leq \rho u \|Z_{t,i}\|$.

To continue from Eq. (13), we first observe that we can rewrite

$$Z_{t,i} Z_{t,i}^\top \nabla f(x_t) = (Z_{t,i} Z_{t,i}^\top - I) \nabla f(x_t) + \nabla f(x_t),$$

so that

$$\left\| \frac{1}{m} \sum_{i=1}^m Z_{t,i} Z_{t,i}^\top \nabla f(x_t) \right\|^2 \leq 2 \left\| \frac{1}{m} \sum_{i=1}^m (Z_{t,i} Z_{t,i}^\top - I) \nabla f(x_t) \right\|^2 + 2 \|\nabla f(x_t)\|^2.$$

Observe that we can apply the bound in Proposition 2 to $\left\|\sum_{i=1}^{m}(Z_{t,i}Z_{t,i}^{\top} - I)\nabla f(x_t)\right\|$, and since $Z_{t,i}$ is independent of $\mathcal{F}_{t-1}$ for all $i$, we know there exist absolute constants $\mathfrak{c}_1 > 0, C_1 \geq 1$ such that for any $\delta \in (0, 1/e]$ and $\theta > 0$, with probability at least $1 - \delta$ conditioned on $\mathcal{F}_{t-1}$,

$$\left\|\sum_{i=1}^{m}(Z_{t,i}Z_{t,i}^{\top} - I)\nabla f(x_t)\right\| \leq \mathfrak{c}_1\theta\sum_{i=1}^{m} d(\mathrm{lr}(C_1 m/\delta))^2\|\nabla f(x_t)\|^2 + \frac{1}{\theta}\log(C_1 dm/\delta)$$

$$= \mathfrak{c}_1\theta md(\mathrm{lr}(C_1 m/\delta))^2\|\nabla f(x_t)\|^2 + \frac{1}{\theta}\log(C_1 dm/\delta). \quad (14)$$

Moreover, since $C_1 \geq 1$, $\log(C_1 dm/\delta)$ and $\mathrm{lr}(C_1 m/\delta)$ both are at least 1 as long as $\delta \in (0, 1/e]$. Observe that conditioned on $\mathcal{F}_{t-1}$, $\nabla f(x_t)$ is fixed. Hence, we can pick

$$\theta = \frac{1}{\sqrt{\mathfrak{c}_1 md \ \mathrm{lr}(C_1 dm/\delta)}\|\nabla f(x_t)\|}$$

which is $\mathcal{F}_{t-1}$-measurable, and plug it into Eq. (14) to find that the probability conditioned on $\mathcal{F}_{t-1}$ of the following event

$$\left\|\sum_{i=1}^{m}(Z_{t,i}Z_{t,i}^{\top} - I)\nabla f(x_t)\right\| \leq 2\sqrt{\mathfrak{c}_1}(\mathrm{lr}(C_1 dm/\delta))^{3/2}\sqrt{md}\|\nabla f(x_t)\| \quad (15)$$

is at least $1 - \delta$. By taking the total expectation, it follows that the event has a total probability at least $1 - \delta$. Thus, with probability at least $1 - \delta$,

$$\left\|\frac{1}{m}\sum_{i=1}^{m}Z_{t,i}Z_{t,i}^{\top}\nabla f(x_t)\right\|^2 \leq 2\left\|\frac{1}{m}\sum_{i=1}^{m}(Z_{t,i}Z_{t,i}^{\top} - I)\nabla f(x_t)\right\|^2 + 2\|\nabla f(x_t)\|^2$$

$$\leq 4\mathfrak{c}_1(\mathrm{lr}(C_1 dm/\delta))^3\frac{d}{m}\|\nabla f(x_t)\|^2 + 2\|\nabla f(x_t)\|^2$$

$$\leq \mathfrak{c}_2(\mathrm{lr}(C_1 dm/\delta))^3\frac{d}{m}\|\nabla f(x_t)\|^2, \quad (16)$$

where the last inequality comes from the fact that $\mathrm{lr}(C_1 dm/\delta) \geq 1$, our assumption at the outset of the appendix that $d \geq m$, and denoting $\mathfrak{c}_2 := 4\mathfrak{c}_1 + 2$.

Denote the event $\tilde{H}_{0,\tau}(\delta)$ as the event that

$$f(x_\tau) - f(x_0) \leq -\sum_{t=0}^{\tau-1}\frac{3\eta}{4m}\sum_{i=1}^{m}\left|Z_{t,i}^{\top}\nabla f(x_t)\right|^2 + L\eta^2\frac{\mathfrak{c}_2 d(\mathrm{lr}(C_1 dm/\delta))^3}{m}\sum_{t=0}^{\tau-1}\|\nabla f(x_t)\|^2$$

$$+ \frac{\eta u^4 \rho^2}{4m}\sum_{t=0}^{\tau-1}\sum_{i=1}^{m}\|Z_{t,i}\|^6 + \frac{L\eta^2 u^4 \rho^2}{2m}\sum_{t=0}^{\tau-1}\sum_{i=1}^{m}\|Z_{t,i}\|^8$$

$$- \eta\sum_{t=0}^{\tau-1}\langle\nabla f(x_t), Y_t\rangle + 2L\eta^2\sum_{t=0}^{\tau-1}\|Y_t\|^2 \quad (17)$$

holds.

Now, continuing from Eq. (13), and using the bound in Eq. (16), summing over the iterations from $t = 0$ to $\tau - 1$, we find using the union bound that $\mathbb{P}(\cap_{\tau=1}^{\tau'}\tilde{H}_{0,\tau}(\delta)) \geq 1 - \tau'\delta$, $\mathbb{P}(\tilde{H}_{0,\tau}(\delta)) \geq 1 - \tau\delta$.

Now, by Lemma 6, for any $\delta \in (0, 1), \alpha > 0$, with probability at least $1 - \delta$, there exists an absolute constant $\mathfrak{c}_3 > 0$ such that

$$-\eta\sum_{t=0}^{\tau-1}\langle\nabla f(x_t), Y_t\rangle \leq \eta\left(\frac{1}{\alpha}\sum_{t=0}^{\tau-1}\|\nabla f(x_t)\|^2 + \mathfrak{c}_3\alpha r^2\log(1/\delta)\right). \quad (18)$$

Meanwhile, since $Y_t \sim N(0, (r^2/d)I)$, $\|Y_t\|^2$ is sub-exponential with sub-exponential norm $cr^2$ for some absolute constant $c > 0$, and by Bernstein's inequality (Lemma 8), there exists some absolute constant $\mathfrak{c}_4 > 0$ such that

$$\sum_{t=0}^{\tau-1}\|Y_t\|^2 \leq \mathfrak{c}_4 r^2(\tau + \log(1/\delta)) \quad (19)$$

with probability at least $1 - \delta$.

To bound $\sum_{t=0}^{\tau-1} \frac{1}{m} \sum_{i=1}^{m} \|Z_{t,i}\|^6$ and $\sum_{t=0}^{\tau-1} \frac{1}{m} \sum_{i=1}^{m} \|Z_{t,i}\|^8$, both sums of heavy tailed Gaussian moments, we use Lemma 11, which states that for any $k \in \mathbb{Z}^+$ and $\delta \in (0, 1)$, with probability at least $1 - \delta$,

$$\frac{1}{m} \sum_{t=0}^{\tau-1} \sum_{i=1}^{m} \|Z_{t,i}\|^{2k} \leq \mathfrak{c}_5 \tau (\mathfrak{c}_6)^k d^k (1 + (\log(1/\delta))^k) \tag{20}$$

for some absolute constants $\mathfrak{c}_5, \mathfrak{c}_6 > 0$. As in the statement of the proof, using $\chi := \mathrm{lr}(C_1 dm/\delta)$ to ease the notation, denote the event that

$$\begin{aligned}
f(x_\tau) - f(x_0) \leq \; & -\frac{3\eta}{4} \sum_{t=0}^{\tau-1} \frac{1}{m} \sum_{i=1}^{m} |Z_{t,i}^\top \nabla f(x_t)|^2 + \left(\frac{\eta}{\alpha} + \frac{\mathfrak{c}_2 L \eta^2 \chi^3 d}{m}\right) \sum_{t=0}^{\tau-1} \|\nabla f(x_t)\|^2 \\
& + \frac{\tau \eta u^4 \rho^2}{2} \cdot \mathfrak{c}_5 \mathfrak{c}_6^3 d^3 \left(\log \frac{1}{\delta}\right)^3 + \tau L \eta^2 u^4 \rho^2 \cdot \mathfrak{c}_5 \mathfrak{c}_6^4 d^4 \left(\log \frac{1}{\delta}\right)^4 \\
& + \eta (\mathfrak{c}_3 \alpha r^2 + 2\mathfrak{c}_4 \eta L r^2) \log \frac{1}{\delta} + 2\mathfrak{c}_4 L \eta^2 \tau r^2
\end{aligned}$$

holds as $\mathcal{H}_{0,\tau}(\delta)$.

Plugging Eq. (18), Eq. (19), and Eq. (20) into Eq. (17), by union bound, we see that

$$\mathbb{P}(\cap_{\tau=1}^{\tau'} \mathcal{H}_{0,\tau}(\delta)) \geq 1 - (\tau' + 4\tau')\delta = 1 - 5\tau'\delta, \qquad \mathbb{P}(\mathcal{H}_{0,\tau}) \geq 1 - (\tau + 4)\delta.$$

The final result then follows by rescaling $\delta$ to $\frac{\delta}{T}$ and denoting $c_1 := \max\{\mathfrak{c}_2, \mathfrak{c}_3, 2\mathfrak{c}_4, \mathfrak{c}_5 \mathfrak{c}_6^3/2, \mathfrak{c}_5 \mathfrak{c}_6^4\}$.
□

**Outline of proof approach.** Similar to the first-order setting, our goal is to show that we can arrive at a contradiction $f(x_T) < \min_x f(x)$ when there is a large number of steps at which $\|\nabla f(x_t)\| \geq \epsilon$. Roughly speaking, as Eq. (3) shows, we need to prove a lower bound of the form

$$\sum_{t=0}^{T-1} \frac{1}{m} \sum_{i=1}^{m} \|Z_{t,i}^\top \nabla f(x_t)\|^2 \geq \Omega\left(\frac{1}{\alpha} + \frac{c_1 L \eta \chi^3 d}{m}\right) \sum_{t=0}^{T-1} \|\nabla f(x_t)\|^2 \tag{21}$$

for some $\alpha$ which is not too large (an example would be picking $\alpha$ such that it only scales logarithmically in the problem parameters). However, it is tricky to prove such a lower-bound in the zeroth-order setting. In particular, for small batch-sizes $m$, $\frac{1}{m} \sum_{i=1}^{m} \|Z_{t,i}^\top \nabla f(x_t)\|^2$ could be small even as $\|\nabla f(x_t)\|^2$ is large; this is because for each $i \in [m]$, $Z_{t,i}$ could have a negligible component in the $\nabla f(x_t)$ direction. This necessitates a more careful analysis to prove a bound similar to Eq. (21). We do so using the following approach.

1. Intuitively, whilst for each individual iteration $t$, $\frac{1}{m} \sum_{i=1}^{m} \|Z_{t,i}^\top \nabla f(x_t)\|^2$ could be small even as $\|\nabla f(x_t)\|^2$ is large, in a small number of (consecutive) iterations $\{t_0, \ldots, t_0 + t_f\}$, with high probability, there will be at least one iteration $t$ within $\{t_0, \ldots, t_0 + t_f - 1\}$, such that $\frac{1}{m} \sum_{i=1}^{m} \|Z_{t,i}^\top \nabla f(x_t)\|^2 = \Omega(\|\nabla f(x_t)\|^2)$. We formalize this intuition in Lemma 14. Thus, we consider breaking the time-steps into chunks where each chunk has $t_f$ consecutive iterations.

2. Consider any such interval $\{t_0, \ldots, t_0 + t_f - 1\}$. There are two cases to consider.

   (a) The first case is when the gradient throughout all $t_f$ iterations is large enough to dominate the perturbation terms. Intuitively, in this case, it is not hard to see that given appropriate parameter choices, the gradient will change little throughout the $t_f$ iterations. In fact, as we formalize in Lemma 16, for an appropriate choice of $t_f$ and $\eta$, we can show that

   $$\frac{1}{2}\|\nabla f(x_{t_0})\| \leq \|\nabla f(x_t)\| \leq 2\|\nabla f(x_{t_0})\| \qquad \forall t \in \{t_0, \ldots, t_0 + t_f - 1\}.$$

As a result, combined with point 1, we see that

$$\sum_{t=t_0}^{t_0+t_f-1} \frac{1}{m} \sum_{i=1}^{m} \left\| Z_{t,i}^{\top} \nabla f(x_t) \right\|^2 \geq \Omega(\|\nabla f(x_{t_0})\|^2).$$

Thus, by choosing $\alpha$ and $\eta$ judiciously, for such intervals, it is possible to show that

$$\sum_{t=t_0}^{t_0+t_f-1} \frac{1}{m} \sum_{i=1}^{m} \left\| Z_{t,i}^{\top} \nabla f(x_t) \right\|^2 \geq \Omega(\|\nabla f(x_{t_0})\|^2) \geq \Omega\left(\frac{1}{\alpha} + \frac{c_1 L \eta \chi^3 d}{m}\right) \sum_{t=t_0}^{t_0+t_f-1} \|\nabla f(x_t)\|^2$$

$$= \Omega\left(\frac{1}{\alpha} + \frac{c_1 L \eta \chi^3 d}{m}\right) \Omega\left(t_f \|\nabla f(x_{t_0})\|^2\right)$$

Thus, in these intervals, it is possible to obtain function improvement on the order of $\eta \Omega(\|\nabla f(x_{t_0})\|^2)$.

(b) The remaining case is when the gradient is small and dominated by the perturbation terms in any one of the $t_f$ iterations. In this case, as we show in Lemma 17, for each of the $t_f$ iterations, the gradient will be small and on the same scale as the perturbation terms. In turn, by choosing $r, u$ and $\eta$ appropriately, we can make the perturbation terms small. Thus, whilst these intervals may not contribute to function decrease, they also contribute little in the way of function increase.

3. When there are at least $T/4$ iterations with large gradient (i.e. $\|\nabla f(x_t)\| \geq \epsilon$), assuming $t_f$ divides $T$, it follows that there are at least $T/(4t_f)$ intervals of length $t_f$ where one iteration in the interval contains a large gradient. By choosing $u, r$ and $\eta$ appropriately such they are dominated by $\epsilon$, it is possible to show that with high probability, such an interval cannot belong to the second case above, and must instead be from the first case. Since $\|\nabla f(x_t)\| \approx \|\nabla f(x_{t_0})\|$ for each $t \in \{t_0, \dots, t_0 + t_f - 1\}$ in this case, and we know that one of the iterations has a gradient with size at least $\epsilon$, it follows that we make function decrease progress of at least $\eta \Omega(\epsilon^2)$ for such intervals. By appropriately choosing $\eta, u$ and $r$ to limit the effects of the intervals of the second form, we can then show a contradiction of the form $f(x_T) < f^*$. We demonstrate this formally in Proposition 4.

We formalize our approach in the following series of results. First, for analytical convenience, we prove the following result showing that for any $t$, the perturbation terms $\|Y_t\|$ and $\frac{1}{m} \sum_{i=1}^{m} \|Z_{t,i}\|^4$ are bounded with high probability.

**Lemma 13.** *There exists an absolute constant $c_3 > 0$ such that, for any $t \in \mathbb{N}$, the event*

$$\mathcal{G}_t(\delta) := \left\{ \|Y_t\|^2 \leq c_3^2 r^2 \left(1 + \frac{\log(T/\delta)}{d}\right) \text{ and } \frac{1}{m} \sum_{i=1}^{m} \|Z_{t,i}\|^4 \leq 2c_3 d^2 \left(\log \frac{T}{\delta}\right)^2 \right\}$$

*has probability at least $1 - 2\delta/T$ for any $\delta \in (0, 1/e]$.*

*Proof.* Noting that $Y_t \sim N(0, (r^2/d)I)$, by applying Bernstein's inequality (Lemma 8), it can be shown that with probability at least $\delta/T$,

$$\|Y_t\|^2 \leq c_3^2 r^2 \left(1 + \frac{\log(T/\delta)}{d}\right),$$

where $c_3 > 0$ is some absolute constant. Then by using Lemma 11, applying the union bound, and redefining the constant $c_3$, we complete the proof. $\qquad \square$

Next, in Lemma 14, we show that in a small number of iterations, with high probability, there exists some iteration $t$ such that $\frac{1}{m} \sum_{i=1}^{m} \left|Z_{t,i}^{\top} \nabla f(x_t)\right|^2 \geq \frac{1}{2} \|\nabla f(x_t)\|^2$.

**Lemma 14.** *There exists an absolute constant $c_2 \geq 1$ such that, upon defining*

$$t_f(\delta) = \left\lceil \frac{c_2}{m} \log \frac{T}{\delta} \right\rceil, \qquad \delta > 0,$$

*and defining the event*

$$\mathcal{B}_{t_0}(\delta; k) := \bigcup_{t=t_0}^{t_0+k-1} \left\{ \frac{1}{m} \sum_{i=1}^{m} |Z_{t,i}^\top \nabla f(x_t)|^2 \geq \frac{1}{2} \|\nabla f(x_t)\|^2 \right\},$$

*we have*

$$\mathbb{P}\left(\mathcal{B}_{t_0}(\delta; k)\right) \geq 1 - \frac{\delta}{T}.$$

*for any $\delta \in (0,1)$, $t_0 \in \mathbb{N}$ and $k \geq t_f(\delta)$.*

*Proof.* Denote the event

$$E_t = \left\{ \frac{1}{m} \sum_{i=1}^{m} |Z_{t,i}^\top \nabla f(x_t)|^2 < \frac{1}{2} \|\nabla f(x_t)\|^2 \right\}.$$

Observe that, conditioned on $\mathcal{F}_{t-1}$, the set of random variables $\{\|\nabla f(x_t)\|^2 - |Z_{t,i}^\top \nabla f(x_t)|^2\}_{i=1}^{m}$ are independent, mean-zero, and subexponential with subexponential norm $\leq c\|\nabla f(x_t)\|^2$ for some absolute constant $c > 0$. Hence

$$\mathbb{P}_{\mathcal{F}_{t-1}}(E_t) = \mathbb{P}_{\mathcal{F}_{t-1}}\left( \frac{1}{m} \sum_{i=1}^{m} |Z_{t,i}^\top \nabla f(x_t)|^2 < \frac{1}{2} \|\nabla f(x_t)\|^2 \right)$$

$$= \mathbb{P}_{\mathcal{F}_{t-1}}\left( \sum_{i=1}^{m} \left( \|\nabla f(x_t)\|^2 - |Z_{t,i}^\top \nabla f(x_t)|^2 \right) > \frac{m}{2} \|\nabla f(x_t)\|^2 \right)$$

$$\leq \exp\left(-c'm\right),$$

where $c'$ is some positive absolute constant. Then, for any $t_0, k \in \mathbb{N}$,

$$\mathbb{P}\left( \frac{1}{m} \sum_{i=1}^{m} |Z_{t,i}^\top \nabla f(x_t)|^2 < \frac{1}{2} \|\nabla f(x_t)\|^2 \text{ for every } t \in [t_0, t_0 + k) \right)$$

$$= \mathbb{E}\left[ \prod_{t=t_0}^{t_0+k-1} \mathbb{1}_{E_t} \right] = \mathbb{E}\left[ \prod_{t=t_0}^{t_0+k-2} \mathbb{1}_{E_t} \cdot \mathbb{E}_{\mathcal{F}_{t_0+k-2}}\left[ \mathbb{1}_{E_{t_0+k-1}} \right] \right]$$

$$\leq \exp(-c'm) \cdot \mathbb{E}\left[ \prod_{t=t_0}^{t_0+k-2} \mathbb{1}_{E_t} \right] \leq \cdots \leq \exp(-c'mk).$$

Therefore, by letting $c_2 = \max\{1, 1/c'\}$ and

$$k \geq t_f(\delta) = \left\lceil \frac{c_2}{m} \log \frac{T}{\delta} \right\rceil,$$

we get

$$\mathbb{P}\left( \frac{1}{m} \sum_{i=1}^{m} |Z_{t,i}^\top \nabla f(x_t)|^2 < \frac{1}{2} \|\nabla f(x_t)\|^2 \text{ for every } t \in [t_0, t_0 + k) \right) \leq \frac{\delta}{T},$$

which completes the proof. $\qquad\qquad\square$

The term $t_f(\delta)$ will frequently appear in the proofs to come; in the sequel we denote

$$t_f(\delta) := \left\lceil \frac{c_2}{m} \log \frac{T}{\delta} \right\rceil, \qquad \delta \in (0, 1/e], \tag{22}$$

where $c_2 \geq 1$ is the absolute constant defined in Lemma 14.

We next show that with high probability, the norm difference term $\|\nabla f(x_{t+1}) - \nabla f(x_t)\|$ can be bounded in terms of $\|\nabla f(x_t)\|$ and the perturbation terms $\left\| \frac{u}{2m} \sum_{i=1}^{m} Z_{t,i} Z_{t,i}^\top \tilde{H}_{t,i} Z_{t,i} \right\|$ as well as $\|Y_t\|$.

**Lemma 15.** *Define*

$$\mathcal{A}_t(\delta) := \left\{ \|\nabla f(x_{t+1}) - \nabla f(x_t)\| \le \frac{\|\nabla f(x_t)\|}{8t_f(\delta)} + \eta L \left( \left\| \frac{u}{2m} \sum_{i=1}^m Z_{t,i} Z_{t,i}^\top \tilde{H}_{t,i} Z_{t,i} \right\| + \|Y_t\| \right) \right\} \tag{23}$$

*where $t_f(\delta)$ is defined in Eq. (22), and let $C_1 \ge 1$ be the corresponding absolute constants defined in Lemma 1. Then there exists an absolute constant $c_4 > 0$ such that, whenever $\eta$ satisfies*

$$\eta L \frac{c_4 (\mathrm{lr}(C_1 dmT/\delta))^{3/2} \sqrt{d}}{\sqrt{m}} \le \frac{1}{8t_f(\delta)}, \tag{24}$$

*we have*

$$\mathbb{P}(\mathcal{A}_t(\delta)) \ge 1 - \frac{\delta}{T}$$

*for any $\delta \in (0, 1/e]$ and $t \in \mathbb{Z}^+$.*

*Proof.* Since $\nabla f$ is $L$-Lipschitz, following the zeroth-order update step, we see that

$$\|\nabla f(x_{t+1}) - \nabla f(x_t)\| \le L\|x_{t+1} - x_t\| \tag{25}$$

$$= \eta L \left\| \frac{1}{m} \sum_{i=1}^m Z_{t,i} Z_{t,i}^\top \nabla f(x_t) + \frac{u}{2m} \sum_{i=1}^m Z_{t,i} Z_{t,i}^\top \tilde{H}_{t,i} Z_{t,i} + Y_t \right\|. \tag{26}$$

Now, it follows from Eq. (16) (with a slight modification in the absolute constant terms since here the norm is not squared) that there exists some absolute constant $c_4 > 0$ such that for any $\delta \in (0, 1/e]$, we have that with probability at least $1 - \delta/T$, the event

$$\left\| \frac{1}{m} \sum_{i=1}^m Z_{t,i} Z_{t,i}^\top \nabla f(x_t) \right\| \le c_4 (\mathrm{lr}(C_1 dmT/\delta))^{3/2} \sqrt{\frac{d}{m}} \|\nabla f(x_t)\|,$$

Hence, continuing from Eq. (26), it follows that with probability at least $1 - \delta/T$,

$$\|\nabla f(x_{t+1}) - \nabla f(x_t)\|$$

$$\le \eta L \left( c_4 (\mathrm{lr}(C_1 dmT/\delta))^{3/2} \sqrt{\frac{d}{m}} \|\nabla f(x_t)\| + \left\| \frac{u}{2m} \sum_{i=1}^m Z_{t,i} Z_{t,i}^\top \tilde{H}_{t,i} Z_{t,i} \right\| + \|Y_t\| \right),$$

and by plugging in the condition Eq. (24), we see that the event

$$\mathcal{A}_t(\delta) = \left\{ \|\nabla f(x_{t+1}) - \nabla f(x_t)\| \le \frac{\|\nabla f(x_t)\|}{8t_f(\delta)} + \eta L \left( \left\| \frac{u}{2m} \sum_{i=1}^m Z_{t,i} Z_{t,i}^\top \tilde{H}_{t,i} Z_{t,i} \right\| + \|Y_t\| \right) \right\}$$

has probability at least $1 - \delta/T$. $\qquad\square$

We show now that if the norm of the gradient dominates the norm of the perturbation terms, and we choose the step-size $\eta$ sufficiently small, then in a small number of iterations, the norm of the gradient does not change very much. For notational simplicity, we denote the event

$$\mathcal{E}(t_1, t_2, \delta) := \bigcap_{t=t_1}^{t_1+t_2-1} \left\{ \|\nabla f(x_t)\| > 8t_f(\delta)\eta L \left( \frac{u}{2} \left\| \frac{1}{m} \sum_{i=1}^m Z_{t,i} Z_{t,i}^\top \tilde{H}_{t,i} Z_{t,i} \right\| + \|Y_t\| \right) \right\}.$$

**Lemma 16.** *Let $\delta \in (0, 1/e]$ and $T \in \mathbb{Z}^+$ be such that $T > 2t_f(\delta) + 1$. Consider any positive integer $t'_f \le 2t_f(\delta)$, and any $t_0 \in \{0, \ldots, T - 1 - t'_f\}$. Suppose $\eta$ satisfies the condition Eq. (24). Then, on the event*

$$\mathcal{E}(t_0, t'_f, \delta) \cap \left( \bigcap_{t=t_0}^{t_0+t'_f-1} \mathcal{A}_t(\delta) \right),$$

*we have*

$$\frac{1}{2}\|\nabla f(x_0)\| \le \|\nabla f(x_t)\| \le 2\|\nabla f(x_0)\|$$

*for all $t \in \{t_0, \ldots, t_0 + t'_f - 1\}$.*

*Proof.* By plugging

$$\|\nabla f(x_t)\| > 8t_f(\delta)\eta L\left(\frac{u}{2}\left\|\frac{1}{m}\sum_{i=1}^m Z_{t,i}Z_{t,i}^\top \tilde{H}_{t,i}Z_{t,i}\right\| + \|Y_t\|\right)$$

into the definition of $\mathcal{A}_t(\delta)$, we see that, on the event $\mathcal{E}(t_0, t_f', \delta) \cap \left(\bigcap_{t=t_0}^{t_0+t_f'-1}\mathcal{A}_t(\delta)\right)$, we have

$$\|\nabla f(x_{t+1}) - \nabla f(x_t)\| \le \frac{\|\nabla f(x_t)\|}{4t_f(\delta)},$$

and consequently,

$$\left(1 - \frac{1}{4t_f(\delta)}\right)\|\nabla f(x_t)\| \le \|\nabla f(x_{t+1})\| \le \left(1 + \frac{1}{4t_f(\delta)}\right)\|\nabla f(x_t)\|,$$

which leads to

$$\left(1 - \frac{1}{4t_f(\delta)}\right)^{t-t_0}\|\nabla f(x_0)\| \le \|\nabla f(x_t)\| \le \left(1 + \frac{1}{4t_f(\delta)}\right)^{t-t_0}\|\nabla f(x_0)\|$$

for all $t \in \{t_0, \ldots, t_0 + t_f'\}$. Then, since $(1 + 1/(4x))^{2x} \le 2$ and $(1 - 1/(4x))^{2x} \ge 1/2$ for any $x \ge 1$, noting that $t_f' \le 2t_f(\delta)$, we get the desired result. $\square$

Conversely, in the following result, we show that in a small number of consecutive iterations, if the gradient is smaller than the perturbation terms in any one of the iterations, then for each of the iterations in this range, the gradient will be small and be on the same scale as the size of the perturbation terms.

**Lemma 17.** *Let $\delta \in (0, 1/e]$ and $T \in \mathbb{Z}^+$ be such that $T > 2t_f(\delta) + 1$. Consider any positive integer $t_f' \le 2t_f(\delta)$, and any $t_0 \in \{0, \ldots, T - 1 - t_f'\}$. Suppose $\eta$ satisfies the condition Eq. (24). Then, on the event*

$$\mathcal{E}^c(t_0, t_f', \delta) \cap \left(\bigcap_{t=t_0}^{t_0+t_f'-1}\mathcal{A}_t(\delta)\right) \cap \left(\bigcap_{t=t_0}^{t_0+t_f'-1}\mathcal{G}_t(\delta)\right),$$

*we have*

$$\|\nabla f(x_t)\| \le c_5 t_f(\delta)\eta L\left(u^2 d^2 \rho \left(\log\frac{T}{\delta}\right)^2 + \sqrt{1 + \frac{\log(T/\delta)}{d}}r\right) \quad \forall t \in \{t_0, t_0+1, \ldots, t_0+t_f'-1\},$$

*where $c_5$ is some absolute constant.*

*Proof.* Let $t'$ be the first iteration in $\{t_0, t_0 + 1, \ldots, t_0 + t_f' - 1\}$ such that

$$\|\nabla f(x_{t'})\| \le 8t_f(\delta)\eta L\left(\frac{u}{2}\left\|\frac{1}{m}\sum_{i=1}^m Z_{t',i}Z_{t',i}^\top \tilde{H}_{t',i}Z_{t',i}\right\| + \|Y_{t'}\|\right). \tag{27}$$

Since we are working on an event which is a subset of $\mathcal{E}^c(t_0, t_f', \delta)$, $t'$ is well-defined. By $\|\tilde{H}_{t',i}\| \le \rho u\|Z_{t',i}\|$, we see that

$$\|\nabla f(x_{t'})\| \le 8t_f(\delta)\eta L\left(\frac{u^2\rho}{2m}\sum_{i=1}^m \|Z_{t',i}\|^4 + \|Y_{t'}\|\right)$$

$$\le 8t_f(\delta)\eta L\left(c_3 u^2 d^2 \rho \left(\log\frac{T}{\delta}\right)^2 + c_3\sqrt{1 + \frac{\log(T/\delta)}{d}}r\right),$$

where we used the definition of $\mathcal{G}_t(\delta)$.

Recall that $t'$ is the first time step such that Eq. (27) holds. By deriving similarly as in the proof of Lemma 16, we can show that for any $j \in \{t_0, t_0 + 1, \ldots, t' - 1\}$,

$$\|\nabla f(x_j)\| \le 2\|\nabla f(x_{t'})\| \le 16 t_f(\delta) \eta L c_3 \left( u^2 d^2 \rho \left( \log \frac{T}{\delta} \right)^2 + \sqrt{1 + \frac{\log(T/\delta)}{d}} r \right).$$

Meanwhile, for iterations $t \in [t', t_0 + t'_f)$, by using the definitions of $\mathcal{A}_t(\delta)$ and $\mathcal{G}_t(\delta)$, we have

$$
\begin{aligned}
\|\nabla f(x_{t+1})\| &\le \left( 1 + \frac{1}{8 t_f(\delta)} \right) \|\nabla f(x_t)\| + \eta L c_3 \left( u^2 d^2 \rho \left( \log \frac{T}{\delta} \right)^2 + \sqrt{1 + \frac{\log(T/\delta)}{d}} r \right) \\
&= \left( 1 + \frac{1}{8 t_f(\delta)} \right)^{t+1-t'} \|\nabla f(x_{t'})\| \\
&\quad + \sum_{i=0}^{t-t'} \left( 1 + \frac{1}{8 t_f(\delta)} \right)^{t-t'-i} \eta L c_3 \left( u^2 d^2 \rho \left( \log \frac{T}{\delta} \right)^2 + \sqrt{1 + \frac{\log(T/\delta)}{d}} r \right) \\
&\le \left( 1 + \frac{1}{8 t_f(\delta)} \right)^{t'_f} \|\nabla f(x_{t'})\| \\
&\quad + 8 t_f(\delta) \left( \left( 1 + \frac{1}{8 t_f(\delta)} \right)^{t'_f} - 1 \right) \eta L c_3 \left( u^2 d^2 \rho \left( \log \frac{T}{\delta} \right)^2 + \sqrt{1 + \frac{\log(T/\delta)}{d}} r \right) \\
&\le e^{1/4} \cdot 8 t_f(\delta) \eta L c_3 \left( u^2 d^2 \rho \left( \log \frac{T}{\delta} \right)^2 + \sqrt{1 + \frac{\log(T/\delta)}{d}} r \right) \\
&\quad + 8 t_f(\delta)(e^{1/4} - 1) \cdot \eta L c_3 \left( u^2 d^2 \rho \left( \log \frac{T}{\delta} \right)^2 + \sqrt{1 + \frac{\log(T/\delta)}{d}} r \right) \\
&\le 16 t_f(\delta) \eta L c_3 \left( u^2 d^2 \rho \left( \log \frac{T}{\delta} \right)^2 + \sqrt{1 + \frac{\log(T/\delta)}{d}} r \right),
\end{aligned}
$$

where we used $t'_f \le 2 t_f(\delta)$ and the fact that $(1 - 1/(8x))^{2x} \le e^{1/4}$ for all $x > 0$. By defining $c_5 := 16 c_3$, we complete the proof. $\qquad\square$

We next derive a useful result showing that the function change $f(x_\tau) - f(x_0)$ can be decomposed into one component arising from intervals when the gradient dominates noise (which improves function value) and another component arising from intervals with small gradient which may add to function value but whose contributions are bounded in terms of $\eta$, $u$ and $r$. For now, we focus on the case $\tau \ge t_f(\delta)$, since it will be useful to us in proving that there cannot be more than $T/4$ iterations with large gradient.

**Lemma 18** (Function change for large $\tau$). *Let $c_1 > 0, c_4 > 0, c_5 > 0, C_1 \ge 1$ be the absolute constants defined in the statements of the previous lemmas. Let $\delta \in (0, 1/e]$, and let $\tau \ge t_f(\delta))$ be arbitrary. Consider splitting $\{0, 1 \ldots, \tau - 1\}$ into $K := \lfloor \tau / t_f(\delta) \rfloor$ intervals:*

$$
\begin{aligned}
J_k &= \{k t_f(\delta), \ldots, (k+1) t_f(\delta) - 1\}, \ \ 0 \le k < K - 1, \\
J_{K-1} &= \{(K-1) t_f(\delta), \ldots, \tau - 1\}.
\end{aligned}
$$

*Let $I_1$ denote the set of indices $k$ such that for every time-step $t$ in the interval $J_k$, the gradient dominates the noise terms as*

$$\|\nabla f(x_t)\| > 8 t_f(\delta) \eta L \left( \frac{u}{2} \left\| \frac{1}{m} \sum_{i=1}^m Z_{t,i} Z_{t,i}^\top \tilde{H}_{t,i} Z_{t,i} \right\| + \|Y_t\| \right). \tag{28}$$

*Suppose we choose $\eta$ such that*

$$\eta \le \frac{1}{L t_f(\delta)} \cdot \min \left\{ \frac{\sqrt{m}}{8 c_4 (\mathrm{lr}(C_1 dmT/\delta))^{3/2} \sqrt{d}}, \frac{m}{128 c_1 (\mathrm{lr}(C_1 dmT/\delta))^3 d} \right\}. \tag{29}$$

*Then, on the event*

$$\mathcal{E}_\tau(\delta) := \mathcal{H}_\tau(\delta) \cap \left( \bigcap_{t=0}^{\tau-1} \mathcal{A}_t(\delta) \right) \cap \left( \bigcap_{t=0}^{\tau-1} \mathcal{G}_t(\delta) \right) \cap \left( \bigcap_{k=0}^{K-2} \mathcal{B}_{kt_f(\delta)}(\delta; t_f(\delta)) \right) \cap \mathcal{B}_{(K-1)t_f(\delta)}(\delta; \tau - (K-1)t_f(\delta)),$$

*we have the following upper bound on function value change:*

$$f(x_\tau) - f(x_0) \leq -\sum_{k \in I_1} \frac{\eta}{2} \min_{t \in J_k} \|\nabla f(x_t)\|^2 + \tau \frac{c_5^2}{64} \eta^3 t_f(\delta)^2 L^2 \left( u^2 d^2 \rho \left( \log \frac{T}{\delta} \right)^2 + \sqrt{2 \log(T/\delta)} r \right)^2$$

$$+ \tau \eta u^4 \rho^2 \cdot c_1 d^3 \left( \log \frac{T}{\delta} \right)^3 + \tau L \eta^2 u^4 \rho^2 \cdot c_1 d^4 \left( \log \frac{T}{\delta} \right)^4$$

$$+ \eta c_1 r^2 (128 t_f(\delta) + \eta L) \log \frac{T}{\delta} + \tau c_1 L \eta^2 r^2. \tag{30}$$

*Moreover,* $\mathbb{P}(\mathcal{E}_\tau(\delta)) \geq 1 - \frac{(5\tau+4)\delta}{T}$.

*Proof.* Without loss of generality, we may assume that $\tau$ is a multiple of $t_f(\delta)$.[4] Then, any interval $J_k = \{t_0, \ldots, t_0 + t_f(\delta) - 1\}$ belongs to one of the following two cases:

**Case 1)** (Gradient dominates noise): Recall that this means that for every $t \in J_k$, we have

$$\|\nabla f(x_t)\| > 8 t_f(\delta) \eta L \left( \frac{u}{2} \left\| \frac{1}{m} \sum_{i=1}^m Z_{t,i} Z_{t,i}^\top \tilde{H}_{t,i} Z_{t,i} \right\| + \|Y_t\| \right).$$

By our choice of $\eta$ in Eq. (29), we can apply Lemma 16 to get

$$\min_{t \in J_k} \|\nabla f(x_t)\| \geq \frac{1}{4} \max_{t \in J_k} \|\nabla f(x_t)\|.$$

We now consider the two cases when $J$ has fewer than $t_f(\delta)$ iterations and when $J = J_k$ f

Note also that on the event $\mathcal{B}_{kt_f(\delta)}(\delta; t_f(\delta))$, there exists some $t \in J_k$ such that

$$\frac{1}{m} \sum_{i=1}^m |Z_{t,i}^\top \nabla f(x_t)|^2 \geq \frac{1}{2} \|\nabla f(x_t)\|^2.$$

This implies then that

$$\frac{1}{4} \sum_{t \in J_k} \frac{1}{m} \sum_{i=1}^m |Z_{t,i}^\top \nabla f(x_t)|^2 \geq \frac{1}{4} \min_{t \in J_k} \|\nabla f(x_t)\|^2 \geq \frac{1}{64} \max_{t \in J_k} \|\nabla f(x_t)\|^2$$

$$\geq \frac{1}{64 t_f(\delta)} \sum_{t \in J_k} \|\nabla f(x_t)\|^2. \tag{31}$$

Thus by setting $\alpha = 128 t_f(\delta)$ in Eq. (3) and by choosing $\eta$ such that

$$\frac{c_1 L \eta^2 \chi^3 d}{m} \leq \frac{\eta}{\alpha} = \frac{\eta}{128 t_f(\delta)} \iff \eta \leq \frac{m}{128 c_1 L t_f(\delta) d \chi^3},$$

it follows that

$$-\frac{3\eta}{4} \sum_{t \in J_k} \frac{1}{m} \sum_{i=1}^m |Z_{t,i}^\top \nabla f(x_t)|^2 + \left( \frac{\eta}{128 t_f(\delta)} + \frac{c_1 L \eta^2 \chi^3 d}{m} \right) \sum_{t \in J_k} \|\nabla f(x_t)\|^2$$

$$= -\frac{3\eta}{4} \sum_{t \in J_k} \frac{1}{m} \sum_{i=1}^m |Z_{t,i}^\top \nabla f(x_t)|^2 + \frac{\eta}{64 t_f(\delta)} \sum_{t \in J_k} \|\nabla f(x_t)\|^2$$

---

[4]To accommodate the last interval which has length at most $2 t_f(\delta) - 1$, we note that the results we require for the proof, namely Lemma 14, Lemma 16 and Lemma 17, all hold for any interval length $t'_f \leq 2 t_f(\delta)$.

$$\leq -\frac{\eta}{2} \sum_{t \in J_k} \frac{1}{m} \sum_{i=1}^{m} |Z_{t,i}^\top \nabla f(x_t)|^2$$

$$\leq -\frac{\eta}{2} \min_{t \in J_k} \|\nabla f(x_t)\|^2 \tag{32}$$

**Case 2**) (Gradient does not dominate noise): there exists some $t \in J_k$ such that

$$\|\nabla f(x_t)\| \leq 8t_f(\delta)\eta L \left( \frac{u}{2} \left\| \frac{1}{m} \sum_{i=1}^{m} Z_{t,i} Z_{t,i}^\top \tilde{H}_{t,i} Z_{t,i} \right\| + \|Y_t\| \right).$$

By our choice of $\eta$ in Eq. (29), we can apply Lemma 17 to get

$$\|\nabla f(x_t)\| \leq c_5 t_f(\delta)\eta L \left( u^2 d^2 \rho \left( \log \frac{T}{\delta} \right)^2 + \sqrt{1 + \frac{\log(T/\delta)}{d}} r \right) \qquad \forall t \in J_k.$$

Hence, by setting $\alpha = 128 t_f(\delta)$ in Eq. (3) and choosing $\eta$ such that

$$\frac{c_1 L \eta^2 \chi^3 d}{m} \leq \frac{\eta}{\alpha} = \frac{\eta}{128 t_f(\delta)},$$

it follows that

$$\left( \frac{\eta}{128 t_f(\delta)} + \frac{c_1 L \eta^2 \chi^3 d}{m} \right) \sum_{t \in J_k} \|\nabla f(x_t)\|^2$$

$$\leq \frac{\eta}{64 t_f(\delta)} \sum_{t \in J_k} \left( c_5 t_f(\delta)\eta L \left( u^2 d^2 \rho \left( \log \frac{T}{\delta} \right)^2 + \sqrt{1 + \frac{\log(T/\delta)}{d}} r \right) \right)^2$$

$$\leq \frac{c_5^2}{64} t_f(\delta)^2 \eta^3 L^2 \left( u^2 d^2 \rho \left( \log \frac{T}{\delta} \right)^2 + \sqrt{1 + \frac{\log(T/\delta)}{d}} r \right)^2 \tag{33}$$

Without loss of generality, we may assume that $\tau$ is a multiple of $t_f(\delta)$.[5] Then, any interval $J_k = \{t_0, \ldots, t_0 + t_f(\delta) - 1\}$ belongs to one of the following two cases:

Having studied the two cases, we may now proceed to use them to complete the proof. Let $I_1^c$ denote the complement of $I_1$ in $\{0, 1, \ldots, K-1\}$. Then,

$$-\frac{3\eta}{4} \sum_{t=0}^{\tau-1} \frac{1}{m} \sum_{i=1}^{m} |Z_{t,i}^\top \nabla f(x_t)|^2 + \left( \frac{\eta}{\alpha} + \frac{c_1 L \eta^2 \chi^3 d}{m} \right) \sum_{t=0}^{\tau-1} \|\nabla f(x_t)\|^2$$

$$= \sum_{k \in I_1} \left( -\frac{3\eta}{4} \sum_{t=\in J_k} \frac{1}{m} \sum_{i=1}^{m} |Z_{t,i}^\top \nabla f(x_t)|^2 + \left( \frac{\eta}{128 t_f(\delta)} + \frac{c_1 L \eta^2 \chi^3 d}{m} \right) \sum_{t \in J_k} \|\nabla f(x_t)\|^2 \right)$$

$$+ \sum_{k \in I_1^c} \left( -\frac{3\eta}{4} \sum_{t \in J_k} \frac{1}{m} \sum_{i=1}^{m} |Z_{t,i}^\top \nabla f(x_t)|^2 + \left( \frac{\eta}{128 t_f(\delta)} + \frac{c_1 L \eta^2 \chi^3 d}{m} \right) \sum_{t \in J_k} \|\nabla f(x_t)\|^2 \right)$$

$$\leq -\sum_{k \in I_1} \frac{\eta}{2} \min_{t \in J_k} \|\nabla f(x_t)\|^2 + \sum_{k \in I_1^c} t_f(\delta) \left( \frac{c_5^2}{64} t_f(\delta)^2 \eta^3 L^2 \left( u^2 d^2 \rho \left( \log \frac{T}{\delta} \right)^2 + \sqrt{1 + \frac{\log(T/\delta)}{d}} r \right)^2 \right)$$

$$\leq -\sum_{k \in I_1} \frac{\eta}{2} \min_{t \in J_k} \|\nabla f(x_t)\|^2 + \tau \frac{c_5^2}{64} t_f(\delta)^2 \eta^3 L^2 \left( u^2 d^2 \rho \left( \log \frac{T}{\delta} \right)^2 + \sqrt{1 + \frac{\log(T/\delta)}{d}} r \right)^2.$$

$$\tag{34}$$

---

[5]To accommodate the last interval which has length at most $2t_f(\delta) - 1$, we note that the results we require for the proof, namely Lemma 14, Lemma 16 and Lemma 17, all hold for any interval length $t'_f \leq 2t_f(\delta)$.

and so by Eq. (3),

$$f(x_\tau) - f(x_0) \le -\sum_{k \in I_1} \frac{\eta}{2} \min_{t \in J_k} \|\nabla f(x_t)\|^2 + \tau \frac{c_5^2}{64} t_f(\delta)^2 \eta^3 L^2 \left( u^2 d^2 \rho \left( \log \frac{T}{\delta} \right)^2 + \sqrt{1 + \frac{\log(T/\delta)}{d}} r \right)^2$$

$$+ \tau \eta u^4 \rho^2 \cdot c_1 d^3 \left( \log \frac{T}{\delta} \right)^3 + \tau L \eta^2 u^4 \rho^2 \cdot c_1 d^4 \left( \log \frac{T}{\delta} \right)^4$$

$$+ \eta c_1 r^2 (\alpha + \eta L) \log \frac{T}{\delta} + \tau c_1 L \eta^2 r^2.$$

Note that we choose $\alpha = 128 t_f(\delta)$. In addition, observe that by our choice of $\delta$ (such that $\delta \le \frac{1}{e}$), it follows that $\sqrt{1 + \frac{\log(T/\delta)}{d}} \le \sqrt{2 \log(T/\delta)}$.

We can now complete our proof by using the union bound (suppressing the dependence of some of the events on $\delta$ for notational simplicity) to derive

$$\mathbb{P}(\mathcal{E}_\tau^c) \le \mathbb{P}(\mathcal{H}_\tau^c) + \sum_{t=0}^{\tau-1} \mathbb{P}(\mathcal{A}_t^c) + \sum_{t=0}^{\tau-1} \mathbb{P}(\mathcal{G}_t^c) + \sum_{k=0}^{K-1} \mathbb{P}(\mathcal{B}_{k t_f(\delta)}^c (\delta; t_f(\delta)))$$

$$\le \frac{(\tau + 4)\delta}{T} + \frac{\tau}{T} \delta + 2 \frac{\tau}{T} \delta + \frac{K\delta}{T} \le \frac{(5\tau + 4)}{T} \delta. \qquad \square$$

We are now ready to show that if sufficiently many iterations have a large gradient, then with high probability, the function value of the last iterate $f(x_T)$, will be less than $\min_x f(x)$, a contradiction. Hence this limits the number of iterations that can have a large gradient.

**Proposition 4.** *Let $c_1 > 0, c_2 \ge 1, c_4 > 0, c_5 > 0, C_1 \ge 1$ be the absolute constants defined in the statements of the previous lemmas, and let $\delta \in (0, 1/e]$ be arbitrary. Suppose we choose $u$, $r$, $\eta$ and $T$ such that*

$$u \le \frac{\sqrt{\epsilon}}{d\sqrt{\rho} \log(T/\delta)} \cdot \min \left\{ \frac{1}{64 c_5^2 c_2}, \frac{1}{2048 c_1 c_2} \right\}^{1/4}, \qquad r \le \epsilon \cdot \min \left\{ \frac{1}{8 c_5 \sqrt{2 c_2}}, \frac{1}{32\sqrt{c_1}} \right\},$$

$$\eta \le \frac{1}{L t_f(\delta)} \min \left\{ \frac{1}{\log(T/\delta)}, \frac{\sqrt{m}}{8 c_4 (\mathrm{lr}(C_1 d m T/\delta))^{3/2} \sqrt{d}}, \frac{m}{128 c_1 (\mathrm{lr}(C_1 d m T/\delta))^3 d} \right\},$$

$$T \ge \max \left\{ \frac{256 t_f(\delta) \left( (f(x_0) - f^*) + \epsilon^2/L \right)}{\eta \epsilon^2}, 4 \right\}.$$

*Then, with probability at least $1 - 6\delta$, there are at most $T/4$ iterations for which $\|\nabla f(x_t)\| \ge \epsilon$.*

*Proof.* Without loss of generality, we assume that $T$ is a multiple of $t_f(\delta)$, and we similarly split $\{0, 1, \ldots, T\}$ into $K = \lfloor T/t_f(\delta) \rfloor$ intervals $J_0, \ldots, J_{K-1}$. Let $I_1$ denote the set of indices $k$ such that for every $t \in J_k$,

$$\|\nabla f(x_t)\| > 8 t_f(\delta) \eta L \left[ \left( \frac{u}{2} \left\| \frac{1}{m} \sum_{i=1}^m Z_{t,i} Z_{t,i}^\top \tilde{H}_{t,i} Z_{t,i} \right\| \right) + \|Y_t\| \right]. \tag{35}$$

We let $I_1^c$ denote the complement of $I_1$ in $\{0, 1, \ldots, K-1\}$. We denote

$$\mathcal{E}_T(\delta) := \mathcal{H}_T(\delta) \cap \left( \bigcap_{t=0}^{T-1} \mathcal{A}_t(\delta) \right) \cap \left( \bigcap_{t=0}^{T-1} \mathcal{G}_t(\delta) \right) \cap \left( \bigcap_{k=0}^{K-1} \mathcal{B}_{k t_f(\delta)}(\delta; t_f(\delta)) \right).$$

In the remaining part of the proof, unless otherwise stated, we shall always assume that we are working on the event $\mathcal{E}_T(\delta)$.

By Lemma 18 with $\tau = T$ and our choices of $\eta$ and $\delta$ in the statement of the lemma, we have

$$f(x_T) - f(x_0) \le -\sum_{k \in I_1} \frac{\eta}{2} \min_{t \in J_k} \|\nabla f(x_t)\|^2 + T \frac{c_5^2}{64} t_f(\delta)^2 \eta^3 L^2 \left( u^2 d^2 \rho \left( \log \frac{T}{\delta} \right)^2 + \sqrt{2 \log(T/\delta)} r \right)^2$$

$$+ T\eta u^4 \rho^2 \cdot c_1 d^3 \left(\log \frac{T}{\delta}\right)^3 + TL\eta^2 u^4 \rho^2 \cdot c_1 d^4 \left(\log \frac{T}{\delta}\right)^4$$

$$+ \eta c_1 r^2 (128 t_f(\delta) + \eta L) \log \frac{T}{\delta} + T c_1 L \eta^2 r^2. \tag{36}$$

Suppose that there are at least $T/4$ iterations where $\|\nabla f(x_t)\| \geq \epsilon$. Let $I_\epsilon$ denote the set of indices $k$ for which there exists some $t \in J_k$ with $\|\nabla f(x_t)\| \geq \epsilon$. Then, by the pigeonhole principle, the set $I_\epsilon$ has at least $\lceil T/(4t_f(\delta)) \rceil$ members. Note that, by our choices of the parameters $\eta, u, r$, it can be shown that

$$c_5 t_f(\delta) \eta L \left(u^2 d^2 \rho \left(\log \frac{T}{\delta}\right)^2 + \sqrt{1 + \frac{\log(T/\delta)}{d}} r\right) < \epsilon, \tag{37}$$

while by Lemma 17, if $k$ is in $I_1^c$, we have

$$\|\nabla f(x_t)\| \leq c_5 t_f(\delta) \eta L \left(u^2 d^2 \rho \log(T/\delta) + \sqrt{1 + \frac{\log(T/\delta)}{d}} r\right), \qquad \forall t \in J_k.$$

This implies that $I_\epsilon \subseteq I_1$.

Observe that by Lemma 16, for any $k \in I_1$, we have

$$\frac{1}{2} \|\nabla f(x_{kt_f(\delta)})\| \leq \|\nabla f(x_t)\| \leq 2 \|\nabla f(x_{kt_f(\delta)})\|, \qquad \forall t \in J_k.$$

This implies in particular that for any $k \in I_\epsilon$, we have $\min_{t \in J_k} \|\nabla f(x_t)\|^2 \geq \frac{1}{16}\epsilon^2$, and consequently

$$-\sum_{k \in I_1} \frac{\eta}{2} \min_{t \in J_k} \|\nabla f(x_t)\|^2 \leq -\sum_{k \in I_\epsilon} \frac{\eta}{2} \cdot \frac{\epsilon^2}{16} \leq -\frac{T}{4t_f(\delta)} \cdot \frac{\eta}{2} \cdot \frac{\epsilon^2}{16} = -\frac{T\eta\epsilon^2}{128 t_f(\delta)}.$$

Hence, by Eq. (36),

$$f(x_T) - f(x_0) \leq -\frac{T\eta\epsilon^2}{128 t_f(\delta)} + T\frac{c_5^2}{64} t_f(\delta)^2 \eta^3 L^2 \left(u^2 d^2 \rho \left(\log \frac{T}{\delta}\right)^2 + \sqrt{2\log(T/\delta)} r\right)^2$$

$$+ T\eta u^4 \rho^2 \cdot c_1 d^3 \left(\log \frac{T}{\delta}\right)^3 + T\eta \cdot (\eta L) u^4 \rho^2 \cdot c_1 d^4 \left(\log \frac{T}{\delta}\right)^4$$

$$+ \eta c_1 r^2 (128 t_f(\delta) + \eta L) \log \frac{T}{\delta} + T\eta \cdot c_1 \eta L r^2. \tag{38}$$

Now, by our choices of $u$, $r$ and $\eta$, we have

$$T\frac{c_5^2}{64} t_f(\delta)^2 \eta^3 L^2 \left(u^2 d^2 \rho \left(\log \frac{T}{\delta}\right)^2 + \sqrt{2\log(T/\delta)} r\right)^2$$

$$\leq T\eta \cdot \frac{c_5^2}{32} t_f(\delta)^2 (\eta L)^2 \left(u^4 d^4 \rho^2 \left(\log \frac{T}{\delta}\right)^4 + 2\log(T/\delta) r^2\right)$$

$$\leq T\eta \cdot \left(\frac{\epsilon^2}{2048 c_2 \left(\log \frac{T}{\delta}\right)^2} + \frac{\epsilon^2}{2048 c_2 \log(T/\delta)}\right) \leq \frac{T\eta\epsilon^2}{512 t_f(\delta)},$$

where we used $\log(T/\delta) \geq 1$ and $2c_2 \log(T/\delta) \geq t_f(\delta)$. We also have

$$T\eta u^4 \rho^2 \cdot c_1 d^3 \left(\log \frac{T}{\delta}\right)^3 + T\eta \cdot (\eta L) u^4 \rho^2 \cdot c_1 d^4 \left(\log \frac{T}{\delta}\right)^4 + T c_1 L \eta^2 r^2$$

$$\leq T\eta \cdot \frac{\epsilon^2}{2048 c_2 d \log(T/\delta)} + T\eta \cdot \frac{\epsilon^2}{2048 c_2 t_f(\delta) \log(T/\delta)} + T\eta \cdot \frac{\epsilon^2}{1024 t_f(\delta) \log(T/\delta)}$$

$$\leq \frac{T\eta\epsilon^2}{512 t_f(\delta)},$$

where we used $c_2 d \log(T/\delta) \geq t_f(\delta)$, $c_2 \geq 1$ and $\log(T/\delta) \geq 1$. Finally,

$$\eta c_1 r^2 (128 t_f(\delta) + \eta L) \log \frac{T}{\delta} \leq \frac{(128 t_f(\delta) + 1)\epsilon^2}{1024 L t_f(\delta)} < \frac{\epsilon^2}{L}.$$

By plugging these bounds into Eq. (38), we get

$$f(x_T) - f(x_0) < -\frac{T\eta\epsilon^2}{128 t_f(\delta)} + \frac{T\eta\epsilon^2}{512 t_f(\delta)} + \frac{T\eta\epsilon^2}{512 t_f(\delta)} + \frac{\epsilon^2}{L} \leq -\frac{T\eta\epsilon^2}{256 t_f(\delta)} + \frac{\epsilon^2}{L}.$$

Therefore, as long as

$$T \geq \frac{256 t_f(\delta) \left( (f(x_0) - f^*) + \epsilon^2/L \right)}{\eta\epsilon^2},$$

we will get $f(x_T) < f^*$, which is a contradiction. Thus, we can conclude that on the event $\mathcal{E}_T(\delta)$, there are at most $T/4$ iterations for which $\|\nabla f(x_t)\| \geq \epsilon$.

We can now complete our proof by using the union bound (suppressing the dependence of some of the events on $\delta$ for notational simplicity) to derive

$$\mathbb{P}(\mathcal{E}_T^c) \leq \mathbb{P}(\mathcal{H}_T^c) + \sum_{t=0}^{T-1} \mathbb{P}(\mathcal{A}_t^c) + \sum_{t=0}^{T-1} \mathbb{P}(\mathcal{G}_t^c) + \sum_{k=0}^{K-1} \mathbb{P}(\mathcal{B}_{k t_f(\delta)}^c(\delta; t_f(\delta)))$$

$$\leq \frac{(T+4)\delta}{T} + \delta + 2\delta + \frac{K\delta}{T} \leq 6\delta. \qquad \square$$

# E ESCAPING SADDLE POINT

In this section, we first show that the travelling distance of the iterates can be bounded in terms of the function value improvement (Appendix E.2). Utilizing this result, as well as Proposition 2 in Appendix C.3 which provides a concentration bound on the the zeroth-order noise, we then prove that sufficient function value decrease can be made near a saddle point in Appendix E.3.

## E.1 KEY QUANTITIES AND NOTATION

We will use $\gamma$ to denote $-\lambda_{\min}(\nabla^2 f(x_0))$, where we know that $\gamma \geq \sqrt{\rho\epsilon}$.

## E.2 IMPROVE OR LOCALIZE

In this subsection, we aim to bound the movement of the iterates across a number of steps in terms of the function value improvement made during these number of steps.

We first state a simple result separating the norm of the difference between $x_{t_0+\tau}$ and $x_{t_0}$ into a few different terms.

**Lemma 19.** *Consider the perturbed zeroth-order update Algorithm 1. Then, for any $t_0 \in \mathbb{N}$ and $\tau \in \mathbb{N}$,*

$$\|x_{t_0+\tau} - x_{t_0}\|^2 \leq V_1(t_0, \tau) + V_2(t_0, \tau) + V_3(t_0, \tau) + V_4(t_0, \tau), \qquad (39)$$

*where*

$$V_1(t_0, \tau) := 8\eta^2 \tau \sum_{t=t_0}^{t_0+\tau-1} \|\nabla f(x_t)\|^2, \quad V_2(t_0, \tau) := 8\eta^2 \left\| \sum_{t=t_0}^{t_0+\tau-1} \frac{1}{m} \sum_{i=1}^m (Z_{t,i} Z_{t,i}^\top - I) \nabla f(x_t) \right\|^2$$

$$V_3(t_0, \tau) := 4\eta^2 \left\| \sum_{t=t_0}^{t_0+\tau-1} Y_t \right\|^2, \qquad V_4(t_0, \tau) := 4\eta^2 \left\| \sum_{t=t_0}^{t_0+\tau-1} \frac{1}{m} \sum_{i=1}^m u Z_{t,i} Z_{t,i}^\top \tilde{H}_{t,i} Z_{t,i} \right\|^2. \qquad (40)$$

*Proof.* For notational convenience, let $t_0 := 0$. Then, applying the form of the perturbed zeroth-order update in Algorithm 1, we get

$$\|x_\tau - x_0\|^2$$

$$
\begin{aligned}
&= \left\| \sum_{t=0}^{\tau-1} x_{t+1} - x_t \right\|^2 \\
&= \eta^2 \left\| \sum_{t=0}^{\tau-1} \frac{1}{m} \sum_{i=1}^{m} Z_{t,i} Z_{t,i}^\top \nabla f(x_t) + \frac{1}{m} \sum_{i=1}^{m} u Z_{t,i} Z_{t,i}^\top \tilde{H}_{t,i} Z_{t,i} + Y_t \right\|^2 \\
&\leq 4\eta^2 \left\| \sum_{t=0}^{\tau-1} \frac{1}{m} \sum_{i=1}^{m} Z_{t,i} Z_{t,i}^\top \nabla f(x_t) \right\|^2 + 4\eta^2 \left\| \sum_{t=0}^{\tau-1} \frac{1}{m} \sum_{i=1}^{m} u Z_{t,i} Z_{t,i}^\top \tilde{H}_{t,i} Z_{t,i} \right\|^2 + 4\eta^2 \left\| \sum_{t=0}^{\tau-1} Y_t \right\|^2 \\
&\leq 4\eta^2 \left\| \sum_{t=0}^{\tau-1} \frac{1}{m} \sum_{i=1}^{m} (Z_{t,i} Z_{t,i}^\top - I) \nabla f(x_t) + \sum_{t=0}^{\tau-1} \nabla f(x_t) \right\|^2 + 4\eta^2 \left\| \sum_{t=0}^{\tau-1} \frac{1}{m} \sum_{i=1}^{m} u Z_{t,i} Z_{t,i}^\top \tilde{H}_{t,i} Z_{t,i} \right\|^2 + 4\eta^2 \left\| \sum_{t=0}^{\tau-1} Y_t \right\|^2 \\
&\leq \underbrace{8\eta^2 \tau \sum_{t=0}^{\tau-1} \|\nabla f(x_t)\|^2}_{V_1(0,\tau)} + \underbrace{8\eta^2 \left\| \sum_{t=0}^{\tau-1} \frac{1}{m} \sum_{i=1}^{m} (Z_{t,i} Z_{t,i}^\top - I) \nabla f(x_t) \right\|^2}_{V_2(0,\tau)} + \underbrace{4\eta^2 \left\| \sum_{t=0}^{\tau-1} Y_t \right\|^2}_{V_3(0,\tau)} + \underbrace{4\eta^2 \left\| \sum_{t=0}^{\tau-1} \frac{1}{m} \sum_{i=1}^{m} u Z_{t,i} Z_{t,i}^\top \tilde{H}_{t,i} Z_{t,i} \right\|^2}_{V_4(0,\tau)}.
\end{aligned}
$$

$\square$

We now proceed to bound the terms $V_1(t_0, \tau)$, $V_2(t_0, \tau)$, $V_3(t_0, \tau)$ and $V_4(t_0, \tau)$.

First, we have the following result bounding $V_1(t_0, \tau)$.

**Lemma 20.** *Let $c_1 > 0, c_2 \geq 1, c_4 > 0, c_5 > 0, C_1 \geq 1$ be the absolute constants defined in the statements of the previous lemmas, and let $\delta \in (0, 1/e]$ be arbitrary.*

*Suppose we choose $\eta$ such that*

$$
\eta \leq \frac{1}{L t_f(\delta)} \cdot \min \left\{ \frac{\sqrt{m}}{8 c_4 (\mathrm{lr}(C_1 dmT/\delta))^{3/2} \sqrt{d}}, \frac{m}{128 c_1 (\mathrm{lr}(C_1 dmT/\delta))^3 d} \right\}.
$$

*There are two cases to consider.*

1. *The first is when $\tau \geq t_f(\delta)$. In this case, split $\{t_0, t_0+1, \ldots, t_0+\tau-1\}$ into $K := \lfloor \tau/t_f(\delta) \rfloor$ intervals:*

$$
\begin{aligned}
J_k &= \{t_0 + k t_f(\delta), \ldots, t_0 + (k+1) t_f(\delta) - 1\}, \quad 0 \leq k < K-1, \\
J_{K-1} &= \{t_0 + (K-1) t_f(\delta), \ldots, t_0 + \tau - 1\}.
\end{aligned}
$$

*Then, on the event*

$$
\mathcal{E}_{t_0,\tau}(\delta) := \mathcal{H}_{t_0,\tau}(\delta) \cap \left( \bigcap_{t=t_0}^{t_0+\tau-1} \mathcal{A}_t(\delta) \right) \cap \left( \bigcap_{t=t_0}^{t_0+\tau-1} \mathcal{G}_t(\delta) \right) \cap \left( \bigcap_{k=0}^{K-2} \mathcal{B}_{t_0+k t_f(\delta)}(\delta; t_f(\delta)) \right) \cap \mathcal{B}_{t_0+(K-1) t_f(\delta)}(\delta; \tau - (K-1) t_f(\delta)),
$$

*we have that*

$$
\begin{aligned}
V_1(t_0, \tau) &= 8\eta^2 \tau \sum_{t=t_0}^{t_0+\tau-1} \|\nabla f(x_t)\|^2 \\
&\leq 64 \eta \tau t_f(\delta) \left( (f(x_0) - f(x_\tau)) + N_{u,r}(\tau; \delta) \right),
\end{aligned}
$$

*where*

$$
\begin{aligned}
N_{u,r}(\tau; \delta) &:= \tau \frac{c_5^2}{64} \eta^3 t_f(\delta)^2 L^2 \left( u^2 d^2 \rho \left( \log \frac{T}{\delta} \right)^2 + \sqrt{2 \log(T/\delta)} r \right)^2 \\
&\quad + \tau \eta u^4 \rho^2 \cdot c_1 d^3 \left( \log \frac{T}{\delta} \right)^3 + \tau L \eta^2 u^4 \rho^2 \cdot c_1 d^4 \left( \log \frac{T}{\delta} \right)^4 \\
&\quad + \eta c_1 r^2 (128 t_f(\delta) + \eta L) \log \frac{T}{\delta} + \tau c_1 L \eta^2 r^2 \\
&\quad + c_5^2 t_f^3(\delta) \eta^3 L^2 \left( u^2 d^2 \rho \log(T/\delta) + \sqrt{2 \log(T/\delta)} r \right)^2. \tag{41}
\end{aligned}
$$

2. *The second is when $\tau < t_f(\delta)$. Suppose we choose $u$ and $r$ such that*

$$u \leq \frac{\sqrt{\epsilon}}{d\sqrt{\rho}\log(T/\delta)} \cdot \min\left\{\frac{1}{64c_5^2c_2}, \frac{1}{2048c_1c_2}\right\}^{1/4}, \qquad r \leq \epsilon \cdot \min\left\{\frac{1}{8c_5\sqrt{2c_2}}, \frac{1}{32\sqrt{c_1}}\right\}.$$

*Suppose the event $\cap_{t=t_0}^{t_0+\tau-1}(\mathcal{A}_t(\delta) \cap \mathcal{G}_t(\delta)$ holds. Suppose also that $\|\nabla f(x_{t_0})\| \leq \epsilon$. Then,*

$$V_1(t_0, \tau) \leq 32\eta^2\tau^2\epsilon^2 \leq 32\eta^2(t_f(\delta))^2\epsilon^2$$

*Proof.* 1. We first consider the case where $\tau \geq t_f(\delta)$. Let $I_1$ denote the set of indices $k$ such that for every time-step $t$ in the interval $J_k$, the gradient dominates the noise terms as

$$\|\nabla f(x_t)\| > 8t_f(\delta)\eta L\left(\frac{u}{2}\left\|\frac{1}{m}\sum_{i=1}^{m}Z_{t,i}Z_{t,i}^\top\tilde{H}_{t,i}Z_{t,i}\right\| + \|Y_t\|\right). \tag{42}$$

WLOG, we may assume that $t_0 := 0$, and denote $V_1(\tau) := V_1(0, \tau)$. WLOG, we also assume that $\tau$ is a multiple of $t_f(\delta)$. From Lemma 18, on the event that $\mathcal{E}_\tau(\delta)$ holds and by our choice of $\eta$, we have

$$f(x_\tau) - f(x_0) \leq -\sum_{k\in I_1}\frac{\eta}{2}\min_{t\in J_k}\|\nabla f(x_t)\|^2 + \tau\frac{c_5^2}{64}\eta^3t_f(\delta)^2L^2\left(u^2d^2\rho\left(\log\frac{T}{\delta}\right)^2 + \sqrt{2\log(T/\delta)}r\right)^2$$

$$+ \tau\eta u^4\rho^2 \cdot c_1d^3\left(\log\frac{T}{\delta}\right)^3 + \tau L\eta^2u^4\rho^2 \cdot c_1d^4\left(\log\frac{T}{\delta}\right)^4$$

$$+ \eta c_1r^2(128t_f(\delta) + \eta L)\log\frac{T}{\delta} + \tau c_1L\eta^2r^2.$$

By Lemma 16 (and our choice of $\eta$), it follows that for any $k \in I_1$, on the event $\cap_{t\in J_k}\mathcal{A}_t(\delta)$, we have

$$\sum_{t\in J_k}\|\nabla f(x_t)\|^2 \leq 4t_f\min_{t\in J_k}\|\nabla f(x_t)\|^2.$$

Thus, on the event that $\mathcal{E}_\tau(\delta)$ holds, for our choice of $\eta$, we have

$$\eta\sum_{k\in I_1}\sum_{t\in J_k}\|\nabla f(x_t)\|^2 \leq 4t_f(\delta)\eta\sum_{k\in I_1}\min_{t\in J_k}\|\nabla f(x_t)\|^2$$

$$\leq 8t_f(\delta)\sum_{k\in I_1}\frac{\eta}{2}\min_{t\in J_k}\|\nabla f(x_t)\|^2$$

$$\leq 8t_f(\delta)\left((f(x_0) - f(x_\tau)) + \tau\frac{c_5^2}{64}\eta^3t_f(\delta)^2L^2\left(u^2d^2\rho\left(\log\frac{T}{\delta}\right)^2 + \sqrt{2\log(T/\delta)}r\right)^2\right)$$

$$+ 8t_f(\delta)\left(\tau\eta u^4\rho^2 \cdot c_1d^3\left(\log\frac{T}{\delta}\right)^3 + \tau L\eta^2u^4\rho^2 \cdot c_1d^4\left(\log\frac{T}{\delta}\right)^4\right)$$

$$+ 8t_f(\delta)\left(\eta c_1r^2(128t_f(\delta) + \eta L)\log\frac{T}{\delta} + \tau c_1L\eta^2r^2\right).$$

Similarly, for any $k \in I_1^c$ (where $I_1^c$ denotes the complement of $I_1$ in $\{0, 1, \ldots, K-1\}$, i.e. intervals where the gradient is smaller than than the perturbation terms in some iteration), on the event $(\cap_{t\in J_k}\mathcal{A}_t(\delta)) \cap (\cap_{t\in J_k}\mathcal{G}_t(\delta))$, by Lemma 17 (and our choice of $\eta$), we have

$$\|\nabla f(x_t)\| \leq c_5t_f(\delta)\eta L\left(u^2d^2\rho\log(T/\delta) + \sqrt{2\log(T/\delta)}r\right), \qquad \forall t \in J_k.$$

On the event that $\mathcal{E}_\tau(\delta)$ holds, this gives us then

$$\eta\sum_{k\in I_1^c}\sum_{t\in J_k}\|\nabla f(x_t)\|^2 \leq \eta\tau\left(c_5^2t_f^2(\delta)\eta^2L^2\left(u^2d^2\rho\log(T/\delta) + \sqrt{2\log(T/\delta)}r\right)^2\right).$$

Hence, on the event that $\mathcal{E}_\tau(\delta)$ holds, we have that

$$\eta \sum_{t=0}^{\tau-1} \|\nabla f(x_t)\|^2 = \eta \sum_{k\in I_1} \sum_{t\in J_k} \|\nabla f(x_t)\|^2 + \eta \sum_{k\in I_1^c} \sum_{t\in J_k} \|\nabla f(x_t)\|^2$$

$$\leq 8t_f(\delta)\left( (f(x_0) - f(x_\tau)) + \tau\frac{c_5^2}{64}\eta^3 t_f(\delta)^2 L^2 \left(u^2 d^2 \rho \left(\log\frac{T}{\delta}\right)^2 + \sqrt{2\log(T/\delta)}r\right)^2\right)$$

$$+ \ 8t_f(\delta)\left(\tau\eta u^4\rho^2 \cdot c_1 d^3 \left(\log\frac{T}{\delta}\right)^3 + \tau L\eta^2 u^4 \rho^2 \cdot c_1 d^4 \left(\log\frac{T}{\delta}\right)^4\right)$$

$$+ \ 8t_f(\delta)\left(\eta c_1 r^2(128 t_f(\delta) + \eta L)\log\frac{T}{\delta} + \tau c_1 L\eta^2 r^2\right)$$

$$+ \ 8t_f(\delta)\eta\tau\left(c_5^2 t_f^2(\delta)\eta^2 L^2 \left(u^2 d^2 \rho \log(T/\delta) + \sqrt{2\log(T/\delta)}r\right)^2\right).$$

This yields the final result for the case $\tau \geq t_f(\delta)$.

2. We next consider the case where $1 \leq \tau < t_f(\delta)$. Recall the notation that

$$\mathcal{E}(t_0, t_0 + \tau, \delta) := \cap_{t=t_0}^{t_0+\tau-1}\left\{\|\nabla f(x_t)\| > 8t_f(\delta)\eta L\left(\frac{u}{2}\left\|\frac{1}{m}\sum_{i=1}^m Z_{t,i}Z_{t,i}^\top \tilde{H}_{t,i} Z_{t,i}\right\| + \|Y_t\|\right)\right\}$$

There are two cases to consider.

(a) On the event $\mathcal{E}(t_0, t_0 + \tau, \delta) \cap \left(\cap_{t=t_0}^{t_0+\tau-1}\mathcal{A}_t(\delta)\right)$, we have by Lemma 16 that $\|\nabla f(x_t)\| \leq 2\|\nabla f(x_0)\|$ for each $t \in \{0, 1, \ldots, \tau-1\}$. Then,

$$V_1(t_0, \tau) = 8\eta^2\tau \sum_{t=t_0}^{t_0+\tau-1} \|\nabla f(x_t)\|^2 \leq 8\eta^2\tau^2\left(4\|\nabla f(x_0)\|^2\right) \leq 32\eta^2\tau^2\epsilon^2,$$

where the final inequality uses the assumption that $\|\nabla f(x_0)\| \leq \epsilon$.

(b) Suppose the event $\mathcal{E}^c(t_0, t_0 + \tau, \delta) \cap \left(\cap_{t=t_0}^{t_0+\tau-1}\mathcal{A}_t(\delta)\right) \cap \left(\cap_{t=t_0}^{t_0+\tau-1}\mathcal{G}_t(\delta)\right)$ holds. In this case, by Lemma 17, we have that for each $t \in \{t_0, t_0 + 1, \ldots, t_0 + \tau - 1\}$

$$\|\nabla f(x_t)\| \leq c_5 t_f(\delta)\eta L\left(u^2 d^2 \rho\left(\log\frac{T}{\delta}\right)^2 + \sqrt{1 + \frac{\log(T/\delta)}{d}}r\right)$$

$$\leq \epsilon,$$

where the final inequality follows by our choice of $\eta, u$ and $r$ (cf. Eq. (37)). Hence,

$$V_1(t_0, \tau) = 8\eta^2\tau \sum_{t=t_0}^{t_0+\tau-1} \|\nabla f(x_t)\|^2$$

$$\leq 8\eta^2\tau^2\left(c_5 t_f(\delta)\eta L\left(u^2 d^2 \rho\left(\log\frac{T}{\delta}\right)^2 + \sqrt{1 + \frac{\log(T/\delta)}{d}}r\right)\right)^2$$

$$\leq 8\eta^2\tau^2\epsilon^2 < 32\eta^2\tau^2\epsilon^2.$$

The final result for the case $\tau < t_f(\delta)$ then follows.

$\square$

We proceed to bound $V_2(t_0, \tau)$.

**Lemma 21.** *Let $c_1 > 0, c_2 \geq 1, c_4 > 0, c_5 > 0, C_1 \geq 1$ be the absolute constants defined in the statements of the previous lemmas, and let $\delta \in (0, 1/e]$ be arbitrary and $\tau > 0$ be arbitrary. Suppose we choose $\eta$ such that*

$$\eta \leq \frac{1}{Lt_f(\delta)} \cdot \min\left\{\frac{\sqrt{m}}{8c_4(\mathrm{lr}(C_1 dmT/\delta))^{3/2}\sqrt{d}}, \frac{m}{128c_1(\mathrm{lr}(C_1 dmT/\delta))^3 d}\right\}.$$

*Let $T_s$ denote an integer such that $T_s \geq \max\{\tau, t_f(\delta)\}$, and for any $F > 0$, define*

$$B(\delta; F) := \frac{8t_f(\delta)(F + N_{u,r}(T_s, \delta))}{\eta}\left(T_s + \frac{d}{m}\right)(\mathrm{lr}(CT^2/\delta))^2, \qquad b_\tau(\delta; F) := \frac{t_f(\delta)\tau F}{\eta}.$$

*Let $c', C > 0$ denote the same constants as in the statement of Proposition 2. Denote the event that*

*either* $\displaystyle\sum_{t=t_0}^{t_0+\tau-1} \frac{d}{m}(\mathrm{lr}(CT^2/\delta))^2\|\nabla f(x_t)\|^2 \geq B(\delta; F)$

*or* $\displaystyle\sqrt{\frac{V_2(t_0, \tau)}{8\eta^2}} \leq c'\sqrt{\max\left\{\sum_{t=t_0}^{t_0+\tau-1}\frac{d}{m}(\mathrm{lr}(CT^2/\delta))^2\|\nabla f(x_t)\|^2, b_\tau(\delta; F)\right\}\left(\log\left(\frac{CT^2}{\delta}\right) + \log\left(\log\left(\frac{B(\delta; F)}{b_\tau(\delta; F)}\right) + 1\right)\right)}$

*holds as $\mathcal{L}_{t_0,\tau}(\delta; F)$.[6] We show that $\mathbb{P}(\mathcal{L}_{t_0,\tau}(\delta; F)) \geq 1 - \frac{\delta}{T}$. Finally, denote the event $\mathcal{M}_{t_0,T_s}(F)$ as the event that $f(x_{t_0}) - f(x_{t_0+T_s}) < F$.*

*Then, on the event $\mathcal{L}_{t_0,\tau}(\delta) \cap \mathcal{E}_{t_0,T_s}(\delta) \cap \mathcal{M}_{t_0,T_s}(F)$ (where $\mathcal{E}_{0,T_s}(\delta)$ is as defined in Lemma 20),*

$$V_2(t_0, \tau) \leq 8c'^2\beta_1(\delta; F)\eta t_f(\delta)\max\left\{\frac{8d}{m}(\mathrm{lr}(CT^2/\delta))^2(F + N_{u,r}(T_s, \delta)), \tau F\right\}, \qquad (43)$$

*where*

$$\beta_1(\delta; F) := \log\left(\frac{CT^2}{\delta}\right) + \log\left(\log\left(\frac{B(\delta; F)}{b_1(\delta; F)}\right) + 1\right).$$

*Proof.* We note that $\mathbb{P}(\mathcal{L}_{t_0,\tau}(\delta; F)) \geq 1 - \frac{\delta}{T}$. is a direct consequence of Proposition 2. In the rest of the proof, without loss of generality, we assume that $t_0 = 0$ for notational simplicity. On the event $\mathcal{L}_{0,\tau}(\delta; F) \cap \mathcal{E}_{0,T_s}(\delta) \cap \mathcal{M}_{t_0,T_s}(F)$, suppose that

$$\sum_{t=0}^{\tau-1}\frac{d}{m}(\mathrm{lr}(CT^2/\delta))^2\|\nabla f(x_t)\|^2 \geq B(\delta; F) = \frac{8t_f(\delta)(F + N_{u,r}(T_s, \delta))}{\eta}\left(T_s + \frac{d}{m}\right)(\mathrm{lr}(CT^2/\delta))^2$$

$$\implies \eta\sum_{t=0}^{\tau-1}\|\nabla f(x_t)\|^2 \geq 8t_f(\delta)(F + N_{u,r}(T_s, \delta))$$

$$\implies \eta\sum_{t=0}^{T_s-1}\|\nabla f(x_t)\|^2 \geq 8t_f(\delta)(F + N_{u,r}(T_s, \delta))$$

$$\implies 8\eta^2 T_s\sum_{t=0}^{T_s-1}\|\nabla f(x_t)\|^2 \geq 64\eta T_s t_f(\delta)(F + N_{u,r}(T_s, \delta))$$

$$\implies 8\eta^2 T_s\sum_{t=0}^{T_s-1}\|\nabla f(x_t)\|^2 \geq 64\eta T_s t_f(\delta)(f(x_0) - f(x_{T_s}) + N_{u,r}(T_s, \delta)), \quad \text{since } f(x_0) - f(x_{T_s}) \leq F$$

$$\iff V_1(0, T_s) \geq 64\eta T_s t_f(\delta)(f(x_0) - f(x_{T_s}) + N_{u,r}(T_s, \delta)),$$

where we note the last equation contradicts Lemma 20. For notational simplicity, denote

$$\beta_\tau(\delta; F) := \log\left(\frac{CT^2}{\delta}\right) + \log\left(\log\left(\frac{B(\delta; F)}{b_\tau(\delta; F)}\right) + 1\right).$$

Observe that $\beta_1$ is larger than $\beta_\tau$ for every $\tau \geq 1$. Since $\mathcal{L}_{t_0,\tau}(\delta; F)$ holds, we must have then that

$$\sqrt{\frac{V_2(0, \tau)}{8\eta^2}} \leq c'\sqrt{\max\left\{\sum_{t=0}^{\tau-1}\frac{d}{m}(\mathrm{lr}(CT^2/\delta))^2\|\nabla f(x_t)\|^2, b_\tau(\delta; F)\right\}\beta_1(\delta; F)}.$$

---

[6]We note that by construction, $B(\delta; F) \geq b_\tau(\delta; F)$

Now, continuing, recalling the definition of $V_1(0, T_s) = 8\eta^2 T_s \sum_{t=0}^{T_s-1} \|\nabla f(x_t)\|^2$

$$V_2(0, \tau) \leq c'^2 \beta_1(\delta; F) \max\left\{8\eta^2 \sum_{t=0}^{\tau-1} \frac{d}{m}(\mathrm{lr}(CT^2/\delta))^2 \|\nabla f(x_t)\|^2, 8\eta^2 b_\tau(\delta; F)\right\}$$

$$\leq c'^2 \beta_1(\delta; F) \max\left\{8\eta^2 \sum_{t=0}^{T_s-1} \frac{d}{m}(\mathrm{lr}(CT^2/\delta))^2 \|\nabla f(x_t)\|^2, 8\eta^2 b_\tau(\delta; F)\right\}$$

$$\leq c'^2 \beta_1(\delta; F) \max\left\{\frac{d}{m}(\mathrm{lr}(CT^2/\delta))^2 \frac{V_1(0, T_s)}{T_s}, 8\eta t_f(\delta)\tau F\right\}$$

$$\overset{(i)}{\leq} c'^2 \beta_1(\delta; F) \max\left\{\frac{d}{m}(\mathrm{lr}(CT^2/\delta))^2 \left(64\eta t_f(\delta)(f(x_0) - f(x_{T_s}) + N_{u,r}(T_s, \delta))\right), 8\eta t_f(\delta)\tau F\right\}$$

$$\overset{(ii)}{\leq} c'^2 \beta_1(\delta; F) \max\left\{\frac{d}{m}(\mathrm{lr}(CT^2/\delta))^2 \left(64\eta t_f(\delta)(F + N_{u,r}(T_s, \delta))\right), 8\eta t_f(\delta)\tau F\right\}$$

$$= c'^2 \beta_1(\delta; F)(8\eta t_f(\delta)) \max\left\{\frac{d}{m}(\mathrm{lr}(CT^2/\delta))^2 \left(8(F + N_{u,r}(T_s, \delta))\right), \tau F\right\}.$$

We note that (i) is a consequence of Lemma 20, while (ii) comes from our assumption that the event $\mathcal{M}_{t_0, T_s}(F)$ holds, i.e. $f(x_{t_0}) - f(x_{t_0+T_s}) \leq F$.

$\square$

We next bound $V_3(t_0, \tau)$ and $V_4(t_0, \tau)$.

**Lemma 22.** *Let $c > 0$ denote the same constant in Lemma 7. Consider any arbitrary $0 < \delta \leq 1/e$, and let $\tau \geq t_f(\delta)$ be arbitrary. Let $\mathcal{N}_{t_0, \tau}(\delta)$ denote the event that*

$$V_3(t_0, \tau) := 4\eta^2 \left\|\sum_{t=t_0}^{t_0+\tau-1} Y_t\right\|^2 \leq 4c_6\eta^2\tau \log(2dT/\delta)r^2,$$

*where $c_6 > 0$ is an absolute constant. Then, by Lemma 7, $\mathbb{P}(\mathcal{N}_{t_0, \tau}(\delta)) \geq 1 - \frac{\delta}{T}$. Denote the event*

$$\mathcal{O}_t(\delta) := \left\{\frac{1}{m}\sum_{i=1}^{m} \|Z_{t,i}\|^8 \leq c_7 d^4 \left(\log\left(\frac{T}{\delta}\right)\right)^4\right\},$$

*where $c_7 > 0$ is an absolute constant. Then, on the event $\cap_{t=t_0}^{t_0+\tau-1} \mathcal{O}_t(\delta)$, we have*

$$V_4(t_0, \tau) \leq 4c_7\eta^2\tau^2\rho^2 u^4 d^4 \left(\log\left(\frac{T}{\delta}\right)\right)^4.$$

*Moreover, for each $t$, $\mathbb{P}(\mathcal{O}_t(\delta)) \geq 1 - \frac{\delta}{T}$.*

*Proof.* The proof for $V_3(t_0, \tau)$ follows directly from Lemma 7, by picking $c_6$ to be the $c$ that appears in the statement of Lemma 7. Meanwhile, observe that

$$V_4(t_0, \tau) = 4\eta^2 \left\|\sum_{t=0}^{\tau-1} \frac{1}{m}\sum_{i=1}^{m} u Z_{t,i} Z_{t,i}^\top \tilde{H}_{t,i} Z_{t,i}\right\|^2$$

$$\leq 4\eta^2\tau \left(\sum_{t=t_0}^{t_0+\tau-1} \left\|\frac{1}{m}\sum_{i=1}^{m} u Z_{t,i} Z_{t,i}^\top \tilde{H}_{t,i} Z_{t,i}\right\|^2\right)$$

$$\overset{(iii)}{\leq} 4\eta^2\tau \sum_{t=t_0}^{t_0+\tau-1} \frac{1}{m}\sum_{i=1}^{m} \rho^2 u^4 \|Z_{t,i}\|^8$$

$$\leq 4c_7\eta^2\tau^2\rho^2 u^4 d^4 \left(\log\left(\frac{T}{\delta}\right)\right)^4.$$

Above, to derive (iii), we used the bound that $\left\|\tilde{H}_{t,i}\right\| \leq \rho u\|Z_{t,i}\|$. The final inequality is a consequence of our assumption that $\cap_{t=t_0}^{t_0+\tau-1}\mathcal{O}_t(\delta)$ holds. Finally, the result that $\mathbb{P}(\mathcal{O}_t(\delta)) \geq 1 - \frac{\delta}{T}$ holds due to Lemma 11, where we note that we may pick the absolute constant $c_7$ to be equal to $2Cc^4$, where $c, C > 0$ are the absolute constants that appear in the statement of Lemma 11. $\quad\square$

Finally, combining the earlier results, we have the following technical result, which bounds the travelling distance of the iterates in terms of the decrease in function value decrease.

**Lemma 23** (Improve or Localize). *Consider the perturbed zeroth-order update Algorithm 1. Let $c' > 0, c_1 > 0, c_2 \geq 1, c_4 > 0, c_5 > 0, c_6 > 0, c_7 > 0, C_1 \geq 1$ be the absolute constants defined in the statements of the previous lemmas, and let $\delta \in (0, 1/e]$ be arbitrary. Consider any $T_s \geq t_f(\delta)$. For any $F > 0$, suppose $f(x_{T_s}) - f(x_0) > -F$, i.e. $f(x_0) - f(x_{T_s}) < F$. Suppose that the event*

$$\mathcal{P}_{t_0,T_s}(\delta, F) := \cap_{\tau=1}^{T_s} \left(\mathcal{L}_{t_0,\tau}(\delta; F) \cap \mathcal{N}_{t_0,\tau}(\delta)\right) \cap \left(\cap_{t=t_0}^{t_0+T_s-1}\mathcal{O}_t(\delta) \cap \mathcal{A}_t(\delta) \cap \mathcal{G}_t(\delta)\right) \cap \left(\cap_{\tau=t_f(\delta)}^{T_s-1}\mathcal{E}_{t_0,\tau}(\delta)\right)$$

*holds, where the events $\mathcal{E}_{t_0,\tau}(\delta), \mathcal{L}_{t_0,\tau}(\delta), \mathcal{N}_{t_0,\tau}(\delta), \mathcal{O}_t(\delta)$ are as defined in Lemma 20, Lemma 21 and Lemma 22, and $\mathcal{G}_t(\delta)$ and $\mathcal{A}_t(\delta)$ are as defined in Lemma 13 and Lemma 15. Suppose we choose $u$, $r$ and $\eta$ such that*

$$u \leq \frac{\sqrt{\epsilon}}{d\sqrt{\rho}\log(T/\delta)} \cdot \min\left\{\frac{1}{64c_5^2c_2}, \frac{1}{2048c_1c_2}\right\}^{1/4}, \qquad r \leq \epsilon \cdot \min\left\{\frac{1}{8c_5\sqrt{2c_2}}, \frac{1}{32\sqrt{c_1}}\right\},$$

$$\eta \leq \frac{1}{Lt_f(\delta)}\min\left\{\frac{1}{\log(T/\delta)}, \frac{\sqrt{m}}{8c_4(\mathrm{lr}(C_1dmT/\delta))^{3/2}\sqrt{d}}, \frac{m}{128c_1(\mathrm{lr}(C_1dmT/\delta))^3d}\right\}.$$

*Suppose $\eta \leq \min\left\{1, \frac{1}{t_f(\delta)}, \frac{1}{t_f\delta L}\right\}$. Suppose also we pick $u$ and $r$ small enough such that*

$$u \leq \frac{r^{1/2}}{d\log(T/\delta)\rho^{1/2}}, \qquad r^2 \leq \min\left\{\frac{F}{\eta T_s\log(T/\delta)\left(\frac{65c_5^2}{8} + 132c_1 + 1\right)}, \frac{F}{4c_6\log(2dT/\delta) + 4c_7\eta T_s}\right\}.$$

*Then, for each $\tau \in \{0, 1, \ldots, T_s\}$, we have that*

$$\|x_{t_0+\tau} - x_{t_0}\|^2 \leq \phi_{T_s}(\delta, F),$$

*where*

$$\phi_{T_s}(\delta, F) \leq \max\left\{128\eta T_st_f(\delta)F, 32\eta^2(t_f(\delta))^2\epsilon^2\right\} + 8c'^2\beta_1(\delta; F)\eta t_f(\delta)\max\left\{\frac{16d}{m}(\mathrm{lr}(CT^2/\delta))^2F, T_sF\right\} + T_s\eta t_f(\delta)F,$$

*where $\beta_1(\delta; F)$ is defined as in Lemma 21. Moreover, $\mathbb{P}(\mathcal{P}_{t_0,T_s}(\delta, F)) \geq 1 - \frac{12T_s\delta}{T}$.*

*Proof.* We recall that

$$\|x_{t_0+\tau} - x_{t_0}\|^2 \leq \underbrace{8\eta^2\tau\sum_{t=t_0}^{t_0+\tau-1}\|\nabla f(x_t)\|^2}_{V_1(t_0,\tau)} + \underbrace{8\eta^2\left\|\sum_{t=t_0}^{t_0+\tau-1}\frac{1}{m}\sum_{i=1}^m(Z_{t,i}Z_{t,i}^\top - I)\nabla f(x_t)\right\|^2}_{V_2(t_0,\tau)}$$

$$+ \underbrace{4\eta^2\left\|\sum_{t=t_0}^{t_0+\tau-1}Y_t\right\|^2}_{V_3(t_0,\tau)} + \underbrace{4\eta^2\left\|\sum_{t=t_0}^{t_0+\tau-1}\frac{1}{m}\sum_{i=1}^m uZ_{t,i}Z_{t,i}^\top\tilde{H}_{t,i}Z_{t,i}\right\|^2}_{V_4(t_0,\tau)}$$

By Lemma 20, Lemma 21, and Lemma 22, which bound $V_1(t_0, \tau), V_2(t_0, \tau)$, and $V_3(t_0, \tau), V_4(t_0, \tau)$ respectively, on the event $\mathcal{P}_{t_0,T_s}(\delta, F)$, we have, for any $0 \leq \tau \leq T_s$,

$$\|x_\tau - x_0\|^2 \leq V_1(0, \tau) + V_2(0, \tau) + V_3(0, \tau) + V_4(0, \tau)$$
$$\leq \max\left\{64\eta\tau t_f(\delta)(F + N_{u,r}(\tau; \delta)), 32\eta^2(t_f(\delta))^2\epsilon^2\right\}$$

$$+ 8c'^2\beta_1(\delta; F)\eta t_f(\delta) \max\left\{\frac{8d}{m}(\mathrm{lr}(CT^2/\delta))^2\left(F + N_{u,r}(T_s, \delta)\right), \tau F\right\}$$

$$+ 4c_6\eta^2\tau\log(2dT/\delta)r^2 + 4c_7\eta^2\tau^2\rho^2 u^4 d^4\left(\log(T/\delta)\right)^4,$$

where $N_{u,r}(\tau; \delta)$ is defined as in Lemma 20.

For the simplified bound (which does not contain $N_{u,r}(\tau; \delta)$), it remains for us to show that our choice of $u$ and $r$ ensures that $N_{u,r}(T_s, \delta) \leq F$ and

$$4c_6\eta^2 T_s\log(2dT/\delta)r^2 + 4c_7\eta^2 T_s^2\rho^2 u^4 d^4\left(\log(T/\delta)\right)^4 \leq \eta T_s t_f(\delta)F.$$

First, our choice of $u$ ensures that

$$u^4 d^4\rho^2(\log(T/\delta))^4 \leq r^2.$$

Next, recall that

$$N_{u,r}(\tau; \delta) := \tau\frac{c_5^2}{64}\eta^3 t_f(\delta)^2 L^2\left(u^2 d^2\rho\left(\log\frac{T}{\delta}\right)^2 + \sqrt{2\log(T/\delta)}r\right)^2$$

$$+ \tau\eta u^4\rho^2\cdot c_1 d^3\left(\log\frac{T}{\delta}\right)^3 + \tau L\eta^2 u^4\rho^2\cdot c_1 d^4\left(\log\frac{T}{\delta}\right)^4$$

$$+ \eta c_1 r^2(128 t_f(\delta) + \eta L)\log\frac{T}{\delta} + \tau c_1 L\eta^2 r^2$$

$$+ c_5^2 t_f^3(\delta)\eta^3 L^2\left(u^2 d^2\rho\log(T/\delta) + \sqrt{2\log(T/\delta)}r\right)^2.$$

Recalling our choice of $\eta$ such that

$$\eta \leq \min\{1, \frac{1}{t_f(\delta)}, \frac{1}{t_f(\delta)L}\},$$

it follows that

$$N_{u,r}(T_s; \delta) \leq \eta T_s r^2\left(\frac{8c_5^2}{64}\log(T/\delta) + 2c_1 + 2c_1 + (128c_1 + 1)\log(T/\delta) + c_1 + 8c_5^2\log(T/\delta)\right)$$

$$\leq \eta T_s r^2\log(T/\delta)\left(\frac{65c_5^2}{8} + 132c_1 + 1\right) \leq F,$$

where the last inequality follows choosing $r$ such that $r^2 \leq \dfrac{F}{\eta T_s\log(T/\delta)\left(\frac{65c_5^2}{8} + 132c_1 + 1\right)}$. Similarly, we have

$$4c_6\eta^2 T_s\log(2dT/\delta)r^2 + 4c_7\eta^2 T_s^2\rho^2 u^4 d^4\left(\log(T/\delta)\right)^4$$

$$\leq \eta T_s t_f(\delta)\left(4c_6\eta\log(2dT/\delta)r^2 + 4c_7\eta T_s\rho^2 u^4 d^4(\log(T/\delta))^4\right)$$

$$\leq \eta T_s t_f(\delta)\left(4c_6\eta\log(2dT/\delta)r^2 + 4c_7\eta T_s r^2\right)$$

By choosing $r$ such that

$$r^2 \leq \frac{F}{4c_6\log(2dT/\delta) + 4c_7\eta T_s},$$

it follows that

$$4c_6\eta^2 T_s\log(2dT/\delta)r^2 + 4c_7\eta^2 T_s^2\rho^2 u^4 d^4\left(\log(T/\delta)\right)^4 \leq \eta T_s t_f(\delta)F,$$

as desired.

We next lower bound the probability of

$$\mathcal{P}_{t_0, T_s}(\delta, F) := \cap_{\tau=1}^{T_s}\left(\mathcal{L}_{t_0,\tau}(\delta; F) \cap \mathcal{N}_{t_0,\tau}(\delta)\right) \cap\left(\cap_{t=t_0}^{t_0+T_s-1}\mathcal{O}_t(\delta) \cap \mathcal{A}_t(\delta) \cap \mathcal{G}_t(\delta)\right) \cap\left(\cap_{\tau=t_f(\delta)}^{T_s}\mathcal{E}_{t_0,\tau}(\delta)\right).$$

Observe that

$$\cap_{\tau=t_f(\delta)}^{T_s}\mathcal{E}_{t_0,\tau}(\delta)$$

$$= \cap_{\tau=t_f(\delta)}^{T_s} \left( \mathcal{H}_{t_0,\tau}(\delta) \cap \left( \bigcap_{t=t_0}^{t_0+\tau-1} \mathcal{A}_t(\delta) \cap \mathcal{G}_t(\delta) \right) \cap \left( \bigcap_{k=0}^{K-2} \mathcal{B}_{t_0+kt_f(\delta)}(\delta; t_f(\delta)) \right) \cap \mathcal{B}_{t_0+(K-1)t_f(\delta)}(\delta; \tau-(K-1)t_f(\delta)) \right)$$

$$= \cap_{\tau=t_f(\delta)}^{T_s} \left( \mathcal{H}_{t_0,\tau}(\delta) \cap \left( \bigcap_{k=0}^{K-2} \mathcal{B}_{t_0+kt_f(\delta)}(\delta; t_f(\delta)) \right) \cap \mathcal{B}_{t_0+(K-1)t_f(\delta)}(\delta; \tau-(K-1)t_f(\delta)) \right) \cap \left( \bigcap_{t=t_0}^{T_s-1} \mathcal{A}_t(\delta) \cap \mathcal{G}_t(\delta) \right).$$

Note this implies that $\cap_{\tau=t_f(\delta)}^{T_s} \mathcal{E}_{t_0,\tau}(\delta) \cap \left( \cap_{t=t_0}^{T_s-1} \mathcal{A}_t(\delta) \cap \mathcal{G}_t(\delta) \right) = \cap_{\tau=t_f(\delta)}^{T_s} \mathcal{E}_{t_0,\tau}(\delta)$ We note that by Lemma 1,

$$\mathbb{P} \left( \left( \cap_{\tau=t_f(\delta)}^{T_s} \mathcal{H}_{t_0,\tau}(\delta) \right)^c \right) \leq \frac{5T_s\delta}{T}.$$

Meanwhile, we note that

$$\cap_{t=t_0}^{T_s-1} \mathcal{B}_t(\delta; t_f(\delta)) \subseteq \cap_{\tau=t_f(\delta)}^{T_s} \left( \left( \bigcap_{k=0}^{K-2} \mathcal{B}_{t_0+kt_f(\delta)}(\delta; t_f(\delta)) \right) \cap \mathcal{B}_{t_0+(K-1)t_f(\delta)}(\delta; \tau-(K-1)t_f(\delta)) \right).$$

Hence, by Lemma 14, we have that

$$\mathbb{P} \left( \left( \cap_{\tau=t_f(\delta)}^{T_s} \left( \left( \bigcap_{k=0}^{K-2} \mathcal{B}_{t_0+kt_f(\delta)}(\delta; t_f(\delta)) \right) \cap \mathcal{B}_{t_0+(K-1)t_f(\delta)}(\delta; \tau-(K-1)t_f(\delta)) \right) \right)^c \right)$$
$$\leq \mathbb{P} \left( \left( \cap_{t=t_0}^{T_s-1} \mathcal{B}_t(\delta; t_f(\delta)) \right)^c \right) \leq \frac{T_s\delta}{T}.$$

Meanwhile, by Lemma 13 and Lemma 15, we may bound

$$\mathbb{P} \left( \left( \bigcap_{t=t_0}^{T_s-1} \mathcal{A}_t(\delta) \cap \mathcal{G}_t(\delta) \right)^c \right) \leq \frac{T_s\delta}{T} + \frac{2T_s\delta}{T} = \frac{3T_s\delta}{T}.$$

Hence, it follows that

$$\mathbb{P} \left( \left( \cap_{\tau=t_f(\delta)}^{T_s} \mathcal{E}_{t_0,\tau}(\delta) \cap \left( \cap_{t=t_0}^{T_s-1} \mathcal{A}_t(\delta) \cap \mathcal{G}_t(\delta) \right) \right)^c \right) \leq \frac{5T_s\delta}{T} + \frac{T_s\delta}{T} + \frac{3T_s\delta}{T} = \frac{9T_s\delta}{T}.$$

Meanwhile, it follows from our results in the preceding lemmas that

$$\mathbb{P} \left( \left( \cap_{\tau=1}^{T_s} \left( \mathcal{L}_{t_0,\tau}(\delta; F) \cap \mathcal{N}_{t_0,\tau}(\delta) \right) \cap \left( \cap_{t=t_0}^{T_s-1} \mathcal{O}_t(\delta) \right) \right)^c \right) \leq \frac{3T_s\delta}{T}.$$

Hence, it follows that $\mathbb{P}(\mathcal{P}_{t_0,T_s}(\delta, F)) \geq 1 - \frac{12T_s\delta}{T}$.

$\square$

### E.3 PROVING FUNCTION VALUE DECREASE NEAR SADDLE POINT

We next build on the technical result earlier to prove that each time we are near the saddle point, there is a constant probability of making significant function value decrease. We briefly provide a high-level proof outline below. In our proof, we introduce a coupling argument connecting two closely-related sequences both starting from the saddle, differing only in the sign of their perturbative term along the minimum eigendirection of the Hessian at the saddle. Specifically, when function decrease from a saddle is not sufficiently large, due to the earlier technical result, we know that the coupled sequences will remain within a radius $\phi$ of the original saddle for a large number (which we will denote as $T_s$) of iterations. We then utilize this fact to show that the difference of the coupled sequence will (with some constant probability) grow exponentially large, eventually moving out of their specified radius $\phi$ within $T_s$ iterations, leading to a contradiction.

Our first result formally introduces the coupling, setting the stage for the rest of our arguments. For notational convenience, in this section, unless otherwise specified, we will often assume that the initial iterate $x_0$ is an $\epsilon$-saddle point.

**Lemma 3.** *Suppose $x_0$ is an $\epsilon$-approximate saddle point. Without loss of generality, suppose that the minimum eigendirection of $H := \nabla^2 f(x_0)$ is the $e_1$ direction, and let $\gamma$ to denote $-\lambda_{\min}(\nabla^2 f(x_0))$ (note $\gamma \geq \sqrt{\rho\epsilon}$). Consider the following coupling mechanism, where we run the zeroth-order gradient dynamics, starting with $x_0$, with two isotropic noise sequences, $Y_t$ and $Y_t'$ respectively, where $(Y_t)_1 = -(Y_t)_1'$, and $(Y_t)_j = (Y_t)_j'$ for all other $j \neq 1$. Suppose that the sequence $\{Z_{t,i}\}_{t \in T, i \in [m]}$ is the same for both sequences. Let $\{x_t\}$ denote the sequence with the $\{Y_t\}$ noise sequence, and let the $\{x_t'\}$ denote the sequence with the $\{Y_t'\}$ noise sequence, where*

$$x'_{t+1} = x'_t - \eta \left( \frac{1}{m} \sum_{i=1}^m \left( Z_{t,i} Z_{t,i}^\top \nabla f(x_t') + \frac{u}{2} Z_{t,i} Z_{t,i}^\top \tilde{H}'_{t,i} Z_{t,i} \right) + Y_t' \right), \quad x_0' = x_0,$$

*and $\tilde{H}'_{t,i} := \frac{H'_{t,i,+} - H'_{t,i,-}}{2}$, with $H'_{t,i,+} = \nabla^2 f(x_t' + \alpha'_{t,i,+} u Z_i')$ for some $\alpha'_{t,i,+} \in [0,1]$, and $H'_{t,i,-} = \nabla^2 f(x_t - \alpha'_{t,i,-} u Z_i')$ for some $\alpha'_{t,i,-} \in [0,1]$. Then, for any $t \geq 0$,*

$$\hat{x}_{t+1} := x_{t+1} - x'_{t+1} = \underbrace{-\eta \sum_{\tau=0}^t (I - \eta H)^{t-\tau} \hat{\xi}_{g_0}(\tau)}_{W_{g_0}(t+1)} \underbrace{-\eta \sum_{\tau=0}^t (I - \eta H)^{t-\tau} (\bar{H}_\tau - H)\hat{x}_\tau}_{W_H(t+1)}$$

$$\underbrace{-\eta \sum_{\tau=0}^t (I - \eta H)^{t-\tau} \hat{\xi}_u(\tau)}_{W_u(t+1)} \underbrace{-\eta \sum_{\tau=0}^t (I - \eta H)^{t-\tau} \hat{Y}_\tau}_{W_p(t+1)}$$

*where*

$$\xi_{g_0}(t) = \frac{1}{m} \sum_{i=1}^m (Z_{t,i} Z_{t,i}^\top - I) \nabla f(x_t), \quad \xi'_{g_0}(t) = \frac{1}{m} \sum_{i=1}^m (Z_{t,i}(Z_{t,i})^\top - I) \nabla f(x_t'), \quad \hat{\xi}_{g_0}(t) = \xi_{g_0}(t) - \xi'_{g_0}(t),$$

$$\xi_u(t) = \frac{1}{m} \sum_{i=1}^m \frac{u}{2} Z_{t,i} Z_{t,i} \tilde{H}_{t,i} Z_{t,i}, \quad \xi'_u(t) = \frac{1}{m} \sum_{i=1}^m \frac{u}{2} Z_{t,i} Z_{t,i} \tilde{H}'_{t,i} Z_{t,i}, \quad \hat{\xi}_u(t) = \xi_u(t) - \xi'_u(t),$$

$$\hat{Y}_t = Y_t - Y_t', \quad \bar{H}_t = \int_0^1 \nabla^2 f(ax_t + (1-a)x_t')da.$$

*Proof.* Observe that

$$\hat{x}_{t+1} := x_{t+1} - x'_{t+1}$$
$$= x_t - \eta \left( \nabla f(x_t) + \xi_{g_0}(t) + \xi_u(t) Y_t \right) - \left[ x_t' - \eta \left( \nabla f(x_t') + \xi'_{g_0}(t) + \xi'_u(t) + Y_t' \right) \right]$$
$$= \hat{x}_t - \eta \left[ (\nabla f(x_t) - \nabla f(x_t')) + \left( \xi_{g_0}(t) - \xi'_{g_0}(t) \right) + (\xi_u(t) - \xi'_u(t)) + (Y_t - Y_t') \right]$$
$$= \hat{x}_t - \eta H \hat{x}_t - \eta(\bar{H}_t - H)\hat{x}_t - \eta\hat{\xi}_{g_0}(t) - \eta\hat{\xi}_u(t) - \eta\hat{Y}_t$$
$$= \underbrace{-\eta \sum_{\tau=0}^t (I - \eta H)^{t-\tau} \hat{\xi}_{g_0}(\tau)}_{W_{g_0}(t+1)} \underbrace{-\eta \sum_{\tau=0}^t (I - \eta H)^{t-\tau} (\bar{H}_\tau - H)\hat{x}_\tau}_{W_H(t+1)} \underbrace{-\eta \sum_{\tau=0}^t (I - \eta H)^{t-\tau} \hat{\xi}_u(\tau)}_{W_u(t+1)} \underbrace{-\eta \sum_{\tau=0}^t (I - \eta H)^{t-\tau} \hat{Y}_\tau}_{W_p(t+1)}$$

where

$$\xi_{g_0}(t) = \frac{1}{m} \sum_{i=1}^m (Z_{t,i} Z_{t,i}^\top - I) \nabla f(x_t), \quad \xi'_{g_0}(t) = \frac{1}{m} \sum_{i=1}^m (Z_{t,i}(Z_{t,i})^\top - I) \nabla f(x_t'), \quad \hat{\xi}_{g_0}(t) = \xi_{g_0}(t) - \xi'_{g_0}(t),$$

$$\xi_u(t) = \frac{1}{m} \sum_{i=1}^m \frac{u}{2} Z_{t,i} Z_{t,i} \tilde{H}_{t,i} Z_{t,i}, \quad \xi'_u(t) = \frac{1}{m} \sum_{i=1}^m \frac{u}{2} Z_{t,i} Z_{t,i} \tilde{H}'_{t,i} Z_{t,i}, \quad \hat{\xi}_u(t) = \xi_u(t) - \xi'_u(t),$$

$$\hat{Y}_t = Y_t - Y_t', \quad \bar{H}_t = \int_0^1 \nabla^2 f(ax_t + (1-a)x_t')da.$$

To derive the final equality, we utilized the fact that $x_0' = x_0$. This completes our proof. $\qquad\square$

Suppose $x_0$ is an $\epsilon$-saddle point. Recall that $\gamma > 0$ denotes $-\lambda_{\min}(\nabla^2 f(x_0))$, where we know that $\gamma \geq \sqrt{\rho\epsilon}$.

$$\gamma \geq \bar{\psi} := \begin{cases} \min\{\psi, 1, L\} & \text{if } f(\cdot) \text{ is } (\epsilon, \psi)\text{-strict saddle for any } \psi > \sqrt{\rho\epsilon} \\ \sqrt{\rho\epsilon} & \text{otherwise.} \end{cases}$$

In the sequel, for any $t \geq 0$, it is helpful to define the quantities

$$\beta(t)^2 := \frac{(1+\eta\gamma)^{2t}}{(\eta\gamma)^2 + 2\eta\gamma}, \quad \alpha(t)^2 := \frac{(1+\eta\gamma)^{2t} - 1}{(\eta\gamma)^2 + 2\eta\gamma}. \tag{44}$$

We next introduce some probabilistic events (and their implications) which, if true, can be used to bound the sizes of $\|W_{g_0}(t+1)\|$, $\|W_u(t+1)\|$, $\|W_u(t+1)\|$ (and as we will see in the next result, indirectly bound $\|W_H(t+1)\|$). These bounds will be useful in the final proof of making function value progress near a saddle point.

**Lemma 24.** *We assume $\delta \in (0, 1/e]$ throughout the lemma. Suppose that we pick $u, r$ and $\eta$ as specified in Lemma 23. Suppose $T_s \geq t_f(\delta)$. Suppose also that*

$$f(x_{T_s}) - f(x_0) > -F, \quad f(x'_{T_s}) - f(x_0) > -F.$$

*Then, we have the following results.*

1. *Let $\mathcal{S}_\phi(\delta)$ denote the event*

$$\mathcal{S}_\phi(\delta) := \left\{ \max\{\|x_t - x_0\|^2, \|x'_t - x_0\|^2\} \leq \phi_{T_s}(\delta, F), \quad \forall 0 \leq t \leq T_s \right\}.$$

   *In addition, let $\mathcal{S}_u(\delta)$ denote the event*

$$\mathcal{S}_u(\delta) := \left\{ \|W_u(t+1)\| \leq \eta\beta(t+1) \frac{\sqrt{3}}{\sqrt{\eta\bar{\psi}}} \left(2c_3\rho d^2 (\log(T/\delta))^2\right) u^2, \quad \forall 0 \leq t \leq T_s - 1 \right\},$$

   *where $c_3$ is the same absolute constant as the $c_3$ in the preceding lemmas. Then,*

$$\mathbb{P}(\mathcal{S}_\phi(\delta) \cap \mathcal{S}_u(\delta)) \geq 1 - \frac{24T_s\delta}{T}.$$

2. *Consider defining the event $\mathcal{R}_t(\delta)$, which is the event where*

   *either* $\sum_{\tau=0}^{t}(1+\eta\gamma)^{2(t-\tau)}\frac{dL^2}{m}\|x_\tau - x'_\tau\|^2 (\text{lr}(CT^2/\delta))^2 \geq G_{T_s}(\delta, F)$, *or*

   $\|W_{g_0}(t+1)\|$

$$\leq c'\eta \sqrt{\max\left\{ \left(\text{lr}\left(\frac{CT^2}{\delta}\right)\right)^2 \sum_{\tau=0}^{t} \frac{dL^2}{m}(1+\eta\gamma)^{2(t-\tau)}\|x_\tau - x'_\tau\|^2, g(t+1) \right\} \left(\log\left(\frac{CdT^2}{\delta}\right) + \log\left(\log\left(\frac{G_{T_s}(\delta, F)}{g(t+1)}\right)\right) + 1\right)}$$

   *normalsize holds. Above, $c', C$ refer to the same constants as in Proposition 2, and*

$$G_{T_s}(\delta, F) := 8\sum_{\tau=0}^{T_s-1}(1+\eta\gamma)^{2\tau}\frac{dL^2}{m}(\text{lr}(CT^2/\delta))^2\phi_{T_s}(\delta, F) + \left(\frac{\beta(T_s)\eta r}{60\sqrt{d}}\right)^2, \quad g(t+1) := \left(\frac{\beta(t+1)\eta r}{60\sqrt{d}}\right)^2.$$

   *Then, $\mathbb{P}(\mathcal{R}_t(\delta)) \geq 1 - \frac{\delta}{T}$. Suppose the event*

$$\left(\cap_{t=0}^{T_s-1}\mathcal{R}_t(\delta)\right) \cap \mathcal{S}_\phi(\delta)$$

   *holds. Then, the event $\mathcal{S}_{g_0}(\delta)$ holds, where*

$$\mathcal{S}_{g_0}(\delta) := \cap_{t=0}^{T_s-1}\mathcal{S}_{g_0,t}(\delta),$$

   *and $\mathcal{S}_{g_0,t}(\delta)$ is defined as*

$$\mathcal{S}_{g_0,t}(\delta) := \left\{ \|W_{g_0}(t+1)\| \leq \zeta_1(\delta, F)c'\eta \sqrt{\max\left\{ \left(\text{lr}\left(\frac{CT^2}{\delta}\right)\right)^2 \sum_{\tau=0}^{t} \frac{dL^2}{m}(1+\eta\gamma)^{2(t-\tau)}\|x_\tau - x'_\tau\|^2, g(t+1) \right\}} \right\}$$

   *where*

$$\zeta_1(\delta, F) := \left(\log\left(\frac{CdT^2}{\delta}\right) + \log\left(\log\left(\frac{G_{T_s}(\delta, F)}{g(1)}\right)\right) + 1\right).$$

3. *In addition, let $\mathcal{S}_p(\delta)$ denote the event*

$$\mathcal{S}_p(\delta) := \left\{ \|W_p(t+1)\| \leq \frac{2\sqrt{2\log(T/\delta)}\beta(t+1)\eta r}{\sqrt{d}} \quad \forall 0 \leq t \leq T_s - 1 \right\}.$$

*Then, $\mathbb{P}(\mathcal{S}_p(\delta)) \geq 1 - \frac{T_s\delta}{T}$.*

*Proof.* We consider the three claims separately.

1. Note that our assumptions satisfy the conditions required in Lemma 23. Hence, by Lemma 23, on the event $\mathcal{P}_{0,T_s}(\delta, F)$, we have that $\|x_\tau - x_0\|^2 \leq \phi_{T_s}(\delta, F)$. Simultaneously, on the event $\mathcal{P}_{0,T_s}(\delta, F)$, we know that $\cap_{t=0}^{T_s-1}\mathcal{G}_t(\delta)$ holds, i.e.

$$\frac{1}{m}\sum_{i=1}^{m}\|Z_{t,i}\|^4 \leq 2c_3 d^2 \left(\log(T/\delta)\right)^2, \quad \forall 0 \leq t \leq T_s - 1. \tag{45}$$

Thus, for $W_u(t+1)$, we have that

$$
\begin{aligned}
\|W_u(t+1)\| &= \left\| \eta \sum_{\tau=0}^{t}(I - \eta H)^{t-\tau}\hat{\xi}_u(\tau) \right\| \\
&\leq \left\| \eta \sum_{\tau=0}^{t}(I - \eta H)^{t-\tau}\xi_u(\tau) \right\| + \left\| \eta \sum_{\tau=0}^{t}(I - \eta H)^{t-\tau}\hat{\xi}'_u(\tau) \right\| \\
&\leq \eta \sum_{\tau=0}^{t}(1 + \eta\gamma)^{t-\tau}\left( \left\| \frac{1}{m}\sum_{i=1}^{m}\frac{u}{2}Z_{t,i}Z_{t,i}\tilde{H}_{t,i}Z_{t,i} \right\| + \left\| \frac{1}{m}\sum_{i=1}^{m}\frac{u}{2}Z_{t,i}Z_{t,i}\tilde{H}'_{t,i}Z_{t,i} \right\| \right) \\
&\leq \eta \sum_{\tau=0}^{t}(1 + \eta\gamma)^{t-\tau}\frac{\rho}{m}\sum_{i=1}^{m}\|Z_{t,i}\|^4 u^2 \\
&\overset{(iv)}{\leq} \eta \sum_{\tau=0}^{t}(1 + \eta\gamma)^{t-\tau}\rho(2c_3)d^2(\log(T2/\delta))^2 u^2 \\
&\leq \eta \frac{(1 + \eta\gamma)^{t+1}}{\eta\gamma}\left( 2c_3\rho C d^2(\log(T/\delta))^2 \right) u^2 \\
&\overset{(v)}{=} \eta\beta(t+1)\frac{\sqrt{(\eta\gamma)^2 + 2\eta\gamma}}{\eta\gamma}\left( 2c_3\rho d^2(\log(T/\delta))^2 \right) u^2 \\
&\leq \eta\beta(t+1)\frac{\sqrt{3}}{\sqrt{\eta\gamma}}\left( 2c_3\rho d^2(\log(T/\delta))^2 \right) u^2 \\
&\overset{(vi)}{\leq} \eta\beta(t+1)\frac{\sqrt{3}}{\sqrt{\eta\bar{\psi}}}\left( 2c_3\rho d^2(\log(T/\delta))^2 \right) u^2
\end{aligned}
$$

where the inequality in (iv) holds due to Eq. (45), the equality in (v) holds due to the definition of $\beta(t+1)$, and the inequality in (vi) used the fact that $\gamma \geq \bar{\psi}$.

Hence the event

$$\cap_{t=0}^{T_s}\left\{ \|x_t - x_0\|^2 \leq \phi_{T_s}(\delta, F) \text{ and } \right\} \cap \mathcal{S}_u(\delta)$$

holds with probability at least $1 - \frac{12T_s\delta}{T}$.

Note that by the coupling, the distribution of $x'_\tau$ is the same as that of $x_\tau$. Thus, by the assumption $f(x'_{T_s}) - f(x_0) > -F$, it follows by a similar argument that the bound $\|x'_\tau - x_0\|^2 \leq \phi_{T_s}(\delta, F)$ also holds with probability at least $1 - \frac{12T_s\delta}{T}$. The claim then follows by an application of the union bound.

2. For the second claim, observe first that the claim $\mathbb{P}(\mathcal{R}_t(\delta)) \geq 1 - \frac{\delta}{T}$ is a consequence of Proposition 2. Suppose next that $f(x_{T_s}) - f(x_0) > -F$. Then, by definition of the event $\mathcal{S}_\phi(\delta)$, we know that

$$\|x_\tau - x_0\|^2 \leq \phi_{T_s}(\delta, F), \qquad \|x'_\tau - x_0\|^2 \leq \phi_{T_s}(\delta, F)$$

where $\phi_{T_s}(\delta, F)$ is as defined in Lemma 23.

Suppose now that $\mathcal{R}_t(\delta)$ holds true, and suppose for contradiction that

$$\sum_{\tau=0}^{t} (1 + \eta\gamma)^{2(t-\tau)} \frac{dL^2}{m} \|x_\tau - x'_\tau\|^2 (\mathrm{lr}(CT^2/\delta))^2$$
$$\geq G_{T_s}(\delta, F)$$
$$= 8 \sum_{\tau=0}^{T_s-1} (1 + \eta\gamma)^{2\tau} \frac{dL^2}{m} (\mathrm{lr}(CT^2/\delta))^2 \phi_{T_s}(\delta, F) + \left( \frac{\beta(T_s)\eta r}{60\sqrt{d}} \right)^2.$$

This implies that there exists some $0 \leq \tau \leq t \leq T_s$ such that $\|x_\tau - x'_\tau\|^2 \geq 8\phi_{T_s}(\delta, F)$. However, we also know that on the event $\mathcal{S}_\phi(\delta)$,

$$\|x_\tau - x'_\tau\|^2 \leq 2\|x_\tau - x_0\|^2 + 2\|x'_\tau - x_0\|^2 \leq 4\phi_{T_s}(\delta, F).$$

This leads to a contradiction. We must then have that

$$\|W_{g_0}(t+1)\| \leq \zeta_1(\delta, F) c'\eta \sqrt{\max \left\{ \left( \mathrm{lr}\left( \frac{CT^2}{\delta} \right) \right)^2 \sum_{\tau=0}^{t} \frac{dL^2}{m} (1 + \eta\gamma)^{2(t-\tau)} \|x_\tau - x'_\tau\|^2, g(t+1) \right\}}$$

where

$$\zeta_1(\delta, F) := \sqrt{\log\left( \frac{CdT^2}{\delta} \right) + \log\left( \log\left( \frac{G(\delta, F)}{g(1)} \right) + 1 \right)}$$

3. Observe that

$$W_p(t+1) = \eta \sum_{\tau=0}^{t} (I - \eta H)^{t-\tau} \hat{Y}_\tau = \eta \sum_{\tau=0}^{t} (1 + \eta\gamma)^{t-\tau} (2(Y_\tau)_1),$$

which means that $W_p(t+1)$ is a 1-dimensional Gaussian with variance

$$\eta^2 \sum_{\tau=0}^{t} (1 + \eta\gamma)^{2(t-\tau)} \frac{4r^2}{d} = \frac{4\eta^2 r^2}{d} \frac{(1 + \eta\gamma)^{2(t+1)} - 1}{2\eta\gamma + (\eta\gamma)^2} = \frac{4\eta^2 r^2 \alpha(t+1)^2}{d}. \tag{46}$$

Since $\alpha(t+1) \leq \beta(t+1)$, using the subGaussianity of a Gaussian distribution, it follows that for any $t$, with probability at least $1 - \delta/T$,

$$\|W_p(t+1)\| \leq \frac{2\sqrt{2\log(T/\delta)}\beta(t+1)\eta r}{\sqrt{d}}.$$

$\square$

For any $F > 0$, we are now ready to show that the algorithm makes a function decrease of $F$ with $\Omega(1)$ probability near an $\epsilon$-saddle point.

**Proposition 5.** *Suppose that $x_{t_0}$ is an $\epsilon$-approximate saddle point. Let $c' > 0, c_1 > 0, c_2 \geq 1, c_4 > 0, c_5 > 0, c_6 > 0, c_7 > 0, C_1 \geq 1$ be the absolute constants defined in the statements of the previous lemmas, and let $\delta \in (0, 1/e]$ be arbitrary. Consider any $F > 0$. As in the statement of Lemma 23, suppose we choose $u, r$ and $\eta$ such that*

$$u \leq \frac{\sqrt{\epsilon}}{d\sqrt{\rho}\log(T/\delta)} \cdot \min\left\{ \frac{1}{64c_5^2 c_2}, \frac{1}{2048c_1 c_2} \right\}^{1/4}, \qquad r \leq \epsilon \cdot \min\left\{ \frac{1}{8c_5\sqrt{2c_2}}, \frac{1}{32\sqrt{c_1}} \right\},$$

$$\eta \leq \frac{1}{Lt_f(\delta)} \min\left\{ \frac{1}{\log(T/\delta)}, \frac{\sqrt{m}}{8c_4(\mathrm{lr}(C_1 dmT/\delta))^{3/2}\sqrt{d}}, \frac{m}{128c_1(\mathrm{lr}(C_1 dmT/\delta))^3 d} \right\}.$$

*Suppose we pick*

$$T_s = \max\left\{ \lceil \frac{\iota}{\eta\bar{\psi}} \rceil, t_f(\delta), 4 \right\}, \tag{47}$$

*where*

$$\iota = \max\left\{ \log\left( 2\sqrt{\phi_{T_s}(\delta, F)} \frac{20\sqrt{d}\sqrt{\eta^2\gamma^2 + 2\eta\gamma}}{\eta r} \right), 1 \right\},$$

$$\bar{\psi} := \begin{cases} \min\{\psi, 1, L\} & \textit{if } f(\cdot) \textit{ is } (\epsilon, \psi)\textit{-strict saddle for any } \psi > \sqrt{\rho\epsilon} \\ \sqrt{\rho\epsilon} & \textit{otherwise.} \end{cases}$$

*Suppose in addition that $u, \eta$ also satisfy the conditions*

$$u \leq \sqrt{\frac{r\sqrt{\eta\bar{\psi}}}{120\sqrt{3}c_3\sqrt{d}\rho d^2(\log(T/\delta))^2}},$$

$$\eta \leq \max\left\{ \frac{1}{c'c_9\zeta_1(\delta, F)}, \frac{m\bar{\psi}}{360\iota(c')^2c_9^2 dL^2\left(\mathrm{lr}\left(\frac{CT^2}{\delta}\right)\right)^2 \zeta_1(\delta, F)^2}, \frac{1}{2\bar{\psi}} \right\},$$

*where $\zeta_1(\delta, F)$ is as defined in Lemma 23, $c', c_3, C > 0$ are the same constants as in the previous results, and $c_9 = 2\sqrt{2} + \frac{1}{20}$. Suppose also that $\phi_{T_s}(\delta, F)$ satisfies the bound*

$$\phi_{T_s}(\delta, F) \leq \left( \frac{\bar{\psi}}{60c_9\iota\rho\log(T/\delta)} \right)^2. \tag{48}$$

*Then, with probability at least $\frac{1}{3} - \frac{13T_s\delta}{T}$, $f(x_{t_0+T_s}) - f(x_{t_0}) \leq -F$.*

*Proof of Proposition 5.* Without loss of generality, we assume that $t_0 = 0$. By Lemma 3, we have

$$\hat{x}_{t+1}$$
$$:= x_{t+1} - x'_{t+1}$$
$$= \underbrace{-\eta\sum_{\tau=t_0}^{t}(I-\eta H)^{t-\tau}\hat{\xi}_{g_0}(\tau)}_{W_{g_0}(t+1)} \underbrace{-\eta\sum_{\tau=t_0}^{t}(I-\eta H)^{t-\tau}(\bar{H}_\tau - H)\hat{x}_\tau}_{W_H(t+1)} \underbrace{-\eta\sum_{\tau=t_0}^{t}(I-\eta H)^{t-\tau}\hat{\xi}_u(\tau)}_{W_u(t+1)} \underbrace{-\eta\sum_{\tau=t_0}^{t}(I-\eta H)^{t-\tau}\hat{Y}_\tau}_{W_p(t+1)}$$

where

$$\xi_{g_0}(t) = \frac{1}{m}\sum_{i=1}^{m}(Z_{t,i}Z_{t,i}^\top - I)\nabla f(x_t), \quad \xi'_{g_0}(t) = \frac{1}{m}\sum_{i=1}^{m}(Z_{t,i}(Z_{t,i})^\top - I)\nabla f(x'_t), \quad \hat{\xi}_{g_0}(t) = \xi_{g_0}(t) - \xi'_{g_0}(t),$$

$$\xi_u(t) = \frac{1}{m}\sum_{i=1}^{m}\frac{u}{2}Z_{t,i}Z_{t,i}\tilde{H}_{t,i}Z_{t,i}, \quad \xi'_u(t) = \frac{1}{m}\sum_{i=1}^{m}\frac{u}{2}Z_{t,i}Z_{t,i}\tilde{H}'_{t,i}Z_{t,i}, \quad \hat{\xi}_u(t) = \xi_u(t) - \xi'_u(t),$$

$$\hat{Y}_t = Y_t - Y'_t, \quad \bar{H}_t = \int_0^1 \nabla^2 f(ax_t + (1-a)x'_t)da.$$

Recall that we define for $t \geq 0$,

$$\beta(t)^2 := \frac{(1+\eta\gamma)^{2t}}{(\eta\gamma)^2 + 2\eta\gamma}, \quad \alpha(t)^2 := \frac{(1+\eta\gamma)^{2t} - 1}{(\eta\gamma)^2 + 2\eta\gamma}.$$

Throughout the proof, we suppose for contradiction that

$$f(x_{T_s}) - f(x_0) > -F, \quad f(x'_{T_s}) - f(x_0) > -F,$$

and assume the event

$$\left( \cap_{t=0}^{T_s-1} \mathcal{R}_t(\delta) \right) \cap \mathcal{S}_\phi(\delta) \cap \mathcal{S}_u(\delta) \cap \mathcal{S}_p(\delta)$$

holds, where the events intersected are defined in Lemma 24. Then, by Lemma 24, the event $\mathcal{S}_{g_0}(\delta)$ (also defined in Lemma 24) holds[7].

Consider the following induction argument, where we seek to show that there exists an absolute constant $c_9 > 0$ such that for every $t \in \{0, 1, \ldots, T_s\}$,

$$\|x_t - x_t'\| \le c_9 \log(T/\delta) \frac{\beta(t)\eta r}{\sqrt{d}}, \text{ and } \max\left\{ \|W_{g_0}(t)\|, \|W_H(t)\|, \|W_u(t)\| \right\} \le \frac{\beta(t+1)\eta r}{\sqrt{d}} \tag{49}$$

Combined with a lower bound on $\|W_p(t+1)\|$ (which makes use of the property that $W_p(t+1)$ is a 1-dimensional Gaussian), we will then use the inductive claim in Eq. (49) to show that

$$\|W_p(T_s)\| \ge 2 \left( \|W_{g_0(T_s)}\| + \|W_H(T_s)\| + \|W_u(T_s)\| \right).$$

Since $W_p(t+1)$ is a 1-dimensional Gaussian random variable with a standard deviation that grows exponentially with $t$, by our choice of $T_s$, we will see that $\|x_{T_s} - x'_{T_s}\|$ is larger than what expect (since our assumptions imply that $\max\left\{ \|x_{T_s} - x_0\|^2, \|x'_{T_s} - x_0\|^2 \right\} \le \phi_{T_s}(\delta, F)$, i.e. $x_{T_s}$ and $x'_{T_s}$ both remain close to $x_0$ and hence close to each other). This yields a contradiction, implying that on the event we assumed to hold, i.e.

$$\left( \cap_{t=0}^{T_s-1} \mathcal{R}_t(\delta) \right) \cap \mathcal{S}_\phi(\delta) \cap \mathcal{S}_p(\delta)$$

the assumption

$$f(x_{T_s}) - f(x_0) > -F, \quad \text{and } f(x'_{T_s}) - f(x_0) > -F$$

is not true, i.e. one of the sequences must have made function value progress of at least $F$.

We proceed to prove Eq. (49). Observe that the claim holds for the base case $t = 0$; this is true since $x_0 = x_0'$. Now suppose that this holds for all $\tau \le t$. We will seek to show that Eq. (49) holds for $t + 1$ as well. We do so by bounding the norms of $W_{g_0}(t+1), W_H(t+1), W_u(t+1)$ and $W_p(t+1)$ respectively.

1. (Bounding $\|W_{g_0}(t+1)\|$) Since the event $\mathcal{S}_{g_0}(\delta)$ holds, it follows that for each $0 \le t \le T_s - 1$, we have that

$$\|W_{g_0}(t+1)\| \le \zeta_1(\delta, F) c'\eta \sqrt{\max\left\{ \left( \text{lr}\left(\frac{CT^2}{\delta}\right) \right)^2 \sum_{\tau=0}^{t} \frac{dL^2}{m}(1+\eta\gamma)^{2(t-\tau)} \|x_\tau - x_\tau'\|^2, g(t+1) \right\}}$$

where

$$\zeta_1(\delta, F) := \left( \log\left(\frac{CdT^2}{\delta}\right) + \log\left( \log\left(\frac{G_{T_s}(\delta, F)}{g(1)}\right) \right) + 1 \right),$$

and the terms $G_{T_s}(\delta, F)$ and $g(1)$ are defined as in Lemma 24. Recall by the inductive claim in Eq. (49) that there exists $c_9 > 0$ such that

$$\|x_\tau - x_\tau'\| \le c_9 \log(T/\delta) \frac{\beta(t)\eta r}{\sqrt{d}} \quad \forall\, 0 \le \tau \le t.$$

Hence, it follows that

$$\|W_{g_0}(t+1)\| \le c'\zeta_1(\delta, F)\eta \max\left\{ \sqrt{t+1}\left( \text{lr}\left(\frac{CT^2}{\delta}\right) \right) \frac{c_9\sqrt{d}L}{\sqrt{m}} \frac{\beta(t)\eta r}{\sqrt{d}}, \frac{\beta(t+1)\eta r}{60\sqrt{d}} \right\}.$$

---

[7]We may also directly assume that $\mathcal{S}_{g_0}(\delta)$ also holds, but our way of reasoning prevents double counting of probabilities.

Hence, noting the choice of $T_s$ in Eq. (47), by choosing $\eta$ such that

$$c'c_9\zeta_1(\delta,F)\eta\sqrt{T_s}\left(\mathrm{lr}\left(\frac{CT^2}{\delta}\right)\right)\frac{\sqrt{d}L}{\sqrt{m}}\leq\frac{1}{60}\iff\eta\leq\frac{m\bar{\psi}}{360\iota(c')^2c_9^2dL^2\left(\mathrm{lr}\left(\frac{CT^2}{\delta}\right)\right)^2\zeta_1(\delta,F)^2},\text{ and}$$
$$\tag{50}$$

$$c'c_9\zeta_1(\delta,F)\eta\leq 1.$$

it follows that

$$\|W_{g_0}(t+1)\|\leq\frac{\beta(t+1)\eta r}{60\sqrt{d}}.$$

2. Meanwhile, the term $W_H(t+1)$ can be bounded as follows. By the inductive assumption in Eq. (49), we have that

$$\|\hat{x}_\tau\|=\|x_\tau-x'_\tau\|\leq c_9\log(T/\delta)\frac{\beta(\tau)\eta r}{\sqrt{d}}\quad\forall\,0\leq\tau\leq t.$$

Moreover, on the event our proof assumes, we know that

$$\max\left\{\|x_\tau-x_0\|^2,\|x'_\tau-x_0\|^2\right\}\leq\phi_{T_s}(\delta,F).$$

Thus, using the $\rho$-Hessian Lipschitz property, we have

$$\|W_H(t+1)\|=\eta\left\|\sum_{\tau=0}^{t}(I-\eta H)^{t-\tau}(\bar{H}_\tau-H)\hat{x}_\tau\right\|$$
$$\leq\eta\sum_{\tau=0}^{t}(1+\eta\gamma)^{t-\tau}\rho\sqrt{\phi_{T_s}(\delta,F)}\frac{c_9\log(T/\delta)\beta(\tau)\eta r}{\sqrt{d}}$$
$$\leq c_9(t+1)\log(T/\delta)\eta\rho\sqrt{\phi_{T_s}(\delta,F)}\frac{\beta(t)\eta r}{\sqrt{d}}$$
$$\leq c_9 T_s\log(T/\delta)\eta\rho\sqrt{\phi_{T_s}(\delta,F)}\frac{\beta(t)\eta r}{\sqrt{d}}.$$

Given our choice of $T_s$ in Eq. (47), if

$$c_9 T_s\log(T/\delta)\eta\rho\sqrt{\phi_{T_s}(\delta,F)}\leq\frac{1}{60}\iff\phi_{T_s}(\delta,F)\leq\left(\frac{\bar{\psi}}{60c_9\iota\rho\log(T/\delta)}\right)^2$$

it follows that

$$\|W_H(t+1)\|\leq\frac{\beta(t+1)\eta r}{60\sqrt{d}}.$$

3. Meanwhile, for $W_u(t+1)$, since the event $\mathcal{S}_u(\delta)$ holds, we have that

$$\|W_u(t+1)\|\leq\eta\beta(t+1)\frac{\sqrt{3}}{\sqrt{\eta\bar{\psi}}}\left(2c_3\rho d^2(\log(T/\delta))^2\right)u^2.$$

Now, by picking

$$\eta\beta(t+1)\frac{\sqrt{3}}{\sqrt{\eta\bar{\psi}}}\left(2c_3\rho d^2(\log(T/\delta))^2\right)u^2\leq\frac{\beta(t+1)\eta r}{60\sqrt{d}}\iff u\leq\sqrt{\frac{r\sqrt{\eta\bar{\psi}}}{120\sqrt{3}c_3\sqrt{d}\rho d^2(\log(T/\delta))^2}},$$

it follows that with probability $1-\delta/T$, $\|W_u(t+1)\|\leq\frac{\beta(t+1)\eta r}{60\sqrt{d}}$.

4. Meanwhile, observe that since $\mathcal{S}_p(\delta)$ holds, it follows that

$$W_p(t+1)\leq\frac{2\sqrt{2\log(T/\delta)}\beta(t+1)\eta r}{\sqrt{d}}.$$

Combining the bounds for $W_{g_0}, W_p, W_H$ and $W_u$, it follows that

$$\|\hat{x}_{t+1}\| \le \|W_{g_0}(t+1)\| + \|W_p(t+1)\| + \|W_H(t+1)\| + \|W_u(t+1)\|$$

$$\le \frac{\beta(t+1)\eta r}{\sqrt{d}} \left( \frac{1}{60} + \frac{1}{60} + \frac{1}{60} + 2\sqrt{2\log(T/\delta)} \right)$$

$$\le \frac{\beta(t+1)\eta r}{\sqrt{d}} \left( \frac{1}{20} + 2\sqrt{2} \right) \log(T/\delta),$$

where the final inequality uses the fact that $0 < \delta \le 1/e$ (which implies $\log(T/\delta) \ge 1$). Hence, we see that the first part of the inductive claim of Eq. (49) holds with the constant $c_9 := \frac{1}{20} + 2\sqrt{2}$, and the second part follows naturally as a consequence of our argument above.

Meanwhile, observe that for any $\eta$ such that $\eta\bar{\psi} \le \frac{1}{2}$, we have that $(1 + \eta\gamma)^{\frac{1}{\eta\bar{\psi}}} \ge 2$. Thus, by choosing $\eta$ such that $\eta\bar{\psi} \le \frac{1}{2}$, we have that for any $t \ge \frac{1}{\eta\bar{\psi}}$,

$$\alpha(t+1)^2 \ge \frac{1}{2}\beta(t+1)^2.$$

Hence, following Eq. (46), by choosing $T_s \ge \frac{1}{\eta\bar{\psi}}$, $W_p(T_s)$ is a 1-dimensional Gaussian with variance at least $\frac{2\eta^2 r^2 \beta(T_s)}{d}$, such that with probability at least 2/3,

$$\|W_p(T_s)\| \ge \frac{\beta(T_s)\eta r}{10\sqrt{d}}.$$

Simultaneously, we know that on the event

$$\left( \cap_{t=0}^{T_s-1} \mathcal{R}_t(\delta) \right) \cap \mathcal{S}_\phi(\delta) \cap \mathcal{S}_u(\delta) \cap \mathcal{S}_p(\delta),$$

we have

$$\|W_{g_0}(T_s)\| + \|W_H(T_s)\| + \|W_u(T_s)\| \le \frac{3\beta(T_s)\eta r}{60\sqrt{d}} = \frac{\beta(T_s)\eta r}{20\sqrt{d}}.$$

We note that by Lemma 24, we have

$$\mathbb{P}\left( \left( \cap_{t=0}^{T_s-1} \mathcal{R}_t(\delta) \right) \cap \mathcal{S}_\phi(\delta) \cap \mathcal{S}_u(\delta) \cap \mathcal{S}_p(\delta) \right) \ge 1 - \left( \frac{24T_s\delta}{T} + \frac{T_s\delta}{T} + \frac{T_s\delta}{T} \right) = 1 - \frac{26T_s\delta}{T}.$$

Thus, with probability at least $2/3 - \frac{26T_s\delta}{T}$, we have

$$\|\hat{x}_{T_s}\| \ge \frac{1}{2}\|W_p(T_s)\| \ge \frac{\beta(T_s)\eta r}{20\sqrt{d}}$$

Thus, choosing $T_s \ge \frac{\iota}{\eta\bar{\psi}}$, where

$$\iota = \max\left\{ \log\left( 2\sqrt{\phi_{T_s}(\delta, F)} \frac{20\sqrt{d}\sqrt{\eta^2\gamma^2 + 2\eta\gamma}}{\eta r} \right), 1 \right\},$$

noting that if $\eta\bar{\psi} \le 1/2$, then $(1 + \eta\gamma)^{\frac{1}{\eta\bar{\psi}}} \ge (1 + \eta\bar{\psi})^{\frac{1}{\eta\bar{\psi}}} \ge 2$, we have that with probability at least $2/3 - \frac{26T_s\delta}{T}$,

$$\|\hat{x}_{T_s}\| \ge \frac{\beta(T_s)\eta r}{20\sqrt{d}} = \frac{\eta r}{20\sqrt{d}} \frac{(1 + \eta\gamma)^{T_s}}{\sqrt{2\eta\gamma + (\eta\gamma)^2}}$$

$$\ge \frac{\eta r}{20\sqrt{d}} \frac{(1 + \eta\gamma)^{\frac{\log\left( 2\sqrt{\phi_{T_s}(\delta, F)} \frac{20\sqrt{d}\sqrt{\eta^2\gamma^2 + 2\eta\gamma}}{\eta r} \right)}{\eta\bar{\psi}}}}{\sqrt{2\eta\gamma + (\eta\gamma)^2}},$$

$$\ge \frac{\eta r}{20\sqrt{d}\sqrt{2\eta\gamma + (\eta\gamma)^2}} 2^{\log\left( 2\sqrt{\phi_{T_s}(\delta, F)} \frac{20\sqrt{d}\sqrt{\eta^2\gamma^2 + 2\eta\gamma}}{\eta r} \right)} > 2\sqrt{\phi_{T_s}(\delta, F)} > 2\sqrt{\phi(T_s, \delta)}.$$

Thus, at least one of $\|x_{T_s} - x_0\|$ and $\|x'_{T_s} - x_0\|$ is larger than $\sqrt{\phi(T_s, \delta)}$, a contradiction. Since the two sequences have the same distribution, it follows that with probability at least $1/3 - \frac{13T_s\delta}{T}$, $f(x_{T_s}) - f(x_0) \le -F$. $\qquad \square$

In the result above, we require an upper bound on the norm of $\phi_{T_s}(\delta, F)$ to hold (i.e. equation 48), which in turn necessitates an upper bound on $F$, the function value improvement we can expect to make. Below, we show how to choose $F$ to be as large as possible (up to constants and logarithmic factors) whilst still satisfying equation 48, assuming that $u, r$ and $\eta$ are chosen appropriately small such that the dominant term of $\|\phi_{T_s}(\delta, F)\|$ scales with $F$.

**Lemma 25.** *Consider choosing $F$ such that*

$$F = \frac{1}{2}\left(\frac{\bar{\psi}}{60c_9\iota\rho\log(T/\delta)}\right)^2 \frac{1}{\eta T_s t_f(\delta)\left(129 + 8c'^2\beta_1(\delta; F)\left(16(\text{lr}(CT^2/\delta))^2 + 1)\right)\right)}.$$

*Suppose $\eta \leq \min\left\{1, \frac{1}{t_f(\delta)}, \frac{1}{t_f\delta L}\right\}$. Suppose we pick $u$ and $r$ small enough such that*

$$u \leq \frac{r^{1/2}}{d\log(T/\delta)\rho^{1/2}}, \quad r^2 \leq \min\left\{\frac{F\bar{\psi}}{2\iota\log(T/\delta)\left(\frac{65c_5^2}{8} + 6c_1 + 1\right)}, \frac{F}{4c_6\log(2dT/\delta) + \frac{8c_7\iota}{\psi}}\right\}.$$

*Then, $N_{u,r}(T_s, \delta) \leq F$, and that*

$$4c_6\eta^2 T_s\log(2dT/\delta)r^2 + 4c_7\eta^2 T_s^2\rho^2u^4d^4\left(\log(T/\delta)\right)^4 \leq \eta T_s t_f(\delta)F.$$

*Suppose in addition $\eta$ is small enough so that*

$$32\eta^2(t_f(\delta))^2\epsilon^2 \leq \frac{1}{2}\left(\frac{\bar{\psi}}{60c_9\iota\rho\log(T/\delta)}\right)^2.$$

*Suppose also that $\bar{\psi} \leq 1$[8] and $\eta \leq \frac{m}{d}$, so that $T_s \geq \frac{\iota}{\eta\bar{\psi}} \geq \frac{d}{m}$. Then, the condition in Eq. (48) will be satisfied.*

*Proof.* We note that since $\frac{\iota}{\psi} \leq T_s \leq \frac{2\iota}{\psi}$, it follows by our choice of $r$ that $r$ also satisfies the condition

$$r^2 \leq \min\left\{\frac{F}{\eta T_s\log(T/\delta)\left(\frac{65c_5^2}{8} + 6c_1 + 1\right)}, \frac{F}{4c_6\log(2dT/\delta) + 4c_7\eta T_s}\right\}.$$

Hence, our choice of $\eta, u$ and $r$ satisfies the conditions in Lemma 23, and it follows then that

$$\phi_{T_s}(\delta, F) \leq \max\left\{128\eta T_s t_f(\delta)F, 32\eta^2(t_f(\delta))^2\epsilon^2\right\} + 8c'^2\beta_1(\delta; F)\eta t_f(\delta)\max\left\{\frac{16d}{m}(\text{lr}(CT^2/\delta))^2F, T_sF\right\} + T_s\eta t_f(\delta)F,$$

where $\beta_1(\delta; F)$ is as defined in Lemma 21.

The condition in Eq. (48) requires that

$$\phi_{T_s}(\delta, F) \leq \left(\frac{\bar{\psi}}{60c_9\iota\rho\log(T/\delta)}\right)^2.$$

By our choice of $\eta$ such that

$$32\eta^2(t_f(\delta))^2\epsilon^2 \leq \frac{1}{2}\left(\frac{\bar{\psi}}{60c_9\iota\rho\log(T/\delta)}\right)^2,$$

it suffices for us to show that

$$\frac{1}{2}\left(\frac{\bar{\psi}}{60c_9\iota\rho\log(T/\delta)}\right)^2 \geq 128\eta T_s t_f(\delta)F + 8c'^2\beta_1(\delta; F)\eta t_f(\delta)\max\left\{\frac{16d}{m}(\text{lr}(CT^2/\delta))^2F, T_sF\right\} + \eta T_s t_f(\delta)F$$

$$= 129\eta T_s t_f(\delta)F + 8c'^2\beta_1(\delta; F)\eta t_f(\delta)\max\left\{\frac{16d}{m}(\text{lr}(CT^2/\delta))^2F, T_sF\right\}.$$

---

[8]Without loss of generality, we may set $\bar{\psi} = 1$ if $f(\cdot)$ is $(\epsilon, \psi)$-strict saddle for any $\psi > 1$.

By our assumption, we know that $T_s \geq \frac{d}{m}$. Thus, further simplifying indicates that it suffices for us to show

$$\frac{1}{2}\left(\frac{\bar{\psi}}{60c_9\iota\rho\log(T/\delta)}\right)^2 \geq 129\eta T_s t_f(\delta)F + 8c'^2\beta_1(\delta; F)\eta t_f(\delta)\max\left\{16T_s(\mathrm{lr}(CT^2/\delta))^2 F, T_s F\right\}. \tag{51}$$

By choosing $F$ such that

$$F \leq \frac{1}{2}\left(\frac{\bar{\psi}}{60c_9\iota\rho\log(T/\delta)}\right)^2 \frac{1}{\eta T_s t_f(\delta)\left(129 + 8c'^2\beta_1(\delta; F)\left(16(\mathrm{lr}(CT^2/\delta))^2 + 1\right)\right)},$$

we see that Eq. (51) is satisfied.

$\square$

*Remark* 2. Suppose without loss of generality that $T_s = \frac{\iota}{\eta\bar{\psi}}$. Then, as a consequence of Lemma 25, we note that the amortized function value progress of decreasing function value by $F$ over $T_s$ iterations is

$$\frac{F}{T_s} = \frac{1}{2}\left(\frac{\bar{\psi}}{60c_9\iota\rho\log(T/\delta)}\right)^2 \frac{1}{\eta T_s^2 t_f(\delta)\left(129 + 8c'^2\beta_1(\delta; F)\left(16(\mathrm{lr}(CT^2/\delta))^2 + 1\right)\right)}$$

$$= \eta\frac{\bar{\psi}^4}{\rho^2}\left(\frac{1}{2\iota^2}\frac{1}{(60c_9\iota\log(T/\delta))^2\left(t_f(\delta)\right)\left(129 + 8c'^2\beta_1(\delta; F)\left(16(\mathrm{lr}(CT^2/\delta))^2 + 1\right)\right)}\right)$$

# F    PROVING THE MAIN RESULT (INFORMAL STATEMENT IN THEOREM 1, FULL STATEMENT IN THEOREM 2)

In this section, we prove our main result. First, we need an additional result (Lemma 26) showing that with high probability, we can bound the function value increase if a saddle appears within $t_f(\delta)$ iterations immediately after we have had $T_s$ iterations after the previous saddle. We note that such a bound is necessary because our earlier result upper bounding function increase in $\tau$ iterations (see Lemma 18) focused on the case where $\tau \geq t_f(\delta)$. Next, we state and prove Theorem 2, which is the precise version of Theorem 1 in the main text.

**Lemma 26** (Function change for small $\tau$). *Let $c_1 > 0, c_4 > 0, c_5 > 0, C_1 \geq 1$ be the absolute constants defined in the statements of the previous lemmas. Let $\delta \in (0, 1/e]$, and suppose $\tau < t_f(\delta)$.*

*Let $J$ denote the interval $\{0, 1 \ldots, \tau - 1\}$ where $\tau < t_f(\delta)$.*

*Suppose we choose $\eta$ such that*

$$\eta \leq \frac{1}{Lt_f(\delta)} \cdot \min\left\{\frac{\sqrt{m}}{8c_4(\mathrm{lr}(C_1 dmT/\delta))^{3/2}\sqrt{d}}, \frac{m}{128c_1(\mathrm{lr}(C_1 dmT/\delta))^3 d}\right\}. \tag{52}$$

*Suppose also we pick $u, r$ and $\eta$ as prescribed in the statement of Proposition 4.*

*Suppose that $\min_{t\in J}\|\nabla f(x_t)\| \leq \epsilon$. Then, on the event*

$$\mathcal{D}_\tau(\delta) := \mathcal{H}_{0,\tau}(\delta) \cap \left(\bigcap_{t=0}^{\tau-1} \mathcal{A}_t(\delta)\right) \cap \left(\bigcap_{t=0}^{\tau-1} \mathcal{G}_t(\delta)\right),$$

*we have the following upper bound on function value change:*

$$f(x_\tau) - f(x_0) \leq \frac{\eta}{4}\epsilon^2 + t_f(\delta)\eta u^4\rho^2 \cdot c_1 d^3 \left(\log\frac{T}{\delta}\right)^3 + t_f(\delta)L\eta^2 u^4\rho^2 \cdot c_1 d^4 \left(\log\frac{T}{\delta}\right)^4$$

$$+ \eta c_1 r^2(128 t_f(\delta) + \eta L)\log\frac{T}{\delta} + t_f(\delta)c_1 L\eta^2 r^2.$$

*Moreover, $\mathbb{P}(\mathcal{D}_\tau(\delta)) \geq 1 - \frac{(4t_f(\delta)+4)\delta}{T}$.*

*Proof.* Throughout the proof, we assume that the event $\mathcal{D}_\tau(\delta)$ holds.

Let $J$ denote $\{0, 1 \ldots, \tau - 1\}$ where $\tau < t_f(\delta)$. Then, $J$ belongs to one of the two following cases.

**Case 1**) (Gradient dominates noise): Recall that this means that for every $t \in J$, we have

$$\|\nabla f(x_t)\| > 8t_f(\delta)\eta L \left( \frac{u}{2} \left\| \frac{1}{m} \sum_{i=1}^{m} Z_{t,i} Z_{t,i}^\top \tilde{H}_{t,i} Z_{t,i} \right\| + \|Y_t\| \right).$$

By our choice of $\eta$ in Eq. (29), we can apply Lemma 16 to get

$$\min_{t \in J} \|\nabla f(x_t)\| \geq \frac{1}{4} \max_{t \in J} \|\nabla f(x_t)\|.$$

Thus by setting $\alpha = 128 t_f(\delta)$ in Eq. (3) and by choosing $\eta$ such that

$$\frac{c_1 L \eta^2 \chi^3 d}{m} \leq \frac{\eta}{\alpha} = \frac{\eta}{128 t_f(\delta)} \quad \Longleftrightarrow \quad \eta \leq \frac{m}{128 c_1 L t_f(\delta) d \chi^3},$$

it follows that

$$- \frac{3\eta}{4} \sum_{t \in J} \frac{1}{m} \sum_{i=1}^{m} |Z_{t,i}^\top \nabla f(x_t)|^2 + \left( \frac{\eta}{128 t_f(\delta)} + \frac{c_1 L \eta^2 \chi^3 d}{m} \right) \sum_{t \in J} \|\nabla f(x_t)\|^2$$

$$= - \frac{3\eta}{4} \sum_{t \in J} \frac{1}{m} \sum_{i=1}^{m} |Z_{t,i}^\top \nabla f(x_t)|^2 + \frac{\eta}{64 t_f(\delta)} \sum_{t \in J} \|\nabla f(x_t)\|^2$$

$$\leq \frac{\eta}{64 t_f(\delta)} \sum_{t \in J} \|\nabla f(x_t)\|^2$$

$$\leq \frac{\eta}{64 t_f(\delta)} \sum_{t \in J} \max_{t \in J} \|\nabla f(x_t)\|^2$$

$$\leq \frac{16\eta}{64 t_f(\delta)} \sum_{t \in J} \min_{t \in J} \|\nabla f(x_t)\|^2 \leq \frac{\eta}{4} \min_{t \in J} \|\nabla f(x_t)\|^2 \leq \frac{\eta}{4} \epsilon^2, \tag{53}$$

where the final bound holds since we assumed $\min_{t \in J} \|\nabla f(x_t)\| \leq \epsilon$.

**Case 2**) (Gradient does not dominate noise): there exists some $t \in J$ such that

$$\|\nabla f(x_t)\| \leq 8t_f(\delta)\eta L \left( \frac{u}{2} \left\| \frac{1}{m} \sum_{i=1}^{m} Z_{t,i} Z_{t,i}^\top \tilde{H}_{t,i} Z_{t,i} \right\| + \|Y_t\| \right).$$

By our choice of $\eta$ in Eq. (29), we can apply Lemma 17 to get

$$\|\nabla f(x_t)\| \leq c_5 t_f(\delta)\eta L \left( u^2 d^2 \rho \left( \log \frac{T}{\delta} \right)^2 + \sqrt{1 + \frac{\log(T/\delta)}{d}} r \right) \qquad \forall t \in J.$$

Note that, by our choices of the parameters $\eta, u, r$, it can be shown that

$$c_5 t_f(\delta)\eta L \left( u^2 d^2 \rho \left( \log \frac{T}{\delta} \right)^2 + \sqrt{1 + \frac{\log(T/\delta)}{d}} r \right) < \epsilon,$$

Hence, by setting $\alpha = 128 t_f(\delta)$ in Eq. (3) and choosing $\eta$ such that

$$\frac{c_1 L \eta^2 \chi^3 d}{m} \leq \frac{\eta}{\alpha} = \frac{\eta}{128 t_f(\delta)},$$

it follows that

$$\left( \frac{\eta}{128 t_f(\delta)} + \frac{c_1 L \eta^2 \chi^3 d}{m} \right) \sum_{t \in J} \|\nabla f(x_t)\|^2$$

$$\leq \frac{\eta}{64 t_f(\delta)} \sum_{t \in J} \left( c_5 t_f(\delta) \eta L \left( u^2 d^2 \rho \left( \log \frac{T}{\delta} \right)^2 + \sqrt{1 + \frac{\log(T/\delta)}{d}} r \right) \right)^2$$

$$\leq \frac{\eta}{64 t_f(\delta)} \sum_{t \in J} \epsilon^2$$

$$\leq \frac{\eta}{64} \epsilon^2 \tag{54}$$

Combining both cases above (Eq. (53) and Eq. (54)), we see that for the choice $\alpha = 128 t_f(\delta)$, the bound

$$-\frac{3\eta}{4} \sum_{t \in J} \frac{1}{m} \sum_{i=1}^{m} \left| Z_{t,i}^\top \nabla f(x_t) \right|^2 + \left( \frac{\eta}{128 t_f(\delta)} + \frac{c_1 L \eta^2 \chi^3 d}{m} \right) \sum_{t \in J} \| \nabla f(x_t) \|^2 \leq \frac{\eta}{4} \epsilon^2 \tag{55}$$

always holds.

Recall by Eq. (3) that we have

$$f(x_\tau) - f(x_0) \leq -\frac{3\eta}{4} \sum_{t=0}^{\tau-1} \frac{1}{m} \sum_{i=1}^{m} \left| Z_{t,i}^\top \nabla f(x_t) \right|^2 + \left( \frac{\eta}{\alpha} + \frac{c_1 L \eta^2 \chi^3 d}{m} \right) \sum_{t=0}^{\tau-1} \| \nabla f(x_t) \|^2$$
$$+ \tau \eta u^4 \rho^2 \cdot c_1 d^3 \left( \log \frac{T}{\delta} \right)^3 + \tau L \eta^2 u^4 \rho^2 \cdot c_1 d^4 \left( \log \frac{T}{\delta} \right)^4$$
$$+ \eta c_1 r^2 (\alpha + \eta L) \log \frac{T}{\delta} + \tau c_1 L \eta^2 r^2.$$

By plugging in Eq. (55) above, as well as the choice $\alpha = 128 t_f(\delta)$, we see that

$$f(x_\tau) - f(x_0) \leq \frac{\eta}{4} \epsilon^2 + t_f(\delta) \eta u^4 \rho^2 \cdot c_1 d^3 \left( \log \frac{T}{\delta} \right)^3 + t_f(\delta) L \eta^2 u^4 \rho^2 \cdot c_1 d^4 \left( \log \frac{T}{\delta} \right)^4$$
$$+ \eta c_1 r^2 (128 t_f(\delta) + \eta L) \log \frac{T}{\delta} + t_f(\delta) c_1 L \eta^2 r^2.$$

We can now complete our proof by using the union bound (suppressing the dependence of some of the events on $\delta$ for notational simplicity) to derive

$$\mathbb{P}(\mathcal{D}_\tau^c) \leq \mathbb{P}(\mathcal{H}_\tau^c) + \sum_{t=0}^{\tau-1} \mathbb{P}(\mathcal{A}_t^c) + \sum_{t=0}^{\tau-1} \mathbb{P}(\mathcal{G}_t^c)$$
$$\leq \frac{(\tau+4)\delta}{T} + \frac{\tau}{T}\delta + 2\frac{\tau}{T}\delta \leq \frac{(4t_f(\delta)+4)}{T}\delta \qquad \square$$

Armed with Proposition 5 and Lemma 25, we are now ready to show for $T$ sufficiently large, with high probability, there can be no more than $T/4$ $\epsilon$-saddle points. Combined with Proposition 4, this yields the following result.

**Theorem 2.** *Suppose we pick $u, r, \eta$ such that they satisfy the conditions in Proposition 5 and Lemma 25. Suppose $F$ is chosen as prescribed in Lemma 25. Suppose that $\bar{\psi} \leq 1$, so that $T_s \geq \frac{\iota}{\eta \bar{\psi}} \geq \frac{d}{mL}$[9]. Suppose we pick $T_s$ as prescribed in Proposition 5. Suppose in addition we pick $r$ such that*

$$r^2 \leq \min \left\{ \frac{\epsilon^2}{4(130 c_1 t_f(\delta) + c_1 \log(T/\delta) + c_1)}, \frac{F\bar{\psi}}{80\iota \log(T/\delta) \left( \frac{65 c_5^2}{8} + 132 c_1 + 1 \right)} \right\}.$$

*Suppose also that we choose $\eta$ such that*

---

[9]Recall we focus on the case $\bar{\psi} \leq L$, since otherwise, by the $L$-Lipschitz assumption, $\lambda_{\min}(\nabla^2 f(x)) \geq -L$ for all $x \in \mathbb{R}^d$, i.e. $\epsilon$-first order stationary points are also $\epsilon$-second order stationary points.

$$\eta \le \frac{0.1}{2\epsilon^2} \frac{\bar{\psi}}{2\iota} \frac{1}{2} \left( \frac{\bar{\psi}}{60 c_9 \iota \rho \log(T/\delta)} \right)^2 \frac{1}{t_f(\delta) \left(129 + 8c'^2 \beta_1(\delta; F)\right) \left(16(\mathrm{lr}(CT^2/\delta))^2 + 1)\right)}$$

*Suppose*

$$T \ge \left\{ \frac{256 t_f(\delta) \left((f(x_0) - f^*) + \epsilon^2/L)\right)}{\eta \epsilon^2}, \frac{\varphi \rho^2 (f(x_0) - f^*)}{\eta \bar{\psi}^4}, 256 \lceil \frac{\iota}{\eta \bar{\psi}} \rceil, 256 t_f(\delta), 1024 \right\}, \quad (56)$$

*where*

$$\varphi := 20 \left( 2\iota^2 (60 c_9 \iota \log(T/\delta))^2 \left(t_f(\delta)\right) \left(129 + 8c'^2 \beta_1(\delta; F)\right) \left(16(\mathrm{lr}(CT^2/\delta))^2 + 1)\right) \right).$$

*Then, with probability at least $1 - 22\delta$, there are at least $T/2$ $\epsilon$-approximate second order stationary points.*

*Proof.* Consider defining the following sequence of stopping times:

$$\tau_1 = \inf_t \{ t \le T : \|\nabla f(x_t)\| < \epsilon, \lambda_{\min}(\nabla^2 f(x_t)) \le -\sqrt{\rho\epsilon} \},$$

$$\tau_{i+1} = \inf_t \{ t \le T : t > \tau_i + T_s, \|\nabla f(x_t)\| < \epsilon, \lambda_{\min}(\nabla^2 f(x_t)) \le -\sqrt{\rho\epsilon} \}, \quad \forall 1 \le i \le \lfloor T/T_s \rfloor.$$

$\square$

We note that if $\tau_i = T$, then $\tau_j = T$ for any $j > i$. Let $N_s$ denote the (random) number of saddle points encountered in $T$ iterations.

We observe that we can decompose the function change as

$$
\begin{aligned}
&f(x_T) - f(x_0) \\
&= (f(x_{\tau_{N_s}}) - f(x_0)) + (f(x_T) - f(x_{\tau_{N_s}})) \\
&= (f(x_{\tau_1}) - f(x_0)) + \sum_{i=1}^{N_s} (f(x_{\tau_i + T_s}) - f(x_{\tau_i})) + \sum_{i=1}^{N_s - 1} \left( f(x_{\tau_{i+1}}) - f(x_{\tau_i + T_s}) \right) + (f(x_T) - f(x_{\tau_{N_s}})) \\
&= \underbrace{\sum_{i=1}^{N_s} (f(x_{\tau_i + T_s}) - f(x_{\tau_i}))}_{U_1} + \underbrace{(f(x_{\tau_1}) - f(x_0)) + \sum_{i=1}^{N_s - 1} \left( f(x_{\tau_{i+1}}) - f(x_{\tau_i + T_s}) \right) + (f(x_T) - f(x_{\tau_{N_s}}))}_{U_2}.
\end{aligned}
$$

We first consider $U_1$. Letting $x_j := x_T$ for any $j \ge T$, we have that

$$\sum_{i=1}^{N_s} f(x_{\tau_i + T_s}) - f(x_{\tau_i}) = \sum_{i=1}^{\lfloor T/T_s \rfloor} (f(x_{\tau_i + T_s}) - f(x_{\tau_i})) \mathbb{1}_{\tau_i < T}$$

Now, by Eq. (30), observe that with probability at least $1 - \frac{(5T_s + 4)\delta}{T} \ge 1 - \frac{6T_s \delta}{T}$ (note $T_s \ge 4$), for any $1 \le i \le T/T_s$, we have that

$$
\begin{aligned}
(f(x_{\tau_i + T_s}) - f(x_{\tau_i})) \mathbb{1}_{\tau_i < T} &\le \tau \frac{c_5^2}{64} \eta^3 t_f(\delta)^2 L^2 \left( u^2 d^2 \rho \left( \log \frac{T}{\delta} \right)^2 + \sqrt{2\log(T/\delta)} r \right)^2 \\
&\quad + \tau \eta u^4 \rho^2 \cdot c_1 d^3 \left( \log \frac{T}{\delta} \right)^3 + \tau L \eta^2 u^4 \rho^2 \cdot c_1 d^4 \left( \log \frac{T}{\delta} \right)^4 \\
&\quad + \eta c_1 r^2 (128 t_f(\delta) + \eta L) \log \frac{T}{\delta} + \tau c_1 L \eta^2 r^2 \\
&:= M_{u,r,T_s}.
\end{aligned}
$$

Suppose we pick $u, r$ such that $M_{u,r,T_s} \leq 0.1F$. Recall from Proposition 5 that with probability at least $1/3 - \frac{13T_s\delta}{T}$, $(f(x_{\tau_i+T_s}) - f(x_{\tau_i}))1_{\tau_i<T} \leq -F$. Choosing $\delta$ such that $1/3 - \frac{13T_s\delta}{T} \geq 0.3$, and letting $\mu = 0.1F$, we note that $|-F + \mu| = 0.9F \geq \frac{0.7}{0.3}0.2F \geq \frac{0.7}{0.3}(M_{u,r,T_s} + \mu)$.

Now, let $\mathcal{E}_{\tau_i}$ denote the bad event on which

neither $(f(x_{\tau_i+T_s}) - f(x_{\tau_i}))1_{\tau_i<T} \leq -F$, nor $(f(x_{\tau_i+T_s}) - f(x_{\tau_i}))1_{\tau_i<T} \leq M_{u,r,T_s} \leq 0.1F$.

We know that $\mathcal{E}_{\tau_i}$ has probability at most $\frac{6T_s\delta}{T}$. Let $\mathcal{E}_\tau := \cup_{i=1}^{\lfloor T/T_s \rfloor}\mathcal{E}_{\tau_i}$, such that $\mathbb{P}(\mathcal{E}_\tau) \leq 6\delta$. Then, by applying the weakened supermartingale inequality in Proposition 3, we have

$$\mathbb{P}\left(\sum_{i=1}^{T/T_s} (f(x_{\tau_i+T_s}) - f(x_{\tau_i}))1_{\tau_i<T} \geq -N_s0.9F + s\right) \leq \mathbb{E}\left[\exp\left(-\frac{s^2}{4N_sF^2}\right)\right] + \mathbb{P}(\mathcal{E}_\tau) \leq \exp\left(-\frac{s^2}{4(T/T_s)F^2}\right) + 6\delta.$$

Now, pick $s = 2F\sqrt{\sqrt{\log(1/\delta)}T/T_s}$, then

$$\mathbb{P}\left(\sum_{i=1}^{T/T_s} (f(x_{\tau_i+T_s}) - f(x_{\tau_i}))1_{\tau_i<T} \geq -N_s0.9F + 2F\sqrt{\sqrt{\log(1/\delta)}T/T_s}\right) \leq 7\delta.$$

Note that supposing for contradiction that there are at least $T/4$ saddles, we must then have that $N_s \geq T/(4T_s)$, such that

$$-N_s0.9F + 2F\sqrt{\sqrt{\log(1/\delta)}T/T_s} \leq F(-0.9T/(4T_s) + (2\sqrt{\sqrt{\log(1/\delta)}T/T_s})) \leq F(-0.1T/T_s),$$

where we may ensure the last inequality by picking $T/T_s$ such that

$$T/T_s \geq \left(\frac{2}{0.125}\right)^2\sqrt{\log(1/\delta)} = 256\sqrt{\log(1/\delta)}.$$

Note that our choice of $T$ ensures this.

Thus, with probability at least $1 - 7\delta$,

$$U_1 = \sum_{i=1}^{T/T_s} (f(x_{\tau_i+T_s}) - f(x_{\tau_i}))1_{\tau_i<T} \leq -(0.1T/T_s)F.$$

Next, we bound the summand $U_2$. Recall that

$$U_2 = (f(x_{\tau_1}) - f(x_0)) + \sum_{i=1}^{N_s-1} \left(f(x_{\tau_{i+1}}) - f(x_{\tau_i+T_s})\right).$$

Without loss of generality, we may analyze each of the summands $f(x_{\tau_{i+1}}) - f(x_{\tau_i+T_s})$ in the same way as we treat $(f(x_{\tau_1}) - f(x_0))$. Let us then consider the summand $f(x_{\tau_1}) - f(x_0)$. There are two cases to consider.

1. The first is when $\tau_1 < t_f(\delta)$. In this case, since we know that $\|\nabla f(x_{\tau_1})\| \leq \epsilon$ (as $x_{\tau_1}$ is an $\epsilon$-saddle point), it follows by Lemma 26 that

$$f(x_{\tau_1}) - f(x_0) \leq \frac{\eta}{4}\epsilon^2 + t_f(\delta)\eta u^4\rho^2 \cdot c_1d^3\left(\log\frac{T}{\delta}\right)^3 + t_f(\delta)L\eta^2u^4\rho^2 \cdot c_1d^4\left(\log\frac{T}{\delta}\right)^4$$
$$+ \eta c_1r^2(128t_f(\delta) + \eta L)\log\frac{T}{\delta} + t_f(\delta)c_1L\eta^2r^2$$

with probability at least $1 - \frac{(4t_f(\delta)+4)\delta}{T}$.

2. The second case is when $\tau_1 \geq t_f(\delta)$. In this case, by Lemma 18, we have that

$$f(x_{\tau_1}) - f(x_0) \leq \tau_1\frac{c_5^2}{64}\eta^3t_f(\delta)^2L^2\left(u^2d^2\rho\left(\log\frac{T}{\delta}\right)^2 + \sqrt{2\log(T/\delta)}r\right)^2$$

$$+ \tau_1 \eta u^4 \rho^2 \cdot c_1 d^3 \left( \log \frac{T}{\delta} \right)^3 + \tau_1 L \eta^2 u^4 \rho^2 \cdot c_1 d^4 \left( \log \frac{T}{\delta} \right)^4$$

$$+ \eta c_1 r^2 (128 t_f(\delta) + \eta L) \log \frac{T}{\delta} + \tau_1 c_1 L \eta^2 r^2.$$

with probability at least $1 - \frac{(5\tau_1 + 4)\delta}{T}$.

By our choice of $u$, we know that

$$t_f(\delta) \eta u^4 \rho^2 \cdot c_1 d^3 \left( \log \frac{T}{\delta} \right)^3 + t_f(\delta) L \eta^2 u^4 \rho^2 \cdot c_1 d^4 \left( \log \frac{T}{\delta} \right)^4 + \eta c_1 r^2 (128 t_f(\delta) + \eta L) \log \frac{T}{\delta} + t_f(\delta) c_1 L \eta^2 r^2$$

$$\leq t_f(\delta) r^2 c_1 + t_f(\delta) r^2 c_1 + c_1 r^2 (128 t_f(\delta) + 1) \log(T/\delta) + c_1 r^2$$

$$= r^2 (130 c_1 t_f(\delta) + c_1 \log(T/\delta) + c_1).$$

Hence, by picking $r$ such that

$$r \leq \frac{\epsilon^2}{4(130 c_1 t_f(\delta) + c_1 \log(T/\delta) + c_1)},$$

it follows that

$$\frac{\eta \epsilon^2}{4} \geq t_f(\delta) \eta u^4 \rho^2 \cdot c_1 d^3 \left( \log \frac{T}{\delta} \right)^3 + t_f(\delta) L \eta^2 u^4 \rho^2 \cdot c_1 d^4 \left( \log \frac{T}{\delta} \right)^4$$

$$+ \eta c_1 r^2 (128 t_f(\delta) + \eta L) \log \frac{T}{\delta} + t_f(\delta) c_1 L \eta^2 r^2.$$

Then, if $\tau_1 < t_f(\delta)$, with probability at least $1 - \frac{(5 t_f(\delta) + 4)}{\delta}$,

$$f(x_{\tau_1}) - f(x_0) \leq \frac{\eta \epsilon^2}{2}.$$

Suppose also that we pick $r$ such that

$$r^2 \leq \frac{F \sqrt{\rho \epsilon}}{80 \iota \log(T/\delta) \left( \frac{65 c_5^2}{8} + 132 c_1 + 1 \right)} \leq \frac{F}{40 \eta T_s \log(T/\delta) \left( \frac{65 c_5^2}{8} + 132 c_1 + 1 \right)}.$$

Then, it can be verified that

$$\frac{F}{40} \frac{T}{T_s} \geq T \frac{c_5^2}{64} \eta^3 t_f(\delta)^2 L^2 \left( u^2 d^2 \rho \left( \log \frac{T}{\delta} \right)^2 + \sqrt{2 \log(T/\delta)} r \right)^2$$

$$+ T \eta u^4 \rho^2 \cdot c_1 d^3 \left( \log \frac{T}{\delta} \right)^3 + T L \eta^2 u^4 \rho^2 \cdot c_1 d^4 \left( \log \frac{T}{\delta} \right)^4$$

$$+ \frac{T}{T_s} \eta c_1 r^2 (128 t_f(\delta) + \eta L) \log \frac{T}{\delta} + T c_1 L \eta^2 r^2.$$

Then, by a union bound, it follows that with probability at least $1 - 9\delta$,

$$U_2 = (f(x_{\tau_1}) - f(x_0)) + \sum_{i=1}^{N_s - 1} \left( f(x_{\tau_{i+1}}) - f(x_{\tau_i + T_s}) \right)$$

$$\leq \frac{T}{T_s} \frac{\eta \epsilon^2}{2} + \frac{F}{40} \frac{T}{T_s}$$

Therefore, by the union bound, with probability at least $1 - 16\delta$,

$$f(x_{\tau_{N_s}}) - f(x_0) = U_1 + U_2 \leq \frac{T}{T_s} \left( -0.1 F + \eta \epsilon^2 / 2 + \frac{F}{40} \right)$$

By recalling our choice of $F$ in Lemma 25, by choosing $\eta$ such that

$$\eta \leq \frac{0.1}{2\epsilon^2} \frac{\bar{\psi}}{2\iota} \frac{1}{2} \left( \frac{\bar{\psi}}{60 c_9 \iota \rho \log(T/\delta)} \right)^2 \frac{1}{t_f(\delta) \left( 129 + 8 c'^2 \beta_1(\delta; F) \right) \left( 16 (\mathrm{lr}(CT^2/\delta))^2 + 1 \right)}$$

$$\leq \frac{0.1}{2\epsilon^2} \frac{1}{2} \left( \frac{\sqrt{\rho\epsilon}}{60c_9\iota\rho \log(T/\delta)} \right)^2 \frac{1}{\eta T_s t_f(\delta) \left( 129 + 8c'^2 \beta_1(\delta; F) \left( 16(\mathrm{lr}(CT^2/\delta))^2 + 1 \right) \right)} = \frac{0.1F}{2\epsilon^2},$$

it follows that with probability at least $1 - 16\delta$,

$$f(x_{\tau_{N_s}}) - f(x_0) = U_1 + U_2$$
$$\leq \frac{T}{T_s} \left( -0.1F + \eta\epsilon^2/2 + \frac{F}{40} \right)$$
$$\leq \frac{T}{T_s}(-0.1F + 0.1F/4 + 0.1F/4) = \frac{T}{T_s}(-0.05F).$$

Choose $T$ such that

$$-(0.05T/T_s)F \leq -(f(x_0) - f^*) \iff T \geq \frac{20T_s(f(x_0) - f^*)}{F} \geq \frac{\varphi\rho^2 (f(x_0) - f^*)}{\eta\bar{\psi}^4}$$

yields a contradiction, where

$$\varphi := 20 \left( 2\iota^2 (60c_9\iota \log(T/\delta))^2 (t_f(\delta)) \left( 129 + 8c'^2 \beta_1(\delta; F) \left( 16(\mathrm{lr}(CT^2/\delta))^2 + 1 \right) \right) \right)$$

Hence, with probability at least $1 - 16\delta$, there cannot be more than $T/4$ saddle points. In addition, with probability at least $1 - 6\delta$, by Proposition 4, there cannot be more than $T/4$ iterates with $\|\nabla f(x_t)\| \geq \epsilon$. Hence, with probability at least $1 - 22\delta$, there are at least $T/2$ $\epsilon$-approximate second order stationary points.

## G    SIMULATIONS

We test the performance of our proposed algorithm with two-point estimators (ZOPGD-2pt) against existing zeroth-order benchmarks using the *octopus function* (proposed in Du et al. (2017)) of varying dimensions. It is known that the octopus function defined on $\mathbb{R}^d$, which chains $d$ saddle points sequentially, takes exponential (in $d$) time for exact gradient descent to escape; it has thus emerged as a popular benchmark to evaluate and compare the performance of algorithms that seek to escape saddle points. In our experiments, we compare the performance of our two-point estimator algorithm (ZOPGD-2pt) with PAGD (Algorithm 1 in Vlatakis-Gkaragkounis et al. (2019)) and ZO-GD-NCF (see Zhang et al. (2022)), which are the only two existing zeroth-order algorithms that have (a) a $\tilde{O}(d/\epsilon^2)$ sample complexity for escaping saddle points (with the latter algorithm yielding the tightest bounds), and (b) performed the best empirically on escaping saddle points (see the simulation results in Zhang et al. (2022)). We note that both PAGD and ZO-GD-NCF have to use $2d$ function evaluations per iteration to estimate the gradient while our algorithm only needs to use 2 function evaluations. In our plots, we plot the function value against the number of function evaluations. For completeness, we also plot the performance of exact gradient descent (normalized such that its $x$-axis is also the number of function queries).

We tested the algorithms for $d = 10$ and $d = 30$. To account for the stochasticity in the algorithms, for each algorithm, we computed the average and standard deviation over 30 trials, and plotted the mean trajectory with an additional band that represents 1.5 times the standard deviation. For our algorithmś hyperparameters, we picked

$$\eta = \frac{1}{4dL}, u = 10^{-2}, r = 0.05, m = 1( \text{ i.e. two-point estimator}) \tag{57}$$

For PAGD, we used the hyperparameters listed in their paper, and for ZO-GD-NCF, we used the code from their Neurips submission. We note in particular that both methods used the step-size $\frac{1}{4L}$. For initialization, we chose a random $x_0$ near the saddle point at the origin, drawn from $N(0, 10^{-3}I_{d\times d})^{10}$ (fixed for all trials and all algorithms).

As we can see in Fig. 1, in both cases, our algorithm reaches the global minimum of the octopus function in significantly fewer function evaluations than PAGD and ZO-GD-NCF (approximately 2.5

---

[10]Using the random seed in our code, we note that $\|\nabla f(x_0)\| = 0.011$ for $d = 10$ and $\|\nabla f(x_0)\| = 0.030$ for $d = 30$.

times faster than ZO-GD-NCF, and approximately 3 times faster than PAGD), despite our algorithm only using 2 function evaluations per iteration compared to $2d$ function evaluations per iteration for both PAGD and ZO-GD-NCF. As a sanity check, we note that the number of function evaluations required for PAGD and ZO-GD-NCF to reach the global minimum approximately matches that in Figure 1 of Zhang et al. (2022); here the correspondence is only approximate since Zhang et al. (2022) only plots one trial while we compute the mean and standard deviation of 30 trials.

This result suggests that in addition to the theoretical convergence guarantees, there might also be empirical benefits to using two-point estimators versus existing $2d$-point estimators in the zeroth-order escaping saddle point literature.

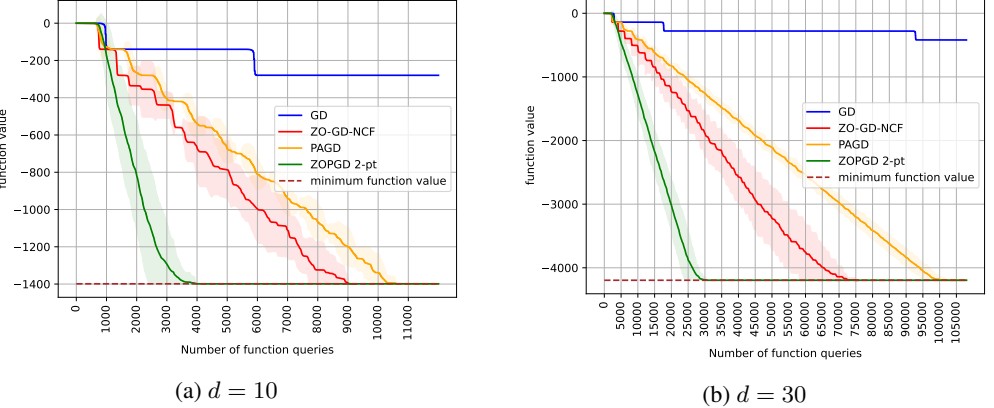

(a) $d = 10$

(b) $d = 30$

Figure 1: Performance on toy octopus function, with $\tau = e, L = e, \gamma = 1$ (Here, $\tau, L, \gamma$ are parameters determining the properties of $f$. Our parameter choice is consistent with that in Zhang et al. (2022). See Du et al. (2017) for details about the definitions of $\tau, L$ and $\gamma$.).

