# OpenReview forum: "Escaping saddle points in zeroth-order optimization:  two function evaluations suffice"
_ICLR.cc/2023/Conference — Submitted to ICLR 2023_

### Official Review · Reviewer_ZbmT · 2022-10-20

**Confidence:** 4
**Correctness:** 4
**Technical Novelty And Significance:** 4
**Empirical Novelty And Significance:** 3
**Recommendation:** 8

**Clarity, Quality, Novelty And Reproducibility:**

From my perspective, the clarity and quality of this paper is in general good, though the paper still has space to improve as I mentioned above. Novelty is excellent -- this is the first paper using fewer than d function evaluations for escaping from saddle points using zeroth-order methods. Reproducibility is not applied because this work is purely theoretical.

**Strength And Weaknesses:**

From my perspective, the most notable strength of this paper is that it is the first work on escaping from saddle points that can use fewer than d evaluation queries in each iteration. This is very much novel because first-order methods and previous zeroth-order methods essentially all use (approximate) gradient information, and it is impressive that escaping saddle points can be achieved by using few than d evaluation queries. It’s also veery nice that there is an explicit relationship between the number of function evaluations per iteration and the overall iteration complexity, expressed by m.

Nevertheless, the paper may still have space to improve from the following aspects:

- I feel that the authors should probably give more explanations about why the current method can only achieve 1/eps^2.5 in terms of eps, and why it cannot reach 1/eps^2 (GD) or 1/eps^1.75 (AGD) at the moment. In particular, discussions about how this 2.5 is formed and the difference compared to those exponents in GD-based methods will be very helpful. Is it mainly due to having difficulty in the large-gradient scenario or when we are near saddle points (improve or localize)?

- There is no numerical experiment in this paper. I think writing a code for Algorithm 1 should not be difficult since it simply takes Gaussian samples and there is only 2m evaluations in each iteration. The paper would be stronger if there are numerical results that corroborate the theoretical findings.

- A typo throughout the paper: In a few places, zero-order should be zeroth-order (just imagine that we never say one-order but always first-order).

At last, I have a suggestion to the authors: It seems that the authors are mainly treating the 2m evaluations as m batches of two-point evaluations. Is it possible to consider fewer batches but more points in each batch? The observation is that Eq. (1) in Definition 3 is simply a first-order central-difference method. In numerical analysis, higher-order central difference formulae have been studied, see for instance Li https://www.sciencedirect.com/science/article/pii/S0377042704006454?via%3Dihub. It would be of general interest to discuss whether using a higher-order central difference formula could further improve the results.

**Summary Of The Paper:**

This paper studies the problem of escaping from saddle points, a basic problem in nonconvex optimization. Specifically, this paper considers function evaluations, i.e., zeroth-order inputs, and furthermore only 2m function evaluations are allowed in each iteration. Under such a setting, the authors give an algorithm with iteration complexity ~O(d/eps^2.5*m).

**Summary Of The Review:**

Overall, I think this is a novel work in studying zeroth-order methods for escaping from saddle points, and it could be nice to see this paper at ICLR 2023. Nevertheless, since the result has relatively large power of 1/eps and numerical experiments are absent, it also makes sense for this work to improve further and save for future conferences/journals.

---

> ### Author Response · Authors · 2022-11-17
> **Response to Reviewer ZbmT**
>
> We thank the reviewer for their appreciation of our paper. We address their questions and concerns below.
>
> **More intuition on why current method can only achieve $1/\epsilon^{2.5}$.** This is a great question. The key reason is the following; for concreteness suppose $m = 1$, i.e. we are working with a two-point estimator. As seen in Equation 50 in the Appendix, to make the saddle point escape part of the algorithm work, the step size $\eta$ has to satisfy $\eta \leq \tilde{O}\left(\frac{\bar{\psi}}{dL^2} \right),$ where
> $$\\bar{\\psi} := \psi  \mbox{ if } f(\cdot) \mbox{ is }(\epsilon,\psi)\mbox{-strict saddle}, \mbox{ and }\bar{\psi} := \sqrt{\rho \epsilon}\mbox{ if otherwise.}$$
> (See Definition 3 in our revision, or [our response to Reviewer QY7d](https://openreview.net/forum?id=5d_yTyTj646&noteId=kdUlwwQkAue), for a definition of $(\epsilon,\psi)$-strict saddles). Essentially, $\bar{\psi}$ is a lower bound on the size of the minimum eigenvalue of the Hessian near all $\epsilon$-saddle points. Satisfying equation 50 is essential in order to ensure that the quantity $W_{g_0}(t+1)$ (corresponding to zeroth-order noise) in the decomposition of the difference of the coupled sequence in Lemma 3 of our proof remains bounded. This then allows us to show that the contribution from the perturbation term $W_p(t+1)$ can be shown to dominate the evolution, leading to an exponential growth in the difference of the coupled sequence (if little function progress has been made and both sequences remain close to the saddle point), creating a contradiction (since if the sequences are bounded, then the difference of the coupled sequences should also be small) which implies that (some sufficient amount of) function progress must be made near saddle points (with constant probability).
>
> In fact, as we can see from the conditions for Proposition 5 (function progress during large gradients), during the large gradient phase, we only need $\eta = \tilde{O}(\frac{1}{dL})$, i.e. no dependence on $\bar{\psi}$. The requirement $\eta \leq \tilde{O}\left(\frac{1}{d \bar{\psi}L^2} \right)$ is only needed for the escape saddle point part of the algorithm. Due to the condition $\eta \leq \tilde{O}\left(\frac{1}{d \bar{\psi}L^2} \right)$ imposed by Equation 50, since our algorithm makes function value progress on average of $\eta \epsilon^2$ during the large gradient iterations (see Proposition 5 in our Appendix), the overall sample complexity is $T = \tilde{O}\left(\frac{f(x_0) - f^*}{\eta \epsilon^2}\right) = \tilde{O}\left(\frac{d(f(x_0) - f^*)}{ \bar{\psi}\epsilon^2}\right)$, where the denominator is equal to $\sqrt{\rho} \epsilon^{2.5}$ in the general case when $f(\cdot)$ is not $(\epsilon,\psi)$-strict saddle for any $\psi > \sqrt{\rho \epsilon}$.
>
> In brief, the extra $\sqrt{\rho \epsilon}$ in the denominator of the sample complexity is a consequence of a specific part of our proof in the escape saddle point part of the algorithm rather than the function decrease part of the argument. It is not clear to us if this can be improved or is a fundamental limitation of two-point zeroth-order escape saddle point methods.
>
>
> **"zero-order vs zeroth order"** We agree with the reviewer that zeroth-order is more grammatical. We will consistently use zeroth-order.
>
> **no numerical experiment.** We have now included numerical experiments in our revision, which demonstrates the advantages of our algorithm over existing state-of-the-art zeroth-order escape saddle point algorithms. We believe this gives more empirical support for the importance of considering two-point estimators. For a more complete discussion, we refer the reviewer to [our response here to Reviewer nRGh](https://openreview.net/forum?id=5d_yTyTj646&noteId=KODtxzIzwq).
>
> **Is it possible to consider fewer batches but more points in each batch?** We thank the reviewer for this interesting suggestion. While it is known that for stochastic convex zeroth-order optimization, two-point estimators suffice to attain the optimal rate (see [3]), it is possible that using a higher-order central difference formula may improve the sample complexity for the saddle point escape problem we study. While it is beyond the scope of our current work, it does represent an interesting future research direction.
>
>
> [1]: Du, S. S., Jin, C., Lee, J. D., Jordan, M. I., Singh, A., and Poczos, B. (2017). Gradient descent
> can take exponential time to escape saddle points. In Advances in neural information processing
> systems, pages 1067–1077.
>
> [2]: Vlatakis-Gkaragkounis, E.-V., Flokas, L., and Piliouras, G. (2019). Efficiently avoiding saddle points
> with zero order methods: No gradients required. Advances in Neural Information Processing
> Systems, 32.
>
> [3]: Duchi, J. C., Jordan, M. I., Wainwright, M. J., and Wibisono, A. (2015). Optimal rates for zero-order
> convex optimization: The power of two function evaluations. IEEE Transactions on Information
> Theory, 61(5):2788–2806.

---

### Official Review · Reviewer_nRGh · 2022-10-23

**Confidence:** 3
**Correctness:** 2
**Technical Novelty And Significance:** 3
**Empirical Novelty And Significance:** Not applicable
**Recommendation:** 6

**Clarity, Quality, Novelty And Reproducibility:**

**Clarity:**

The overall structure and main contribution of this paper look very clear.

**(My second major concern)** I scanned Section 3 and felt it very hard to fully understand the intuition. I believe that the technique is novel and correct, and that the authors have devoted much time and energy to outline the proof intuition. Is it possible to write 1-2 paragraphs at the beginning of Sections 3.1 and 3.2 to briefly sketch the proof with no equation or only a few core equations, so that we can more easily understand how the function value decrease OR $||\nabla f(x_t)||$, $\lambda_{\min}[\nabla^2 f(x_t)]$ are bounded for large and small gradients? I think that's perhaps not easy but worth trying.

Also, what is $\tau'$ in Lemma 1? It's recommended to define each new notion and notation at its first appearance.

The citation of [1] misses "In Advances in neural information processing systems".

[1] Vlatakis-Gkaragkounis, E. V., Flokas, L., and Piliouras, G. (2019). Efficiently avoiding saddle points with zero order methods: No gradients required. In Advances in neural information processing systems, 32.

**Quality:**

**(My major concern)** The advantage of the proposed algorithm in only $2m$ (batchsize) function evaluations per iteration is theoretically proved. I think this is probably a practical advantage. However, practical advantage needs to be supported by experiments. Also, the authors acknowledged the larger dependence of their total number of function evaluations on $\epsilon$ than that in [1], which further theoretically undermines the advantage of the proposed algorithm, since the total number of function evaluations is more important in the overall efficiency. To well support the advantage, I suggest to do **either of the following two**.

(1) Add experiments to compare with at least the zeroth order algorithms that find second-order stationary points, in terms of CPU time or total number of function evaluations. Hopefully the proposed algorithm can outperform in most cases.

(2) Improve the proof to reduce the total number of function evaluations to at most that of [1]. You might carefully compare your proof with [1] to see where the additional $\epsilon^{-0.5}$ is imported.

Also, what is your advantage compared with [2]? You could cite [2].

[2] Zhang, H., Xiong, H. and Gu, B. (2022). Zeroth-Order Negative Curvature Finding: Escaping Saddle Points without Gradients. ArXiv:2210.01496.


**Novelty:**

Both the achievement in $2m$ function evaluations per iteration and the techniques to deal with small batchsize $2m$ and large variance without subGaussianity assumption look novel.


**Reproducibility:**

Not applicable as there is no experiment.


**Minor comments:**

(1) In the second paragraph of Introduction, you might say "several earlier works on stochastic gradient methods (Jin et al., 2017; 2018)" to correspond to "stochastic gradient methods" at the beginning of the third paragraph.

(2) In Lemma 3, should $j\ne 2$ be $j\ge 2$?

**Strength And Weaknesses:**

Pros: Both the achievement in $2m$ function evaluations per iteration and the techniques to deal with small batchsize $2m$ and large variance without subGaussianity assumption look novel. The overall structure and main contribution of this paper look very clear.

Cons: The advantage of the proposed algorithm is not well supported as I elaborated in "**(my major concern)**" below. The proof intuition could be more direct and brief before diving into technical details, as I elaborated in "**(my second major concern)**" below.

**Summary Of The Paper:**

This work proposes zeroth-order perturbed gradient descent algorithm which is the first zeroth order algorithm to the authors' knowledge that finds second-order stationary point with only $2m$ function evaluations per iteration for any $m\in [1,d]$ ($d$ is the dimensionality), in contrast to $\Omega(d)$ per iteration in previous similar algorithms. The total number of function evaluations is $\widetilde{\mathcal{O}}(d\epsilon^{-2.5})$. New proof techniques are used to deal with small batchsize $2m$ and large variance without subGaussianity assumption.

**Summary Of The Review:**

Both the achievement in $2m$ function evaluations per iteration and the techniques to deal with small batchsize $2m$ and large variance without subGaussianity assumption look novel. The overall structure and main contribution of this paper look very clear. However, since the advantage of the proposed algorithm is not well supported, as I elaborated in "**(my major concern)**" above, I recommend borderline rejection. I would like to raise my rating if the authors can solve my major concern by either adding experiment or improving the total complexity as I elaborated in "**(my major concern)**" above.

---

> ### Author Response · Authors · 2022-11-17
> **Response to Reviewer nRGh**
>
> We thank reviewer nRGh for your appreciation of our paper, as well as the numerous suggestions you raised. Based on your (and other reviewers') feedback, we have revised our paper. A summary of the changes in the revision can be found in our [comment here](https://openreview.net/forum?id=5d_yTyTj646&noteId=Pn5xxt7wjqp).
>
> Below are our replies to your specific questions.
>
> **Is it possible to write 1-2 paragraphs at the beginning of Sections 3.1 and 3.2 to briefly sketch the proof with no equation or only a few core equations, so that we can more easily understand how the function value decrease OR are bounded for large and small gradients? I think that's perhaps not easy but worth trying."**
>
> While properly motivating the proof does require an extended discussion, we agree it may be helpful to add short proof sketch paragraph(s) at the start of Section 3.1 (to explain the intuition behind function decrease at large gradients) and also at the start of Section 3.2 (to explain the intuition behind making progress at saddle points). Accordingly, in the revision, we have added a short paragraph at the start of Section 3.1 and Section 3.2, outlining our overall proof approach as well as key challenges.
>
>
> **(1) Add experiments to compare with at least the zeroth order algorithms that find second-order stationary points, in terms of CPU time or total number of function evaluations. Hopefully the proposed algorithm can outperform in most cases. OR
> (2) Improve the proof to reduce the total number of function evaluations to at most that of [1]. You might carefully compare your proof with [1] to see where the additional [sic] is imported.**
>
> Empirical evidence suggests that two-point estimators can in fact converge faster to second-order stationary points in nonconvex functions with saddle points, compared with existing state-of-the-art zeroth-order algorithms that can escape saddle points, namely PAGD in [1] and ZO-GD-NCF [2], which both use $2d$ function evaluations per iteration to estimate the gradient. In our updated draft, we include simulation results of our algorithm (with two-point estimators), which we refer to as ZOPGD-2pt, with PAGD and ZO-GD-NCF. The test function we evaluated on is an octopus function (of dimensions $d = 10,30$). It is known that the octopus function defined on $\mathbb{R}^d$, which chains $d$ saddle points sequentially, takes exponential (in $d$) time for exact gradient descent to escape [3]; it has thus emerged as a popular benchmark to evaluate and compare the performance of algorithms that seek to escape saddle points. In our plots, we plot the function value against the the number of function evaluations. For completeness, we also plot the performance of exact gradient descent (normalized such that its $x$-axis is also the number of function queries). To account for the stochasticity in the algorithms, for each algorithm, we computed the average and standard deviation over 30 trials, and plotted the mean trajectory with an additional band that represents $1.5$ times the standard deviation. For PAGD, we used the hyperparameters listed in their paper, and for ZO-GD-NCF, we used the code from their Neurips submission.
>
> As we can see in Figure 1 of Appendix G, in both cases, our algorithm reaches the global minimum of the octopus function significantly faster than PAGD and ZO-GD-NCF (approximately 3 times faster than ZO-GD-NCF, and approximately 4 times faster than PAGD), despite our algorithm only using $2$ function evaluations per iteration compared to $2d$ function evaluations per iteration for both PAGD and ZO-GD-NCF. As a sanity check, we note that the number of function evaluations required for PAGD and ZO-GD-NCF to reach the global minimum approximately matches that in Figure 1 of [2]; here the correspondence is only approximate since [2] only plots one trial while we compute the mean and standard deviation of 30 trials. This result suggests that in addition to the theoretical convergence guarantees, there are also empirical benefits to using two-point estimators versus existing $2d$-point estimators in the zeroth-order escaping saddle point literature.
>
> As the reviewer pointed out, the $\epsilon$ dependence in our sample complexity is currently $\frac{1}{\epsilon^{0.5}}$ worse than the result in [1], which uses $\Omega(d)$ function evaluations per iteration. We note that actually, if a function satisfies the strict saddle property, a property that holds in several non-convex optimization problems (e.g. orthogonal tensor decomposition, see [4]), then our sample complexity can improve significantly. We refer the reviewer to point 3 in [this reply to reviewer QY7d](https://openreview.net/forum?id=5d_yTyTj646&noteId=kdUlwwQkAue) for more details.

---

> > ### Author Response · Authors · 2022-11-17
> > **Response to reviewer nRGh (continued)**
> >
> > **What is $\tau'$ in Lemma 1?** The $\tau'$ here is an index that ranges from $1$ to $T$. As the final equation of the result points out (the equation after (2)), we wish to say that for any $1 \leq \tau' \leq T$, the event $\cap_{\tau = 1}^{\tau'}\mathcal{H}_{0,\tau} (\delta)$ holds is at least $1 - \frac{5 \tau' \delta}{T}$. We understand that the notation is fairly delicate here, which may be confusing. We have changed our writing of Lemma 1 to make the role of $\tau'$ clearer; hopefully the reviewer finds this helpful.
> >
> > **"Missing in advances in Neurips"** Thank you for catching this typo. We have edited the reference accordingly.
> >
> > **What is our advantage to [2]** Thank you for referring us to this new paper (we had missed this at the time of our initial submission), which is indeed highly relevant. The chief difference is that our paper allows for two-point estimators, while the paper in [2] requires using $2d$ function evaluations per iteration to measure the gradient accurately. In our revision, we have now compared the simulation performance of our algorithm to that in [2].
> >
> > **"...you might say "several earlier works on SGD"** We are not quite sure what the reviewer meant here, but we have edited the corresponding parts in the introduction. Hopefully, we have improved the revision's flow.
> >
> > **"In Lemma 3, should $j \neq 2$ be $j \geq 2$."** Thank you for noting a typo here. We had in fact meant to write $j \neq 1$; since $e_1$ is assumed to be the minimum eigendirection of the Hessian at $x_0$ in Lemma 3, what we wished to express is that for all other $j \neq 1$, the coupled noise sequences $Y$ and $Y'$ share the same value, i.e. $Y_j = -Y_j'$ for any $j \neq 1$.
> >
> > [1]: Vlatakis-Gkaragkounis, E.-V., Flokas, L., and Piliouras, G. (2019). Efficiently avoiding saddle points
> > with zero order methods: No gradients required. Advances in Neural Information Processing
> > Systems, 32.
> >
> > [2]: Zhang, H., Xiong, H., and Gu, B. (2022). Zeroth-order negative curvature finding: Escaping saddle
> > points without gradients. arXiv preprint arXiv:2210.01496.
> >
> > [3]: Du, S. S., Jin, C., Lee, J. D., Jordan, M. I., Singh, A., and Poczos, B. (2017). Gradient descent
> > can take exponential time to escape saddle points. In Advances in neural information processing
> > systems, pages 1067–1077.
> >
> > [4]: Ge, R., Huang, F., Jin, C., and Yuan, Y. (2015). Escaping from saddle points—online stochastic
> > gradient for tensor decomposition. In Conference on Learning Theory, pages 797–842.

---

> > > ### Comment · Reviewer_nRGh · 2022-11-18
> > > **Reviewer nRGh's 2nd reply**
> > >
> > > The authors have addressed my concerns and I increase score to 6.
> > > Just one question: What's the batch size m in your algorithm in experiment? You could add that.

---

> > > > ### Author Response · Authors · 2022-11-18
> > > > **On the batch size $m$**
> > > >
> > > > We thank the reviewer for raising their score in consideration of our replies. For the simulations, we used batch size $m = 1$, i.e. 2 function evaluations per iteration.

---

> > > > > ### Comment · Reviewer_nRGh · 2022-11-18
> > > > > **You might add batch size to the experiment section.**
> > > > >
> > > > > Thanks. You might add batch size m=1 to the experiment section.

---

> > > > > > ### Author Response · Authors · 2022-11-18
> > > > > > **Updated revision**
> > > > > >
> > > > > > We previously wrote in words in Appendix G that we simulated two-point estimators, but we agree with the reviewer it might be clearer to state $m = 1$ explicitly. We have now done so in equation (57) in Appendix G in the updated revision. Thanks!

---

> > > > > > > ### Comment · Reviewer_nRGh · 2022-11-18
> > > > > > > **OK. Thanks.**
> > > > > > >
> > > > > > > OK. Thanks.

---

### Official Review · Reviewer_Rytu · 2022-10-23

**Confidence:** 5
**Correctness:** 4
**Technical Novelty And Significance:** 2
**Empirical Novelty And Significance:** Not applicable
**Recommendation:** 3

**Clarity, Quality, Novelty And Reproducibility:**

The problem is not new in the literature and the complexity results in this manuscript is new only in a limited setting. Moreover, the authors have only founded on the deterministic case.

**Strength And Weaknesses:**

It's interesting to provide analysis of zeroth-order methods in escaping from the saddle points without estimating the Hessian matrix. However, as mentioned in the manuscript, this problem has been already studied in the literature and a sample complexity of ${\cal O}(d/\epsilon^{2})$ has been provided. The authors argue that the major limitation of obtaining this complexity bound is to compute ${\cal \Omega }(d)$ per iteration. However, this is not very convincing and the proposed complexity bound in this manuscript is worse than that of the existing methods.

**Summary Of The Paper:**

In this manuscript, the authors generalize the well-known idea of adding perturbation to avoid converging to the saddle points in nonconvex optimization, to the zeroth-order setting in which only function evaluations are available. They also provide convergence analysis of their zeroth-order perturbed gradient method and show that to finding an $\epsilon$ second-order stationary point, it requires at most ${\cal O}(d/\epsilon^{2.5})$ number of function evaluations.

**Summary Of The Review:**

The results are new only in a limited regime and the existing results outperform the presented ones in the manuscript in most settings.

---

> ### Author Response · Authors · 2022-11-17
> **Response to Reviewer Rytu**
>
> We thank the reviewer for your comments. Based on your (and other reviewers') feedback, we have revised our paper. A summary of the changes in the revision can be found in our [comment here](https://openreview.net/forum?id=5d_yTyTj646&noteId=Pn5xxt7wjqp).
>
> We next reply to their specific concerns below.
>
> **"The results are new only in a limited regime and the existing results outperform the presented ones in the manuscript in most settings."** Our understanding of this concern is that the reviewer's key question is the purpose of studying the two-point estimator case when existing works (e.g. [1]) have already established a sample complexity of $\frac{d}{\epsilon^2}$, and our current complexity result is $\frac{d}{\epsilon^{2.5}}$. In response, we note that (i) two-point estimator has been widely studied in the zeroth-order convex/nonconvex optimization literature, due to their ability to make fast progress for high-dimensional problems where waiting for $2d$ function evaluations per iteration may be too slow (particularly valuable when compute is limited) as well as time-varying online optimization problems where waiting for $2d$ function evaluations may cause the function to change significantly in the mean time.  (ii) It is thus important to understand the ability of such estimators to escape saddle points; there exists no existing work which can show that zeroth-order methods that utilize fewer than $\Omega(d)$ function evaluations can escape saddle points. Our work is the first to demonstrate that this is possible, and give polynomial rates that are tight in terms of the dimension $d$ and almost tight in terms of the epsilon $\epsilon$. (iii) Our experiments suggest that our proposed two-point estimators can actually outperform state-of-the-art zeroth-order estimators that use $2d$ function evaluations per iteration for a benchmark nonconvex function with saddle points which has been designed to be hard for gradient descent to escape from. This indicates that two-point methods can be useful numerically, and suggests the important of establishing the theoretical convergence of these methods. We refer the reviewer to [our response to Reviewer QY7d](https://openreview.net/forum?id=5d_yTyTj646&noteId=58oeYOeIVx) for a more complete discussion of the importance of examining two-point estimators.
>
> **Focus on deterministic noise.** We acknowledge that this is a limitation of our result. Nonetheless, given that there has been no earlier work studying the escaping saddle point properties of two-point estimators, we believe it is valuable to focus first on the deterministic case. Indeed, one of the most important works in the zeroth-order literature, namely [2] by Nesterov and Sponoiky, devotes a significant portion of its results to the study of two-point estimators for deterministic optimization. Finally, we believe it is not hard to use the technical tools we develop (e.g. the high-probability function decrease bound for two-point zeroth order estimators, as well as Proposition 2, which gives a tight concentration bound for the vector sum of products of Gaussians) to generalize our result to the stochastic two-point estimator setting.
>
> [1]: Vlatakis-Gkaragkounis, E.-V., Flokas, L., and Piliouras, G. (2019). Efficiently avoiding saddle points
> with zero order methods: No gradients required. Advances in Neural Information Processing
> Systems, 32.
>
> [2]: Nesterov, Y. and Spokoiny, V. (2017). Random gradient-free minimization of convex functions.
> Foundations of Computational Mathematics, 17(2):527–566.

---

> ### Author Response · Authors · 2022-12-02
> **Any questions regarding our author response?**
>
> Thanks again for your feedback and comments on our paper. This is a friendly reminder that we have made relevant changes to our paper (see a summary of our changes [here](https://openreview.net/forum?id=5d_yTyTj646&noteId=Pn5xxt7wjqp)) and answered your questions (see the link [here](https://openreview.net/forum?id=5d_yTyTj646&noteId=nLr6iFfFyVF)). If you have any further questions, please do let us know.

---

### Official Review · Reviewer_czK6 · 2022-10-25

**Confidence:** 3
**Clarity, Quality, Novelty And Reproducibility:** The paper is well written. The techni…
**Correctness:** 3
**Technical Novelty And Significance:** 3
**Empirical Novelty And Significance:** Not applicable
**Recommendation:** 6

**Strength And Weaknesses:**

Strengths:
- The paper addresses an important problem of the zeroth-order optimization.
- The theoretical results are novel and interesting: even with 2 samples to estimate the gradient instead of $d$ samples as in existing works, we still find the $\epsilon$ second-order stationary points after $tidle{\mathcal O} (d\/epsilon^2.5)$ function evaluations. This result is valuable for the high-dimensional optimization problem.
- The approach is based on existing works but provides some new techniques.

Weaknesses:
- Although the number of evaluations to estimate gradient is reduced from $\mathcal O(d)$ to $2m$, however, the number of evaluations for convergence guarantees increases compared to existing works. Overall, it seems that their work has no progress compared to existing works. Therefore, the parameter $m$ can be considered as a trading coefficient between the number of samples to estimate gradients and the performance of the optimization.
- The paper is so long to estimate. I've checked some proof for the case when the gradient is large, but could not check all proofs.


**Summary Of The Paper:**

This paper addresses the zeroth-order optimization of black-box functions. Due to the unavailability of gradients, gradient estimations are used. The paper proposes an algorithm as follows: at each iteration, it uses only $2m$, where $1 \le m \le d$ and $d$ is the input dimension, to approximate gradients. Then it adds an isotropic perturbation to the gradient estimate in order to escape from saddle points. The authors show that their algorithm can find $\epsilon$ second-order stationary points using only $tidle{\mathcal O} (d\/epsilon^2.5)$ function evaluations. The idea of their convergence analysis is based on that of Jin et al. (2019a) with non-trivial modifications. If the gradient estimate is large, they show that the function can decrease even with only $2m$ evaluations at every iteration. If the current point is near saddle points, they show that there is a constant probability of making the function value decrease.



**Summary Of The Review:**

I am toward accepting this paper.

---

> ### Author Response · Authors · 2022-11-17
> **Response to reviewer czK6**
>
> We thank the reviewer for your appreciation of our paper. Based on your (and other reviewers') feedback, we have revised our paper. A summary of the changes in the revision can be found in our [comment here](https://openreview.net/forum?id=5d_yTyTj646&noteId=Pn5xxt7wjqp).
>
> Below, we answer the specific concerns raised by the reviewer.
>
> **Progress relative to existing works.** We thank the reviewer for raising this question. For a discussion of the importance of studying using two-point estimators to escape saddle points in zeroth-order optimization (as opposed to using $\Omega(d)$ function evaluations per iteration), we refer the reviewer to [our response here to QY7d, broken into three replies](https://openreview.net/forum?id=5d_yTyTj646&noteId=58oeYOeIVx). In particular, we would like to highlight that (a) two-point algorithms are particularly important for high-dimensional and or time-varying problems where waiting for $2d$ function evaluations can be impractical (due to compute limitations, or drifting of the underlying problem in the online case) (b) in practice, our algorithm (using only two function evaluations per iteration) may actually require fewer function evaluations to converge than state-of-the-art zeroth order algorithms that use $2d$ function evaluations per iteration. This holds true for escaping saddle points of a test nonconvex functions proposed in [1]. For more details regarding the numerical simulations, please refer to [our response to Reviewer nRGh](https://openreview.net/forum?id=5d_yTyTj646&noteId=KODtxzIzwq), or the discussion in our newly added simulation section Appendix G, for a more complete discussion.
>
> [1]: Du, S. S., Jin, C., Lee, J. D., Jordan, M. I., Singh, A., and Poczos, B. (2017). Gradient descent
> can take exponential time to escape saddle points. In Advances in neural information processing
> systems, pages 1067–1077.
>
> **Proofs are too long.** We acknowledge that our proofs are quite long. In our revision, we have now added short paragraphs in front of Section 3.1 (Function decrease in large gradient regime) and Section 3.2 (Saddle point function decrease) explaining the key challenges as well as our high-level proof outline. In addition, at the start of each appendix, we have now added preclude paragraphs indicating our overall strategy. Hopefully these additions will be helpful.

---

### Official Review · Reviewer_QY7d · 2022-10-27

**Confidence:** 5
**Correctness:** 3
**Technical Novelty And Significance:** 3
**Empirical Novelty And Significance:** 3
**Recommendation:** 3

**Clarity, Quality, Novelty And Reproducibility:**

Clarity: The paper is well-written as far as the maths BUT I strongly believe that the authors should add prelude paragraphs in the appendices to keep intuitions. Giving repetitively examples before or after the statements would only benefit the algorithm.

Quality: It involves technical machinery well established last decade in non-convex optimization. The probabilistic machinery had some independent interest

Novelty: In my opinion, the most interesting tool is the derivation of a potential which includes the constant addition of noise in such a detailed form. It is indeed interesting the ability to have even some non-negligible progress with constant number of samples per round. (However, I am not convinced that this is the case...Can the authors elaborate more about that?)

**Details Of Ethics Concerns:**

Non-applicable

**Strength And Weaknesses:**

In order to understand if the model is really interesting, I would like to ask a model where waiting to get d samples can not be done and it is crucial to have some small progress at every round because it improves some regret quantity. Additionally, the authors claimed that adding constantly isotropic noise instead of sporadically is better, because the algorithm of Flokas et al (the optimal one in combination iteration x samples) has to check the gradient if it is small. This is wrong, by looking the referred algorithm, it examines the ''approximate gradient'', which will be used for the descent (so we have access to that for free).

I am wondering, if the authors start to add a noise adaptive, if they will achieve better rate of convergence.

**Summary Of The Paper:**

The authors provide a best-iterate convergence rate for a zeroth-order optimization with constant number of function evaluations per iteration which can escape saddle points efficiently in order to compute a second-order stationary point.

**Summary Of The Review:**

My honest goal is to observe the discussion, since I am not convinced about the importance of the model.
To be more precise, why someone should not just wait to get d estimates to approximate grad with finite differences, since the final iterations \times samples = complexity would be the same, or even better

---

> ### Author Response · Authors · 2022-11-17
> **Why not wait to get $\Omega(d)$ function evaluations per iteration?**
>
> First, we would like to apologize that we took it for granted in our initial submission that it was important to study how zeroth-order methods with 2 or $2m$ (where $m$ is a constant)-points estimator escape saddle point. As a result, we did not sufficiently motivate our work in the initial submission.
>
> In this response, we will first motivate why we started this study of zeroth-order methods with $2$ or $2m$-points estimator (Point 1). Then we will show concrete empirical advantages of $2m$ points estimator in escaping the saddle point over the $2d$ (where $d$ is the problem dimension) points estimator (Point 2). Next, in Point 3, we explain how for strict saddle functions, our theoretical sample complexity results can in fact match that of the best existing zeroth-order results on escaping saddle points using $2d$ function evaluations per iteration; the strict saddle property is recognized as an important property that many nonconvex functions in machine learning satisfy. Finally, we wrap up our discussion in Point 4. Note that we have revised our paper (see the updated revision) accordingly by incorporating most of the discussions.
>
> **Point 1 (motivation to study two-point zeroth-order methods)**  Two (or $2m$ where $m$ is a constant)-point estimators have been widely studied by researchers in the zeroth-order optimization literature, in convex [1,2,3], nonconvex [1], online [3], as well as distributed settings [4,5]. A key reason for doing so is that for applications of zeroth-order optimization arising in wind farms [5],robotics [6], power systems [7], online (time-varying) optimization [3], learning-based control [10,11], and improving adversarial robustness to black-box attacks in deep neural networks [8], it may be costly or impractical to wait for $2d$ (where $d$ denotes the dimension) function evaluations per iteration to make a step.
>
> This is especially true for high-dimensional and/or time-varying problems. Indeed, for high-dimensional problems, two-point estimators can make swift progress even in the initial stage compared to $2d$-point estimator, and can reach a higher-quality solution if computation is limited [4,8]. For instance, consider the performance of the two-point versus $2d$-point zeroth order estimator in Figure 1 of [4] (https://arxiv.org/pdf/1908.11444.pdf), where we see that the two-point estimator makes significantly faster progress in the first half of the algorithm. For another example, consider the work in [8], which studies the use of zeroth-order estimators to perform black-box attacks on deep neural networks, in order to identify (and then defend against) adversarial images that may lead to misclassification. In the paper, the authors employed two-point zeroth-order estimators, due to the high computational cost of using $2d$ function evaluations per iteration for hundreds of iterations (here $d$ is the dimension of an image, which in this case is over 20000); see the discussion near the end of the bullet point "zeroth-order optimization on the loss function" in [9] (https://arxiv.org/pdf/1708.03999.pdf). The authors showed empirically that their two-point estimators worked well, however with no accompanying theoretical results.
>
> For online or time-varying environments, two-points estimators also often preferable. Since zeroth-order methods are often used in physical systems whose environment drifts or changes over time, this leads naturally to a *time-varying* or *online* optimization. For these problems, $2d$-point estimators will not produce a good estimation because the underlying function can drift to a very different problem while waiting for the $2d$ function evaluations. The fewer function evaluations a optimization procedure needs, the faster it can catch up with the time-varying environment. In fact, in online bandit optimization [2], it has been shown that two points estimator is optimal for convex Lipschitz functions.
>
> However, despite the many advantages of zeroth-order methods with two-point estimators,
> there has been a lack of existing work studying the ability of two-point estimators to escape saddle points in nonconvex optimization problems. Since nonconvex problems arise often in practice, it is crucial to know if two-point algorithms can efficiently escape saddle points of nonconvex functions and converge to second-order stationary points. By showing that two-point methods can in fact converge to $\epsilon$-approximate second order stationary points, we provide the first theoretical basis supporting the use of two-point zeroth-order algorithms for nonconvex functions which may contain saddle points.

---

> > ### Author Response · Authors · 2022-11-17
> > **Why not wait to get $\Omega(d)$ function evaluations per iteration? (continued)**
> >
> > **Point 1 (continued)** Indeed, in our own research, we have been using two-point estimators to design reinforcement learning algorithms for continuous control problems and to design multi-agent feedback optimization for various physical-world applications. But in these studies, we have only been able to prove convergence to stationary points because these problems are often nonconvex. Thus, another significant motivation for us to start this study is to understand whether the designed methods would escape saddle points so we have more guarantees for these application problems.
> >
> > [1]: Nesterov, Y. and Spokoiny, V. (2017). Random gradient-free minimization of convex functions.
> > Foundations of Computational Mathematics, 17(2):527–566.
> >
> > [2]: Duchi, J. C., Jordan, M. I., Wainwright, M. J., and Wibisono, A. (2015). Optimal rates for zero-order
> > convex optimization: The power of two function evaluations. IEEE Transactions on Information
> > Theory, 61(5):2788–2806.
> >
> > [3]: Shamir, O. (2017). An optimal algorithm for bandit and zero-order convex optimization with
> > two-point feedback. The Journal of Machine Learning Research, 18(1):1703–1713.
> >
> > [4]: Tang, Y., Zhang, J., & Li, N. (2020). Distributed zero-order algorithms for nonconvex multiagent optimization. IEEE Transactions on Control of Network Systems, 8(1), 269-281.
> >
> > [5]: Tang, Y., Ren, Z., & Li, N. (2020, December). Zeroth-order feedback optimization for cooperative multi-agent systems. In 2020 59th IEEE Conference on Decision and Control (CDC) (pp. 3649-3656). IEEE.
> >
> > [6]: Li, J., Balasubramanian, K., & Ma, S. (2022). Stochastic zeroth-order riemannian derivative estimation and optimization. Mathematics of Operations Research.
> >
> > [7]: Chen, Y., Bernstein, A., Devraj, A., & Meyn, S. (2020, July). Model-free primal-dual methods for network optimization with application to real-time optimal power flow. In 2020 American Control Conference (ACC) (pp. 3140-3147). IEEE.
> >
> > [8]: Chen, P. Y., Zhang, H., Sharma, Y., Yi, J., & Hsieh, C. J. (2017, November). Zoo: Zeroth order optimization based black-box attacks to deep neural networks without training substitute models. In Proceedings of the 10th ACM workshop on artificial intelligence and security (pp. 15-26).
> >
> > [9]: Zhang, H., Xiong, H., and Gu, B. (2022). Zeroth-order negative curvature finding: Escaping saddle
> > points without gradients. arXiv preprint arXiv:2210.01496.
> >
> > [10]: Malik, D., Pananjady, A., Bhatia, K., Khamaru, K., Bartlett, P., & Wainwright, M. (2019, April). Derivative-free methods for policy optimization: Guarantees for linear quadratic systems. In The 22nd international conference on artificial intelligence and statistics (pp. 2916-2925). PMLR.
> >
> > [11]: Li, Y., Tang, Y., Zhang, R., & Li, N. (2021). Distributed reinforcement learning for decentralized linear quadratic control: A derivative-free policy optimization approach. IEEE Transactions on Automatic Control.
> >
> > **Point 2 (empirical benefits)** Empirically, we found evidence suggesting that the two-point estimators we propose can in fact converge faster to second-order stationary points in nonconvex functions with saddle points, compared with existing state-of-the-art zeroth-order algorithms that can escape saddle points, namely PAGD in [12] and ZO-GD-NCF [9], which both use $2d$ function evaluations per iteration to estimate the gradient. We apologize that we did not include any numerical results in our initial submission as we had only treated it as a theoretical paper for studying $2m$-point estimator.  In our updated draft, we include simulation results of our algorithm (with two-point estimators), which we refer to as ZOPGD-2pt, with PAGD and ZO-GD-NCF. The test function we evaluated on is an octopus function (of dimensions $d = 10,30$). It is known that the octopus function defined on $\mathbb{R}^d$, for which any optimization trajectory has to traverse $d$ chained saddle points sequentially, takes exponential (in $d$) time for exact gradient descent to escape [13]; it has thus emerged as a popular benchmark to evaluate and compare the performance of algorithms that seek to escape saddle points. In our plots, we plot the function value against the the number of function evaluations. For completeness, we also plot the performance of exact gradient descent (normalized such that its $x$-axis is also the number of function queries). To account for the stochasticity in the algorithms, for each algorithm, we computed the average and standard deviation over 30 trials, and plotted the mean trajectory with an additional band that represents $1.5$ times the standard deviation. For PAGD, we used the hyperparameters listed in their paper, and for ZO-GD-NCF, we used the code from their Neurips submission.

---

> > > ### Author Response · Authors · 2022-11-17
> > > **Why not wait to get $\Omega(d)$ function evaluations per iteration? (continued)**
> > >
> > > **(Point 2 continued)** As we can see in Figure 1 of Appendix G, in both cases, our algorithm reaches the global minimum of the octopus function significantly faster than PAGD and ZO-GD-NCF (approximately 2 to 3 times faster than ZO-GD-NCF, and approximately 3 times faster than PAGD), despite our algorithm only using $2$ function evaluations per iteration compared to $2d$ function evaluations per iteration for both PAGD and ZO-GD-NCF. As a sanity check, we note that the number of function evaluations required for PAGD and ZO-GD-NCF to reach the global minimum approximately matches that in Figure 1 of [9]; here the correspondence is only approximate since [9] only plots one trial while we compute the mean and standard deviation of 30 trials. For reproducibility, we have added our code to our iclr submission.
> > >
> > > This result suggests that in addition to the benefit of two-point estimator as discussed in the previous bullet point, there might also be empirical benefits to using two-point estimators versus existing $2d$-point estimators in the zeroth-order escaping saddle point literature.
> > >
> > > [12]: Vlatakis-Gkaragkounis, E.-V., Flokas, L., and Piliouras, G. (2019). Efficiently avoiding saddle points
> > > with zero order methods: No gradients required. Advances in Neural Information Processing
> > > Systems, 32.
> > >
> > > [13]: Du, S. S., Jin, C., Lee, J. D., Jordan, M. I., Singh, A., and Poczos, B. (2017). Gradient descent
> > > can take exponential time to escape saddle points. In Advances in neural information processing
> > > systems, pages 1067–1077.
> > >
> > > **Point 3 (Sample complexity improvement in strict saddle case)** In addition,  we note that our sample complexity improves in the case of strict saddle functions. Concretely, for a given $\\epsilon > 0$, we define $f(\\cdot)$ to be an $(\\epsilon,\\psi)$-strict saddle if
> > >     for any point $x$, at least one of the following is true: either $||\nabla f(x)|| \\geq \\epsilon $, or $\\lambda_{\\min}(\\nabla^2 f(x)) \leq -\psi$. There are several important nonconvex optimization problems that satisfy the strict saddle property, such as orthogonal tensor decomposition [12], dictionary learning [13] and phase retrieval [13].
> > >
> > > As we show by slightly modifying the saddle point escape part of our proof (in Appendix E.3 of our updated revision), the number of function evaluations to converge to $\epsilon$-second order stationary points is $\tilde{O}\left(\frac{d}{\bar{\psi} \epsilon^2} \right)$ (see Theorem 1 in our updated revision), where (without loss of generality),
> > > $$\\bar{\\psi} := \psi  \mbox{ if } f(\cdot) \mbox{ is }(\epsilon,\psi)\mbox{-strict saddle}, \mbox{ and }\bar{\psi} := \sqrt{\rho \epsilon}\mbox{ if otherwise.}$$
> > >
> > > Proving this new result takes only a few changes (involving swapping $\sqrt{\rho \epsilon}$ with $\bar{\psi}$ in Appendix E.3), and all changes are made in blue in the revision (in particular, the changes are made in the discussion of the $\gamma$ term following Lemma 3, and in the statements and proofs of Lemma 24, Proposition 6, Lemma 25 and Theorem 2).
> > >
> > > As a result, the sample complexity (measured by number of function evaluations)  scales as $\tilde{\Omega}\left(\frac{d}{ \epsilon^2} \right)$ when $\psi$ is of size $\Omega(1)$. Thus, in this setting, for two-point estimators, the dependence on $d$ and $\epsilon$ in our sample complexity (as measured by function evaluations) matches that achieved by the algorithm in [9,10], which has to use $2d$ function evaluations per iteration to estimate the gradient.
> > >
> > > [12]: Ge, R., Huang, F., Jin, C., and Yuan, Y. (2015). Escaping from saddle points—online stochastic
> > > gradient for tensor decomposition. In Conference on Learning Theory, pages 797–842.
> > >
> > > [13]: Sun, J., Qu, Q., & Wright, J. (2015). When are nonconvex problems not scary?. arXiv preprint arXiv:1510.06096.
> > >
> > > **Point 4.** Finally, we acknowledge that while our rates may not be tight in terms of $\epsilon$, our result is the first work on escaping saddle points using two-point estimators in zeroth-order non-convex optimization, and the new technical tools we introduce in our analysis may prove useful for future researchers who wish to further improve the result for two-point estimators. In particular, analysis for the two-point zeroth order estimator is complicated by the non-subGaussianity of the noise from the zeroth-order term, and we successfully show via a novel result in Proposition 2 how such a term can be bounded. Furthermore, in the general case when the strict saddle condition does not hold, it is unclear if two-point estimators can achieve the same sample complexity rate to reach $\epsilon$-second order stationary points as estimators using $\Omega(d)$ function evaluations per iteration (we note that the octopus function we simulated on satisfies the strict saddle condition). So our work may serve as a foundation for further work that either improves our result, or inspire lower bound results that strictly separate two-point estimators from $2d$-point estimators.

---

> ### Author Response · Authors · 2022-11-17
> **Further responses to questions by Reviewer QY7d**
>
> **Wrong claim?: The algorithm in [12] can check approximate gradient for free.** We thank the reviewer for making this observation. As the reviewer observed, the implementation of the algorithm in [12] only requires checking if the norm of the approximate gradient, $||z_t||$, is larger than $\frac{3}{4} g_{\mathrm{thres}}$, where $g_{\mathrm{thres}} := \frac{\sqrt{c}}{\chi^2} \epsilon$ (see Line 3 in Algorithm 1 of [12]), where $c$ and $\chi$ are depends on the problem dimension $d$ as well as some other user-chosen parameters. In particular, the $\chi$ term appears to depend on $\ell$, which is the Lipschitz constant of the function. Based on our understanding, it is not entirely obvious how the result in [12] may change if $\ell$ is incorrectly estimated. In contrast, in our approach, we show that so long as the perturbation parameter $r$ is smaller than some (problem-dependent) quantity, then our sample complexity result will hold. While perhaps not a major issue in practice, conceptually, the message our algorithm conveys may be simpler (just choose a small enough $r$ and perturb every iteration). Nonetheless, since this is a relatively minor issue, we have decided to remove this point from the main text.
>
> **Adaptive noise and improving convergence rate** This is an interesting suggestion; the challenge is that near saddle points, we need sufficient noise, but not too large such that we can no longer characterize its concentration properties. On a related note, we have some ideas on how adaptively choosing the step-size may in some settings improve the convergence rate. More concretely, based on (our newly added) Remark 1 following our main result (Theorem 1), for the case $m = 1$ (i.e. two-point estimators), we can separate the sample complexity into two terms,
> $$T = \tilde{\Omega} \left(\frac{(f(x_0) - f^*)}{\eta \epsilon^2 } + \frac{\rho^2 (f(x_0) -  f^*)}{ \eta \bar{\psi}^4} \right),$$
> the first term coming from the large gradient iterations, and the second term coming from the saddle point iterations; here $\bar{\psi}$ is the same saddle Hessian minimum eigenvalue parameter we introduced earlier, which in the worst case is $\sqrt{\rho \epsilon}$.
> As we explain in [our response to Reviewr ZbmT](https://openreview.net/forum?id=5d_yTyTj646&noteId=_dd2d3CU9H4), in the large gradient regime, it suffices to set $\eta = \tilde{O}\left(\frac{1}{dL} \right)$, while for the iterates with small gradients, it suffices to set $\eta = \tilde{O}\left(\frac{\bar{\psi}}{dL^2} \right)$
> So, if we can adaptively pick the step size $\eta$, choosing $\eta_1 = \tilde{O}\left(\frac{1}{dL}\right)$ and $\eta_2 = \tilde{O}\left(\frac{\bar{\psi}}{dL^2} \right)$ during the small gradient iterations, then for $(\epsilon,\psi)$-strict saddle functions where $\psi > \sqrt{\rho \epsilon}$ and $\frac{L^2 \rho^2}{\psi^5} \leq \frac{L}{\epsilon^2}$, then we can improve the sample complexity to $\tilde{O}\left(\frac{dL (f(x_0) - f^*)}{\epsilon^2}\right)$, which is the best current upper bound for zeroth-order algorithms to escape saddle points [9,10].
>
> **Add preclude paragraphs for more clarity** Although we have tried hard to make the high-level proof approach clear, since there are many technical lemmas and propositions in our proof, we recognize there may be some results where the intuition is not entirely clear. Hence, we have added a short paragraph in front of Section 3.1 and Section 3.2 in the revision to give more intuition. Furthermore, we have now added a paragraph in front of each appendix illustrating our overall approach for that section.
>
> (response to be continued)

---

> > ### Author Response · Authors · 2022-11-18
> > **Further responses to Reviewer QY7d (continued)**
> >
> >
> > **"It is indeed interesting the ability to have even some non-negligible progress with constant number of samples per round...Can the authors elaborate more about that?".** First, in large gradient regime, it has in fact been known that two-point methods can make progress, at least in expectation [1]. It is possible to use Markov's inequality to derive $(1 - \delta)$ high-probability bounds on function value decrease with an additional sample complexity dependence on $\frac{1}{\delta}$. However, this can make the sample complexity significantly worse when $\delta$ is very small. To the best of our knowledge, we provide the first $(1-\delta)$-high-probability bounds on the function value decrease made across any given number of iterations for two-point zeroth-order algorithms, that have a dependence of $\mathrm{polylog}(\frac{1}{\delta})$ in the sample complexity. The key idea is to rely on the fact that in a small (i.e. $\mathrm{polylog}(\frac{1}{\delta})$) number of iterations, with high probability, there must be a "good" iteration where the zeroth-estimator that is well-aligned with the gradient direction for that iteration. Nonetheless, there can be "bad" iterations where the zeroth-order estimator is not well-aligned with the gradient direction, and the function value can actually increase. Our analysis shows that with high probability, across the iterates with large gradients, the effect of the "good" iterations dominates the effect of the "bad" iterations, allowing the function to make progress. We refer the reviewer to Section 3.1 for a more detailed high-level outline, and to the more detailed proof outline following the statement of Lemma 1 in Appendix D for more details.
> >
> > In terms of making progress near saddle points using two-point estimators, we note that previous works on escaping saddle points using zeroth-order information have only showed how progress can be done using $\Omega(d)$ samples per iteration, as the noise inherent in two-point zeroth-order estimators is hard to characterize and control near the saddle point. The key technical tool that allows us to overcome this issue is Proposition 2 in our paper, from which we can derive a tight bound (in terms of the dimension $d$) on the quantity $W_{g_0}(t+1)$, which is the stochastic quantity that depends on the zeroth-order noise in the coupling argument in Lemma 3 of our paper. This control over $W_{g_0}(t+1)$ allows us to show a similar improvement near saddle point to the first-order case.
> >
> > In addition, empirically, we also observed that two-point estimators are sufficient to escape saddle points and make progress during large gradient iterations, with high-probability. This is demonstrated in the new simulation section in Appendix G which we have added to the draft.
> >
> > [1]: Nesterov, Y. and Spokoiny, V. (2017). Random gradient-free minimization of convex functions. Foundations of Computational Mathematics, 17(2):527–566.

---

> ### Author Response · Authors · 2022-11-17
> **Response to Reviewer QY7d**
>
> We thank the reviewer for raising several important questions, in particular the motivation for studying the two-point zeroth order estimator. Based on your (and other reviewers') feedback, we have revised our paper. A summary of the changes in the revision can be found in our [comment here](https://openreview.net/forum?id=5d_yTyTj646&noteId=Pn5xxt7wjqp).
>
> In our replies to your specific questions, we will first explain the motivation for studying the two-point zeroth order estimator in [this response (broken into three parts)](https://openreview.net/forum?id=5d_yTyTj646&noteId=58oeYOeIVx). We will then address the other questions raised by the reviewer, in [this response](https://openreview.net/forum?id=5d_yTyTj646&noteId=FtLrVd4dCH).

---

> ### Author Response · Authors · 2022-12-02
> **Any questions regarding our author response?**
>
> Thanks again for your feedback and comments on our paper. This is a friendly reminder that we have made relevant changes to our paper (see a summary of our changes [here](https://openreview.net/forum?id=5d_yTyTj646&noteId=Pn5xxt7wjqp)) and answered your questions (see an outline of our response to your questions [here](https://openreview.net/forum?id=5d_yTyTj646&noteId=TT9XuE_yLi)). If you have any further questions, please do let us know.

---

### Author Response · Authors · 2022-11-18
**Summary of changes in the revision**

We have made numerous changes to our draft in our revision (many thanks to the reviewers for their helpful feedback). For the reviewer's reference, we list the main changes here. Note that the changes in the revision are all in blue font.

1. Rewritten introduction to provide motivation for studying the two-point (rather than $2d$)-point estimator. The key motivation is that waiting for $\Omega(d)$ function evaluations per iteration is not practical/suitable in high-dimensional and/or time-varying online optimization problems. For more details on the importance and advantages of two-point estimators in zeroth-order optimization, please refer to [our response to Reviewer QY7d (broken) into three replies](https://openreview.net/forum?id=5d_yTyTj646&noteId=58oeYOeIVx).

2. Updated theorem 1 such that the sample complexity is $\tilde{O}(\frac{d}{\epsilon^2 \bar{\psi}})$, where for $(\epsilon,\psi)$-strict-saddle functions (see Definition 3 in the revision for a definition of strict-saddle functions), $\bar{\psi} := \psi$ (and for general functions, $\bar{\psi} := \sqrt{\rho \epsilon}$). In settings where the strict saddle parameter is $\Omega(1)$ and does not depend on $\epsilon$, this improves the sample complexity to $\tilde{O}(\frac{d}{\epsilon^2})$. Accordingly, all relevant proofs have been updated in the appendix. We note that strict-saddle functions are an important class of functions in nonconvex optimization, with numerous examples such as orthogonal tensor decomposition [1], dictionary learning [2] and phase retrieval [2].

3. Added a "Challenges" and "High-level proof outline" paragraph in front of Section 3.1 (function decrease for large gradients) and Section 3.2 (function value progress near saddle points). We have also added prelude paragraphs in front of each appendix to provide the readers with the overall strategy of each appendix. Hopefully these additions can improve the clarity of the technical exposition.

4. Included a simulation section in Appendix G, illustrating that our proposed (two-point) algorithm takes fewer function evaluations to converge than existing state-of-the-art zeroth-order algorithms that use $2d$ function evaluations per iteration for a test function. For more details, please refer either to Appendix G or to [our response to Reviewer nRGh](https://openreview.net/forum?id=5d_yTyTj646&noteId=KODtxzIzwq). For reproducibility, we have also added our code to the submission.

[1]: Ge, R., Huang, F., Jin, C., and Yuan, Y. (2015). Escaping from saddle points—online stochastic
gradient for tensor decomposition. In Conference on Learning Theory, pages 797–842.

[2]: Sun, J., Qu, Q., & Wright, J. (2015). When are nonconvex problems not scary?. arXiv preprint arXiv:1510.06096.

---

### Decision · Program_Chairs · 2023-01-20

**Decision:**

Reject

**Justification For Why Not Higher Score:**

The work does not improve existing results and uses a batch setting, which is not needed in practice.

**Justification For Why Not Lower Score:**

N/A

**Metareview: Summary, Strengths And Weaknesses:**

The authors use a (batch-based) zero-th order gradient estimates to show avoidance results for non-convex minimization problems.

The obtained sample complexity is worse than the state-of-the-art and does not involve stochastic gradient estimates though the authors use added isotropic noise as a tool for avoidance.